# Organizing space through saccades and fixations between primate posterior parietal cortex and hippocampus

Marie E. Vericel [1], Pierre Baraduc[2], Jean-René Duhamel [1] & Sylvia Wirth [1]

The primate posterior parietal cortex (PPC) withholds a unified representation of the visual space supporting visual exploration, while the hippocampus (HPC) provides a memory-based cognitive place map of the environment. To probe the interactions between these two representations, i.e. between view and place, we compared neural activity in the two regions of macaques navigating a virtual maze. We show that a large proportion of PPC neurons displayed spatial selectivity, along with the HPC. We hypothesized that such modulation by self-position might stem from visual cues processing through saccades and fixations. Accordingly, we found saccade-modulated neurons and cells driven by direct fixations on maze paths or landmarks in both brain regions. These populations of "path" and "landmark cells" gave rise to task-relevant maze segmentation, specific to each region. Finally, both regions anticipated landmarks before they appeared in the field of view, suggesting a shared knowledge of the spatial layout. Altogether, these findings highlight the neural processes that make up place, combining visual exploration of objects in space with memory-driven actions.

While we navigate skillfully from our desks into virtual space, this competence is not a human prerogative. Non-human primates and rodents can find their way in virtual environments too, avoiding obstacles and planning trajectories to reach goals[1–10]. Then, how can a sense of space arise from visual-only stimulation? Optic flow suffices to create an illusion of self-motion and to provide heading informing on travel direction[11]. Further, visual processing of a scene alone allows to identify relevant cues for navigation, such as objects, landmarks, paths or borders, providing goals to reach, and framing them in self-based or other-objects-based spatial coordinates[12–15]. In comparison with the place-selective cells typically described in rodents hippocampus (HPC)[16–18], it has been shown that primate HPC neurons are active when the animal looks at specific landmarks and views of the environment, in a task-driven manner during virtual navigation[5,10,19] or random foraging in real-world[20], suggesting a role in the encoding of a map of the visual space. However, its processing by saccades and fixations is known to arise from neural computations performed in the posterior parietal cortex (PPC), notably in the intraparietal areas[21–24]. In the lateral and the ventral intraparietal areas (LIP and VIP), these computations aim at tracking self- and objects motions, and optic-flow direction and velocity[25–27], the direction of moving stimuli[28–30], or their saliency[31–33]. Further, they encode spatial coordinates of visual stimuli in a continuum between eye- and head-centered reference frames[22,34–36]. Finally, LIP also supports a continuous and stable visual space taking into account the programming of ocular saccades during visual exploration[30,37–43]. In sum, the studies suggest that parietal cortex, along with hippocampal regions, may greatly be recruited during navigational tasks[21,44,45]. However, while rodents PPC neurons were shown to display route-selective activities[46,47] or heading angle[48], there have been no studies that characterized activity in intraparietal regions during landmark-based navigation in the non-human primate.

[1]Institut des Sciences Cognitives Marc Jeannerod, UMR 5229, CNRS & Université Claude Bernard Lyon 1, Bron, France. [2]GIPSA-lab, UMR 5216, CNRS, Grenoble-INP-UGA & Université Grenoble-Alpes, Saint Martin d'Hères, France. ✉e-mail: marie.vericel96@gmail.com; sylvia.wirth@isc.cnrs.fr

Anatomically, parietal inputs to hippocampus take several poly-synaptic pathways *via* the retrosplenial, posterior cingulate, or the parahippocampal cortices before reaching entorhinal cortex or subiculum[49–52]. Further, the hippocampal formation projects back to parietal regions via the reverse path[50,51] or directly[52]. These data, together with lesioning experiments in rodents[53–56], provide more ground to support an interplay between hippocampus and PPC for navigation[57,58].

To understand the nature of neural activity in PPC during a goal-directed, virtual-reality task, we compared the activity of cells of LIP and VIP with that of hippocampal ones, as a function of virtual position, and as a function of saccades and fixations to landmarks. We hypothesized that a "place" representation in PPC would derive from the processing of relevant salient cues of the visual space, and that this representation would complement the role of hippocampus in cues identification and environment mapping. Our results show how parietal cortex and hippocampus displayed position-related activities resulting from a dynamic processing of visual cues at both different and shared strategic task moments.

## Results

### Parietal cells display position selectivity as strongly as in the hippocampus

We analyzed the activity of 111 cells recorded by laminar U-probes (Plexon®) in a continuum between the lateral and ventral banks of the intraparietal sulcus (PPC cells; Supplementary Fig. 1a, b) and 142 cells recorded throughout the whole hippocampus in the right hemisphere (HPC cells; the anatomical coordinates of the recording sites can be found in[10], while macaque monkeys navigated a virtual 5 arms star-maze to obtain a hidden reward. Animals learned to locate the reward based on its position relative to five salient landmarks (see Fig. 1a and[10]). In all figures, we adopted the convention to orient the maze with the reward at North. During the session, animals started each trial by moving from the beginning of one of four inbound paths towards the center of the maze (Fig. 1a, steps 1 and 2, purple paths). Then they chose to enter another path with the joystick and learned via trial and error which path led to the reward (steps 3 and 4, red path). Following the reward delivery, they were passively allocated to a new start point through the "outbound" paths (step 5, dotted lines). The first-person view was uninterrupted between the trials (see bottom panel of Fig. 1a). Following an initial shaping (see "Methods") aimed to train the animals on the task rules, the landmarks identities were changed every day and monkeys quickly solved the task within each session, reaching 90% of correct trial after about 20 trials[10]. For this study, only the correct trials were selected for analyses. In this way, the path chosen by the animal was always the rewarded path, oriented toward the North on our maps.

We first examined whether parietal or hippocampal cells were modulated by animal's virtual position, i.e. the camera's location within the environment (see "neural place maps" on Fig. 1b–m, Supplementary Fig. 2 and Supplementary Fig. 3, with PPC maps outlined in red, and HPC in green; see "Methods"). Not only were parietal cells more active than hippocampal ones (mean firing rate: $\mu FR_{PPC} = 6.19 \pm 1.89$ [mean ± 95% confidence interval] spikes/sec, $\mu FR_{HPC} = 2.97 \pm 0.75$ spikes/sec; Wilcoxon rank sum: $|Z| = 4.04$, $p = 5.27 \times 10^{-5}$; peak firing rate: $\mu peak_{PPC} = 14.96 \pm 3.46$ spikes/sec, $\mu peak_{HPC} = 7.54 \pm 1.35$ spikes/sec; Wilcoxon rank sum: $p = 1.41 \times 10^{-5}$), but more PPC cells displayed activity significantly modulated by position (91/111, 82.0%; defined as information content above that obtained from permutations, see "Methods") than in the hippocampus (64/142, 45.1%; homogeneity Pearson's Chi-squared with Yates' correction: $X^2_{1df} = 34.23$, $p = 4.90 \times 10^{-9}$; see Fig. 1b–m for examples of cells with significant modulation by position and Supplementary Fig. 2 for examples of cells with activity not significantly modulated by position). ANOVAs with repeated-measures (see "Methods"), on various selectivity measures, such as the Information Content (IC; $\mu IC_{PPC} = 0.34 \pm 0.048$, $\mu IC_{HPC} = 0.45 \pm 0.094$; $F_{1df} = 2.15$,

$p = 0.14$), sparsity index (S; $\mu S_{PPC} = 0.76 \pm 0.022$, $\mu S_{HPC} = 0.73 \pm 0.035$; $F_{1df} = 0.75$, $p = 0.39$), or depth of tuning index (DT; $\mu DT_{PPC} = 0.76 \pm 0.031$, $\mu DT_{HPC} = 0.75 \pm 0.039$; $F_{1df} = 0.0025$, $p = 0.96$) of the spatially modulated cells, showed that parietal neurons were as selective to virtual position as hippocampal cells (see Supplementary Table 1 for an overview of the spatial selectivity indices of the cells shown in Fig. 1 and Supplementary Fig. 1). On the other hand, both S and DT (the analysis could not be performed on IC, since the index varies according to sample size) varied significantly according to maze segments (S: $F_{2df} = 166.33$, $p = 1.60 \times 10^{-49}$, DT: $F_{2df} = 327.29$, $p = 9.74 \times 10^{-77}$): selectivity was higher in the inbound ($\mu S = 0.69 \pm 0.035$, $\mu DT = 0.86 \pm 0.030$) and outbound ($\mu S = 0.69 \pm 0.034$, $\mu DT = 0.87 \pm 0.028$) paths than in the reward one ($\mu S = 0.85 \pm 0.024$, $\mu DT = 0.54 \pm 0.043$) for both the PPC and HPC cells. Thus, cells displayed a stronger modulation of amplitude as a function of position in the inbound and outbound paths than in the rewarded path. In both areas, cells often expressed multiple spatial fields (SFs; see "Methods"), but they didn't differ across regions by SFs number ($\mu N_{PPC} = 3.65 \pm 0.33$ SFs, $\mu N_{HPC} = 3.66 \pm 0.34$ SFs; Wilcoxon rank sum: $|Z| = 0.067$, $p = 0.95$) or Euclidean distance separating them ($\mu Dist._{PPC} = 12.68 \pm 0.86$ units, $\mu D_{HPC} = 11.88 \pm 1.12$ units in the HPC; $|Z| = 0.85$, $p = 0.40$). Because the spatial properties of the recorded neurons may echo rodent place cells[59,60], we conducted additional analyses in order to comprehend to which extent they resemble or differ from place cells.

First, 51/91 (56.0%) of PPC position-modulated cells and 26/64 (40.6%) of HPC ones displayed a stable pattern of spatial modulation across the first and second half of the recording session (see "Methods"). The spatial stability of the example cells of the Fig. 1b–m can be appreciated on the corresponding position rasters presented in Supplementary Fig. 4. These rasters were created by aligning the neurons' activity as a function of the animal's virtual position in the maze segment containing its peak FR (see "Methods"). Stability of the spatial modulation was related to the number of fields and to their size: cells with fewer and larger fields were more stable through time than cells with numerous and smaller fields (Pearson correlation between spatial correlation coefficient and number of SFs: $r_{PPC} = -0.30$, $p_{PPC} = 0.0043$; $r_{HPC} = -0.46$, $p_{HPC} = 2.44 \times 10^{-4}$; Supplementary Fig. 5a, b; correlation with SFs size: $r_{PPC} = 0.51$, $p_{PPC} = 2.81 \times 10^{-7}$; $r_{HPC} = 0.65$, $p_{HPC} = 1.43 \times 10^{-8}$; Supplementary Fig. 5c, d). Further, neurons with higher peaks rates were more stable than others, in both PPC ($r = 0.38$, $p = 2.13 \times 10^{-4}$) and HPC ($r = 0.41$, $p = 8.12 \times 10^{-4}$, Supplementary Fig. 5e, f). These results suggest that, in addition to cells presenting a stable modulation, many displayed brief local modulation unlike rodent place cells. Next, we examined whether position-modulated cells were sensitive to head orientation. This analysis could only be performed in the center of the maze, since in other locations, place and direction covaried (see "Methods").

Amongst the cells previously identified as modulated by spatial position, a large proportion of the PPC (76/91, 83.5%) and the HPC cells (35/61, 57.4%), were also significantly modulated by the camera's orientation in the center of the maze (Supplementary Fig. 6). The depth of tuning for orientation was not significantly different between the two areas ($\mu DT_{PPC} = 0.58 \pm 0.047$, $\mu DT_{HPC} = 0.60 \pm 0.065$; Wilcoxon rank sum: $|Z| = 0.39$, $p = 0.69$) suggesting a similar sensitivity to visual cues as a function of orientation in both regions. The results suggest that the spatially-modulated PPC or HPC cells were not akin to classic rodent place cells, which encode the animal's position per se and are invariant to head orientation[61] (though recent works have evidenced orientation-modulated responses in hippocampal cells of rodents[1,17,62,63]). Rather, many cells displayed brief and transient spatial modulation and selectivity to orientation in addition to position, which may be driven by other variables than position itself, such as direction or view.

Further, in PPC, the visual inspection of PPC cells bearing multiple response fields suggested that these were located in similar positions

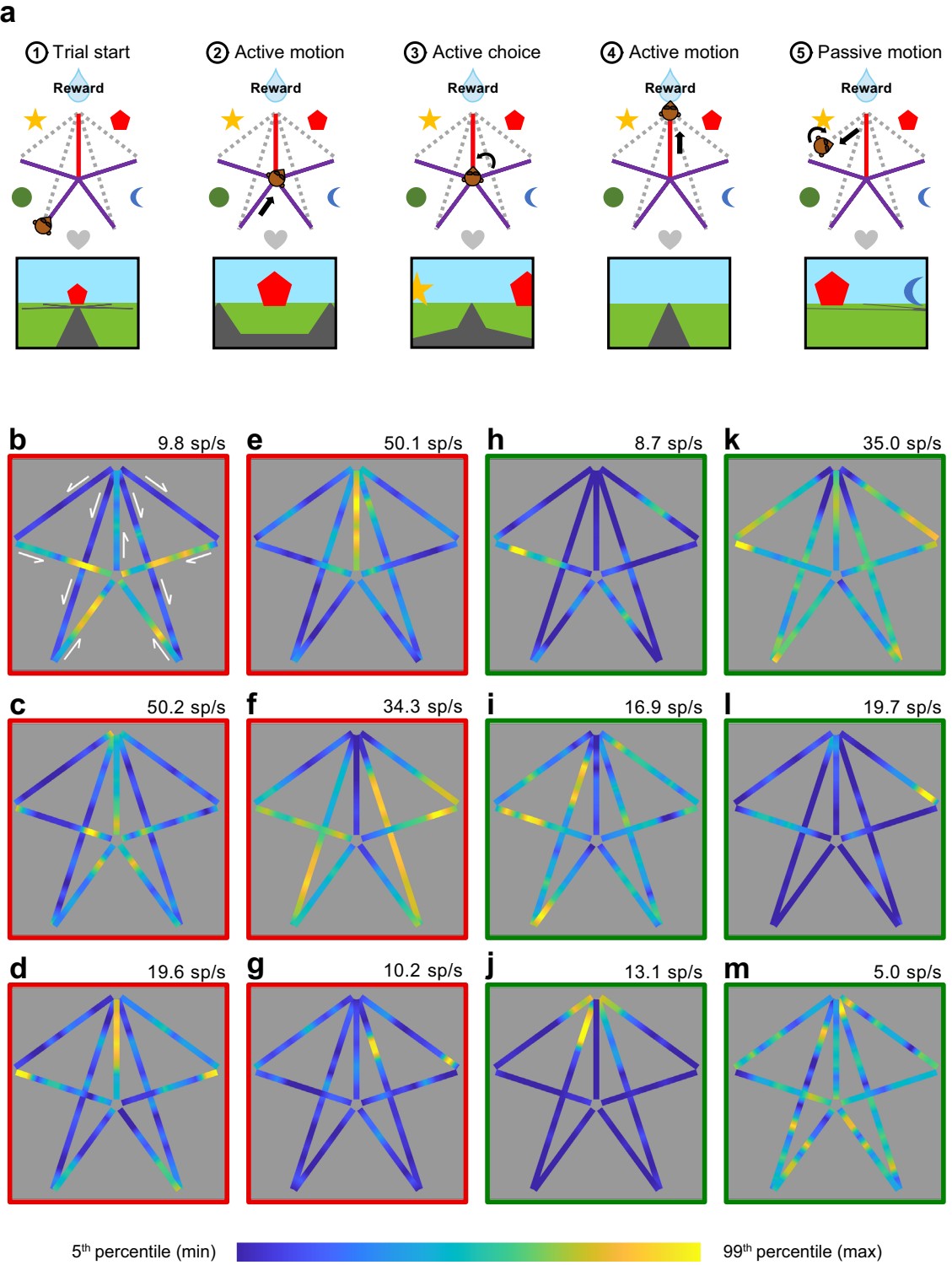

5th percentile (min) 99th percentile (max)

across paths: cells fired for instance at entry positions for all 4 inbound paths, as in example Fig. 1d, or at the end of the inbound paths as in Fig. 1b, c. To assess this apparent symmetry for all the position-modulated cells, we calculated separately the correlation coefficients ($r$) for activity in all inbound and all outbound paths. These correlation coefficients were modestly but significantly higher in the PPC ($\mu r = 0.15 \pm 0.034$) than in the HPC ($\mu r = 0.099 \pm 0.041$; Wilcoxon rank sum: $|Z| = 2.27$, $p = 0.024$), suggesting that cells are recruited for sensorimotor factors that repeat across segments (see examples of symmetrically-spatially-modulated neurons in Supplementary

Fig. 3a, b). We also found a substantial number of cells that displayed asymmetric activity, compatible with similar sensori-motor factors between segments on the same side of the maze. Indeed, when dividing the star-maze along the vertical axis centered on the rewarded path, a substantial fraction of the position-selective PPC (36/91, 39.6%) and HPC (26/64, 40.6%) cells showed greater activity in one side than the other for inbound paths, outbound or both (Wilcoxon rank sum: $p \le 0.05$, see Supplementary Fig. 3c–f). However, at the population level, and for either the inbound paths, the outbound or the whole maze, we observed no preferences for one side over the other, neither

**Fig. 1 | The star-maze and spatial selectivity in posterior parietal and hippo-campal neurons. a** Main steps of the navigation task performed by animals during the recordings, with corresponding points of view below: (1) the trial began from the extremity of one of the four inbound paths of the star-maze (purple); (2) the animal pushed the joystick to move the camera and reach the maze center; (3) in the center, they pushed the joystick left or right as desired, and then chose an exit path by pushing it forward; (4) if animals chose the correct path (red), they obtained a liquid reward; (5) next, the animal was passively moved to the next starting position through the outbound paths (grey dashed lines), via automatic camera rotation and translation, and the next trial began. During the whole session, the first-person view of the scene (bottom panels schematics of the view) was uninterrupted between trials. Five landmarks placed between the inbound paths provided the only visual cues for orientation, since nothing was directly indicating the reward position. The task was presented on a large 152 × 114 cm screen using a

projector, with a horizontal FOV of 74°. Stereopsis was obtained via a pair of shutter glasses synchronized with the projector. **b**–**g** Heat maps of the firing rate of single unit examples of PPC neurons, as a function of monkey's position into the star-maze (i.e. "neural place maps"). The color axis represents the firing rate (maximal firing rate, in spikes/sec, indicated on the top left of each map). All cells had a significant position information content (IC). For a better visualization, data are displayed from the 5th (dark blue; see color bar on the bottom) to the 99th percentiles (bright yellow). The white arrows indicate the direction of camera movements in each maze segment. **h**–**m** Same conventions as for **b**–**g**, for HPC example cells. All cells had a significant position information content (IC). Figure 1a adapted from Wirth S, et al. Gaze-informed, task-situated representation of space in primate hippocampus during virtual navigation. *PLOS Biology*, 15: e2001045 (2017) under a CC BY license: https://creativecommons.org/licenses/by/4.0/.

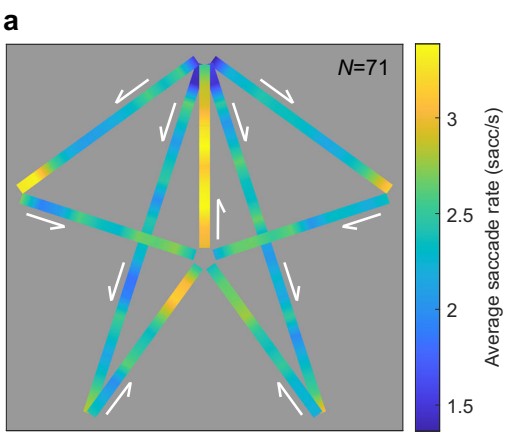

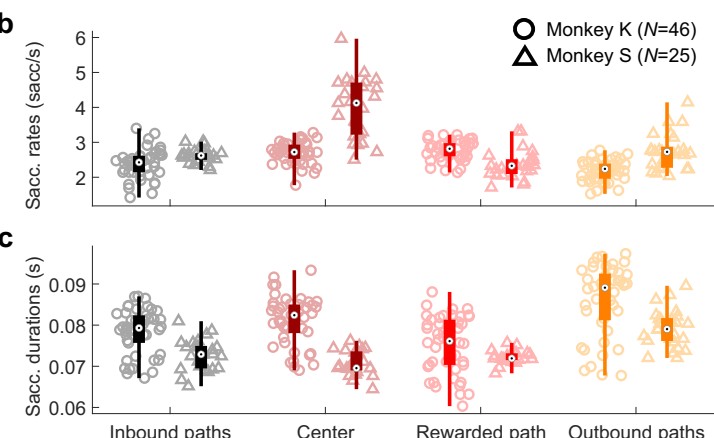

**Fig. 2 | Monkeys' saccadic behavior. a** Averaged position heat map for saccadic behavior, for all the sessions recorded ($N = 71$). The color axis represents the frequency of saccades (saccades/sec) as a function of the animal's position in the maze. The white arrows indicate the direction of camera movements in each maze segment. **b**, **c** Scatter and associated box plots of the saccade rates (**b**) and durations (**c**) of Monkey K ($N = 46$ sessions; circles) and S ($N = 25$ sessions; triangles), for each session (total $N = 71$), depending on the animals' position in the maze. The top and bottom limits of the boxes are the 75th and 25th percentiles of the sample,

respectively; the central dot is the median; the whiskers go to the higher and lower adjacent values. For a better visibility, the outliers are not represented. Source data are provided as a Source Data file. The saccades rates and durations were significantly different between the 4 maze segments in each monkey (rates: Monkey K: Kruskal–Wallis: $X^2_{3df} = 65.14$, $p = 4.69 \times 10^{-14}$, Monkey S: $X^2_{3df} = 48.17$, $p = 1.96 \times 10^{-10}$; durations: Monkey K: $X^2_{3df} = 36.76$, $p = 5.16 \times 10^{-8}$, Monkey S: $X^2_{3df} = 37.29$, $p = 4.00 \times 10^{-8}$).

in the PPC (Wilcoxon matched pairs signed-rank: $|Z_{inbound}| = 1.24$, $p_{inbound} = 0.22$; $|Z_{outbound}| = 0.063$, $p_{outbound} = 0.95$; $|Z_{whole}| = 0.25$, $p_{whole} = 0.81$; see "Methods") nor in the HPC ($|Z_{inbound}| = 0.31$, $p_{inbound} = 0.75$; $|Z_{outbound}| = 0.094$, $p_{outbound} = 0.93$; $|Z_{whole}| = 0.73$, $p_{whole} = 0.47$). This suggests that the preferences observed at the single unit level were overall counterbalanced and thus that no relation between asymmetry and the recorded hemisphere could be found.

Finally, we tested whether cells were modulated by left or right joystick moves, which could explain some of the asymmetries. Briefly, 29/91 (31.9%) parietal and 19/64 (29.7%) hippocampal spatially-modulated neurons displayed significant hand-movement-related activation (see "Methods"), but only 5/91 (5.5%) PPC and 5/64 (7.8%) were significantly selective to left or right movements independent from the orientation of the animal at the moment of the motion. This excluded the hypothesis of purely motor-driven neurons in both areas, rather suggesting that the cells were mainly modulated by sensory inputs.

In sum, parietal neurons displayed strong modulation as a function of virtual position, and the pattern of position-related activity in individual examples suggested that PPC neurons were more recruited for specific task-states, regardless of the path identity, than neurons in the hippocampus. These spatially-modulated neurons in the HPC and the PPC displayed distinct properties than these of the classical place cells. Finally, varying positions in the inbound and outbound paths

triggered stronger selectivities in both areas, compared with the rewarded path, where the visual scene was poorer. In this context, we sought to determine whether these spatial selectivities were primarily attributed to bottom-up visual information or to the top-down systematic exploratory behavior of the animals, rather than to motor events, with respect to the visual maze elements.

**Position selectivity is not exclusively driven by saccade-related activity**

A position in virtual reality is solely provided by specific position-dependent visual information, which could drive specific behaviors, such as explorative saccades and fixations. Saccades occurred at an average rate of $2.68 \pm 0.076$ sacc/sec, for a duration of $77.83 \pm 0.89$ ms (Supplementary Fig. 7 displays example eye-traces on which saccades are identified). The long duration of saccades can be explained by the structural layout of the task maze, in which animals freely moved eyes from one part of the screen to the other, as elements (landmarks, paths) of the environment entered or exited the field of view (see also[3,64]). The individual (Supplementary Fig. 8), and averaged (Fig. 2a) heat maps of the saccades' frequencies as a function of monkey's position (saccades maps; see "Methods") show that they were mostly performed (1) on inbound paths before the center of the maze, (2) in the rewarded path and (3) at the end of the outbound paths, thereby reflecting gazing at landmarks and reward

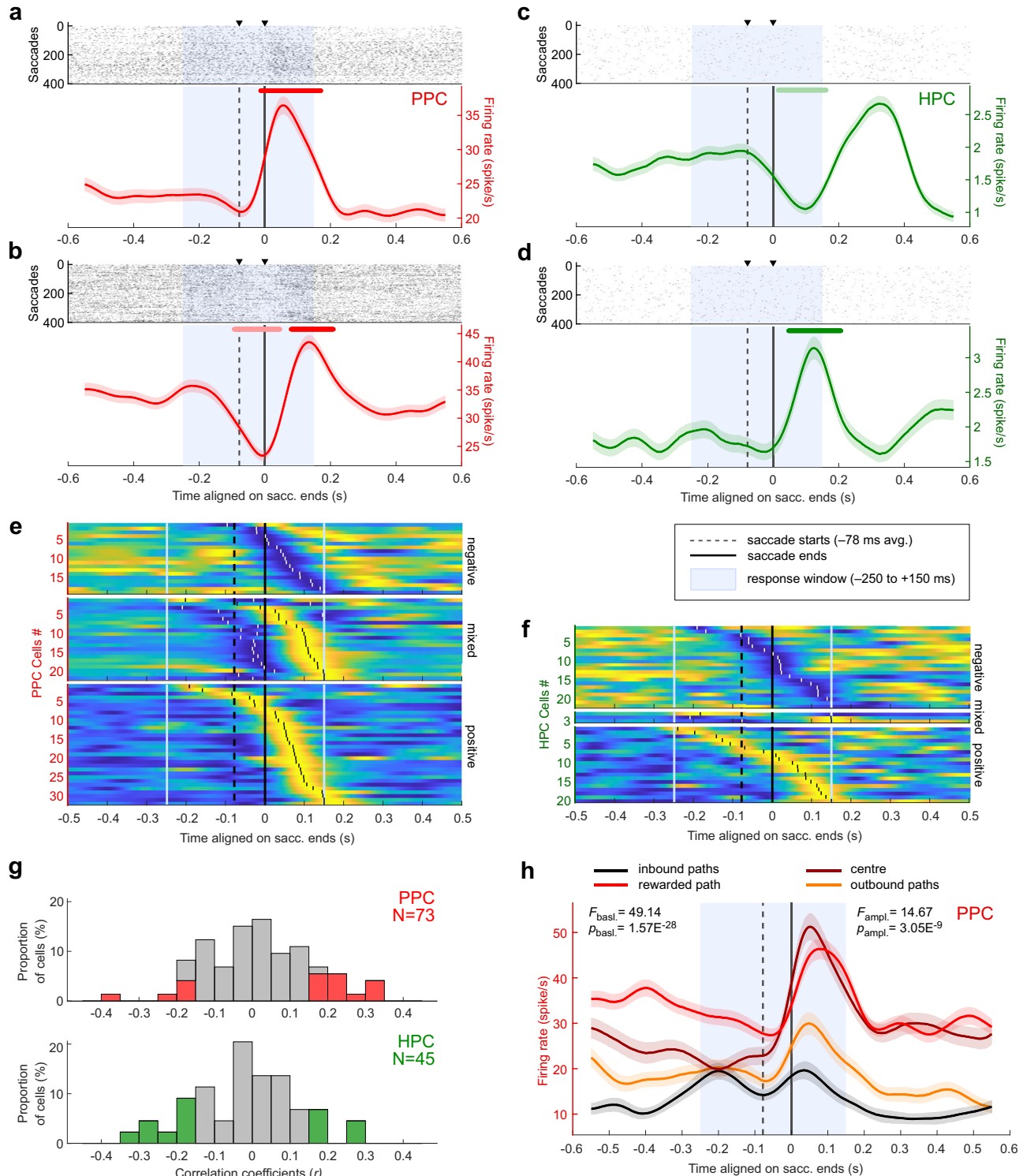

anticipation. This inhomogeneity was supported by a significant difference in saccade frequency in each monkey (see Fig. 2b and Supplementary Table 2; Monkey K: Kruskal–Wallis: $X^2_{3df} = 65.14$, $p = 4.69 \times 10^{-14}$, Monkey S: $X^2_{3df} = 48.17$, $p = 1.96 \times 10^{-10}$) and duration (see Fig. 2c; Monkey K: $X^2_{3df} = 36.76$, $p = 5.16 \times 10^{-8}$, Monkey S: $X^2_{3df} = 37.29$, $p = 4.00 \times 10^{-8}$) between the four portions of the maze. Therefore, a factor potentially accounting for position neural selectivity lies in (i) the link between the saccade-driven neural activity, (ii) the explorative function of the saccades, and (iii) the maze positions in which those saccades were performed due to the presence of key visual information.

The PPC has a known role in saccade planning and fixations[30,38–43]. We determined whether cells responded to saccades by comparing their activity for saccades directed to the left or to the right (see "Methods"). Discounting 10 cells for which the eye traces were not reliably recorded, 73/91 (80.2%) PPC cells displayed significant peri-saccadic activity, as shown in example Fig. 3a, b. Surprisingly, hippo-campal cells also showed peri-saccadic activities (45/128, 35.2%; Fig. 3c, d), yet in a lower proportion than in the PPC (homogeneity Pearson's Chi-squared with Yates' correction: $X^2_{1df} = 41.67$, $p = 1.08 \times 10^{-10}$). The averaged saccade frequency map did not differ between sessions in which responsive cells and the non-responsive

**Fig. 3 | Saccade-related activities in posterior parietal and hippocampal neurons. a, b** Raster plot (top) and average activity (bottom, in red) of PPC neurons showing a peri-saccadic response when aligned on saccade ends. The standard errors of the mean are indicated in light color. The lines above indicate times for which the response activity was significantly higher (dark; (**a**) from −14 to 171 ms; (**b**) from 79 to 209 ms) or lower (light; (**b**) from −95 to 46 ms) than baseline. **c, d** Same conventions as (**a, b**) for hippocampus example cells. The activity was significantly higher than baseline from 46 to 207 ms for (**c**), and lower than baseline from 14 to 162 ms for (**d**). **e** Population maps of the normalized activity of the PPC cells with negative (top; $N = 19$), mixed- (middle; $N = 22$), and positive (bottom; $N = 32$) response to saccades (total $N = 73$), sorted by temporal order of their peak (white or black ticks for negative or positive peaks, respectively) in the response window (light blue solid lines). For the pool of mixed-response cells, the positive peaks were used for alignment. Source data are provided as a Source Data file. **f** Same

conventions as (**e**), for the hippocampal cells showing negative ($N = 22$), mixed ($N = 3$), and positive ($N = 20$) responses to saccades (total $N = 45$). **g** Distribution of the Pearson correlation coefficients ($r$) of the correlation between the saccade frequency maps and the neural place map of each saccade-responsive PPC ($N = 73$, top) and HPC ($N = 45$, bottom) neuron. The colored bars indicate the distribution of the $r$ of significant correlations (determined by one-sided permutation tests). Source data are provided as a Source Data file. The neural map of 17/73 (23.3%) of the PPC cells (in red), and 13/45 (28.9%) of the HPC ones (in green) were significantly correlated with the saccades map. **h** Averaged activity of an example PPC cell (same conventions as in (**a**)), aligned on saccade ends. The peri-saccadic activity is separated by segments in which the saccades are made. The cell displayed a significant difference in response peak amplitude (one-way-ANOVA: $F_{3df} = 14.67$, $p = 3.05 \times 10^{-9}$) as well as in peri-saccadic response rate (one-way-ANOVA: $F_{3df} = 49.14$, $p = 1.57 \times 10^{-28}$).

cells were identified (Pearson $r = 0.80$, one-sided permutation test: $p = 9.99 \times 10^{-4}$), indicating that the difference in behavior was not driving the differences in responsive and non-responsive cells' activity. On the neurons' preferred direction, the peri-saccadic rate modulation could be positive (32/73, 43.8% of PPC responsive cells, 20/45, 44.4% of HPC ones; as shown in Fig. 3a, c), negative (19/73, 26.0% in PPC, 22/45, 48.9% in HPC; Fig. 3d), or mixed (22/73, 30.1% in PPC, 3/45, 6.7% in HPC; Fig. 3b). The population maps showed the same peri-saccadic temporal dynamics in PPC and HPC (see Fig. 3e, f, with positive and negative peak times, respectively indicated by black and white vertical ticks; Kolmogorov–Smirnov tests on the respective distributions of the positively- and negatively-modulated cells' peaks: $D_{pos} = 0.23$, $p_{pos} = 0.47$; $D_{neg} = 0.28$, $p_{neg} = 0.36$), with groups of pre- ($N_{PPC} = 5/73$, 6.8%; $N_{HPC} = 9/45$, 20.0%), trans- ($N_{PPC} = 8/73$, 11.0%; $N_{HPC} = 6/45$, 13.3%), and, for the majority, post- ($N_{PPC} = 60/73$, 82.2%; $N_{HPC} = 30/45$, 66.7%) saccade-active neurons (Supplementary Fig. 9a). In PPC, we found no link between the peak response time and the anatomical dorso-ventral coordinates of the neurons within the IPS (see "Methods"; Supplementary Fig. 9b, c; two-sided Pearson correlation: $r_{monkey\ K} = 0.015$, $p_{monkey\ K} = 0.92$; $r_{monkey\ S} = -0.082$, $p_{monkey\ S} = 0.70$), and were thus unable to clearly separate LIP from the VIP neurons.

Next, to test whether virtual position neural maps were linked to saccade-frequency maps, we computed the correlations between these, calculated for individual cells. Supplementary Fig. 8a–f, g–l show the saccades maps of the cells presented in Fig. 1b–g, h–m, respectively. We found that, out of the saccade-responsive cells, 17/73 (23.3%) of PPC cells and 13/45 (28.9%) of HPC ones showed a significant positive or negative correlation between their neural and saccades maps (Fig. 3g; one-sided permutation test, $p \le 0.05$; see "Methods"). Overall, the absolute values of correlation coefficients were not different between the PPC ($\mu|r| = 0.11 \pm 0.020$) and the HPC ($\mu|r| = 0.11 \pm 0.0026$; Wilcoxon rank sum: $|Z| = 0.31$, $p = 0.76$), showing that hippocampal saccade-responsive cells tended to covariate with saccadic activity as much as parietal ones. In sum, the behavioral saccade rate may explain the position-related neural activity, but only for a small fraction of cells, from both areas.

An alternate possibility is that the general context in which saccades and fixations were made actually impacted neuronal activity. To test this, for each cell, we compared the peri-saccadic activity (see "Methods") across the four segments of the maze, i.e. the inbound paths, the center, the rewarded path and the outbounds. We found significant differences in activity depending on context. First, for a minority of cells, there was a modulation of the amplitude of the saccadic response per se, in PPC only (18/73, 24.7% of responsive cells; one-way-ANOVA: 1df, $p \le 0.05$; example units in Fig. 3h and Supplementary Fig. 10a), since only a single unit of HPC met the selectivity criteria (1/45, 2.2%), which is lower than the chance level (5%). This effect could stem from the difference in saccade durations observed between the four segments of the maze, described above (Fig. 2c).

However, we found that only 4 out of 18 neurons exhibited the highest firing rate in the segments in which saccade durations were also the highest, which is around the chance level (4.5). Together, those results imply that the kinematics of eye movements would not solely drive the activities of these cells.

Second, for the majority of the cells, there was a modulation of the response baseline level depending on the animal's current position in the maze, without necessarily a modulation of saccade-evoked response amplitude per se. Indeed, 53/73 (72.6%) of PPC and 26/45 (57.8%) of HPC saccade-responsive cells displayed a significant response rate difference across segments of the maze (Fig. 3h, Supplementary Fig. 10a, b; one-way-ANOVA, $p \le 0.05$). At the population level, the cells preferred different parts of the maze. While, in PPC, fewer cells responded in the outbound paths compared to the other segments (5/53, 9.4%; Chi-squared goodness-of-fit on a uniform theoretical distribution: $X^2_{3df} = 7.91$, $p = 0.048$), they distributed equally between inbound (13, 24.5%), center (18, 34.0%) and rewarded paths (17, 32.1%; $X^2_{3df} = 0.88$, $p = 0.65$; Supplementary Fig. 10c). In the HPC, the four maze segments were homogeneously distributed among the cells, with 9/26 (34.6%) cells more active in inbound paths, 5 (19.2%) in center, 4 (15.4%) in rewarded path and 8 (30.8%) in outbound paths ($X^2_{3df} = 2.62$, $p = 0.45$). Thus, cells in both areas responded differently depending on the ongoing task context. In PPC, general attentional or motivational processes might have had an impact on the whole population, notably because the passive motion along the outbound paths may require less cognitive resources for the animal. However, the homogenous repartition among the rest of the maze suggests that each neuron responds to a specific sensori-motor context.

Overall, these results show that the heterogeneity in spatial activity patterns observed for single-cell neural place maps (Fig. 1b–m and Supplementary Fig. 3) could only be partially imputed to pure oculomotor dynamics, but were also modulated by task context. Among the main factors likely varying with task context, we next analyzed neurons' activity as a function of directed gaze within the visual scene.

### A spectrum of responses to visual layout gives rise to different spatial selectivities across regions

As the visual landscape making up the virtual environment was simple, we constructed a continuous representation of the fields of view (FOV), allowing plotting the gaze on this panoramic scene (Fig. 4a), comprising the 4 landmarks neighboring each of the paths (the 5th one is usually not visible in correct trials, see "Methods"). The calculation of the point of gaze took into account the camera's position and orientation, combined with the eye positions in the FOV. The point of gaze was then projected on a cylindrical wall at the landmark's position, which we linearized into a panorama centered on the goal (Fig. 4a). To assess which visual cues animals attended in the virtual environment, we quantified landmark or path fixations (see "Methods"). Animals fixated a landmark $0.18 \pm 0.0066$ times per second (i.e. a landmark was

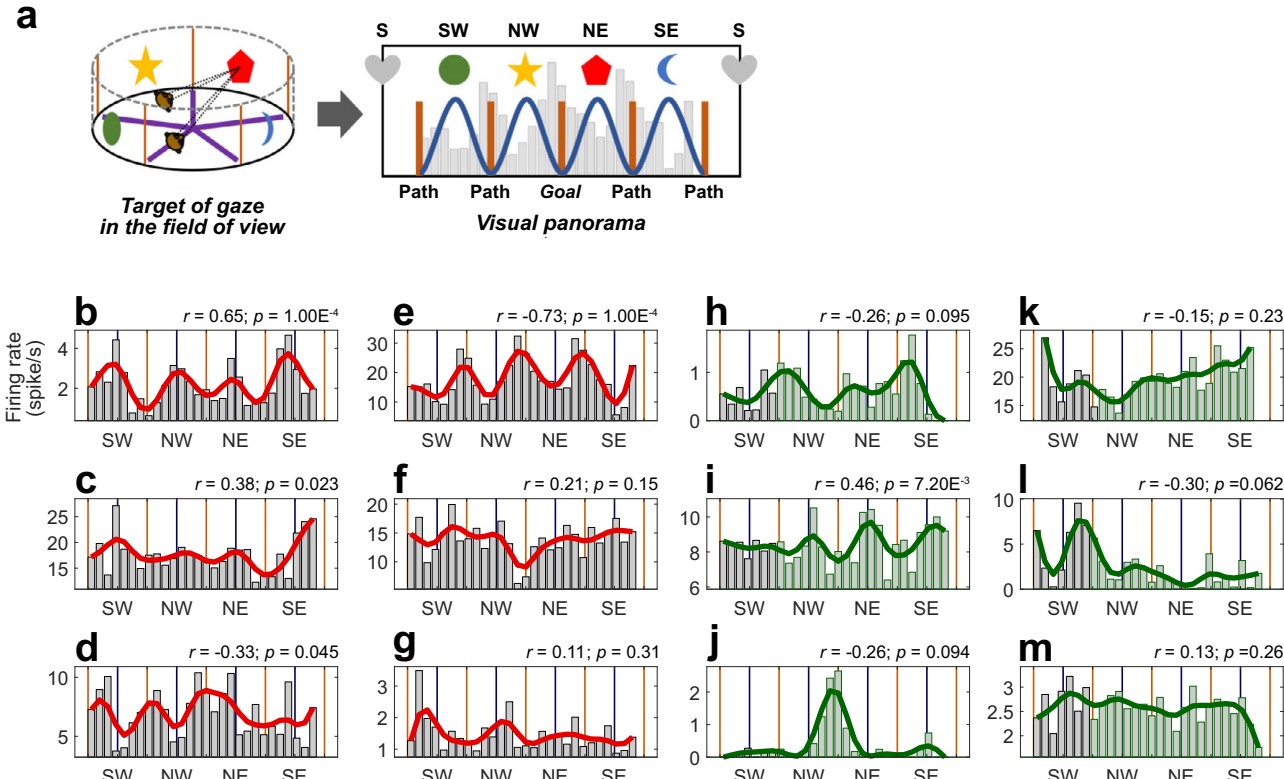

**Fig. 4 | Single-units' activity as a function of the point of gaze. a** Method used to analyze neuron's firing rates as a function of monkey's gaze position in the virtual environment: the point of gaze (left) in the visual panorama was computed by taking into account (1) monkey's virtual facing location as a function of his position and head direction, in which was factored (2) the eyes positions in the field of view (FOV). The points of gaze were represented in a visual panorama (right), with a 10° resolution. The geometric forms on the top represent the position of the landmarks in the panorama, and the brown lines the position of paths. Cells activities were then distributed as a function of visual bins (example with the fainted grey histogram). A 72-degree-period sine-wave (blue) was computed as a reference of a landmark-gazing-responsive cell. **b–g** Map of neuronal activity as a function of

monkey's gaze position into the visual panorama for 6 PPC single unit examples (same units as in Fig. 1b–g). Note that, for a clearer visualization of activity modulations, the ordinate axis scale varies across individual cells' plots, and may not necessarily begin from zero. Source data are provided as a Source Data file. The correlation coefficients ($r$) and $p$-values ($p$) of the two-sided Pearson correlation tests between the averaged activity and the reference sine-wave are indicated for each cell. **h–m** Same conventions as (**b–g**), for hippocampal cells, corresponding to the neurons in Fig. 1h–m. Figure 4a adapted from Wirth S, et al. Gaze-informed, task-situated representation of space in primate hippocampus during virtual navigation. *PLOS Biology*, 15: e2001045 (2017) under a CC BY license: https://creativecommons.org/licenses/by/4.0/.

foveated every 5.49 s). By comparison, animals fixated a path – excluding the rewarded path –0.15 ± 0.0096 times/sec (i.e. a foveation every 6.84 s), which is significantly less than landmarks (paired Wilcoxon: $|Z| = 5.25$, $p = 1.52 \times 10^{-7}$). That animals preferred to direct their gaze to landmarks is in line with the hypothesis that these are more informative than paths.

Then, we represented the neural activity as a function of gaze position in the visual panorama. Five out of the 142 hippocampal cells were removed from the pool for this section, due to unusable eye tracking data. Figure 4b–m shows the firing rates of the 12 examples of PPC and HPC cells from Fig. 1b–m, as a function of the monkeys' point of gaze on the visual panorama, binned by 10° of FOV. Neurons responded when landmarks were directly gazed at (Fig. 4b, c, i) or when they were at the periphery of the visual field, i.e. when maze paths were fixated (Fig. 4d, e, j). To quantify this modulation by gaze, we used a 72°-period sine-wave, with peaks aligned with the 4 usually visible landmarks in the panoramic scene (Fig. 4a, dark blue sine-wave), and calculated the Pearson correlation between this model and each neuron's firing rate as a function of gaze position. The obtained coefficients varied from positive (landmark preference; Fig. 5a top, Fig. 5b, c) to negative (path preference; Fig. 5a bottom, Fig. 5d, e) values. A high absolute value of the coefficient denotes a sharper tuning to the animal's fixation of the exact positions of all

landmarks or paths. On the other hand, a low absolute coefficient implies either a broad tuning for all the cues, or a selectivity for a single one.

The cells distributed between the "landmark cells" (58/111, 52.3% parietal cells and 74/137, 54.0% of hippocampal cells) and the "path cells", suggesting a continuous representation of the FOV, with response fields shifting from foveal (landmark cells) to peri-foveal (path cells) in both regions (Fig. 5a). Yet, there were significantly more PPC cells (40/111, 36.0%) with a significant correlation to the 72°-period wave (one-sided permutation test, $p \leq 0.05$; represented by filled circles in Fig. 5a) compared to HPC cells (25/137, 18.3%; homogeneity Pearson's Chi-squared with Yates' correction: $X^2_{1df} = 9.13$, $p = 0.0025$). Within the PPC, there were as many cells recruited by gazing at landmarks or paths, ($N_{land} = 21/40$, 52.5%; $N_{path} = 19/40$, 47.5%; binomial test: $p = 0.32$), but in the HPC, more cells responded significantly to the gazing of landmarks compared to paths, ($N_{land} = 18/25$, 72.0%; $N_{path} = 7/25$, 28.0%; $p = 7.32 \times 10^{-3}$). At the anatomical level within the intraparietal sulcus, we found no relationship between the depth of cells along the sulcus and the nature of the cells' activity, suggesting that selectivity to landmark and path were expressed equally in LIP and VIP (Supplementary Fig. 11a, b; two-sided Pearson correlation:

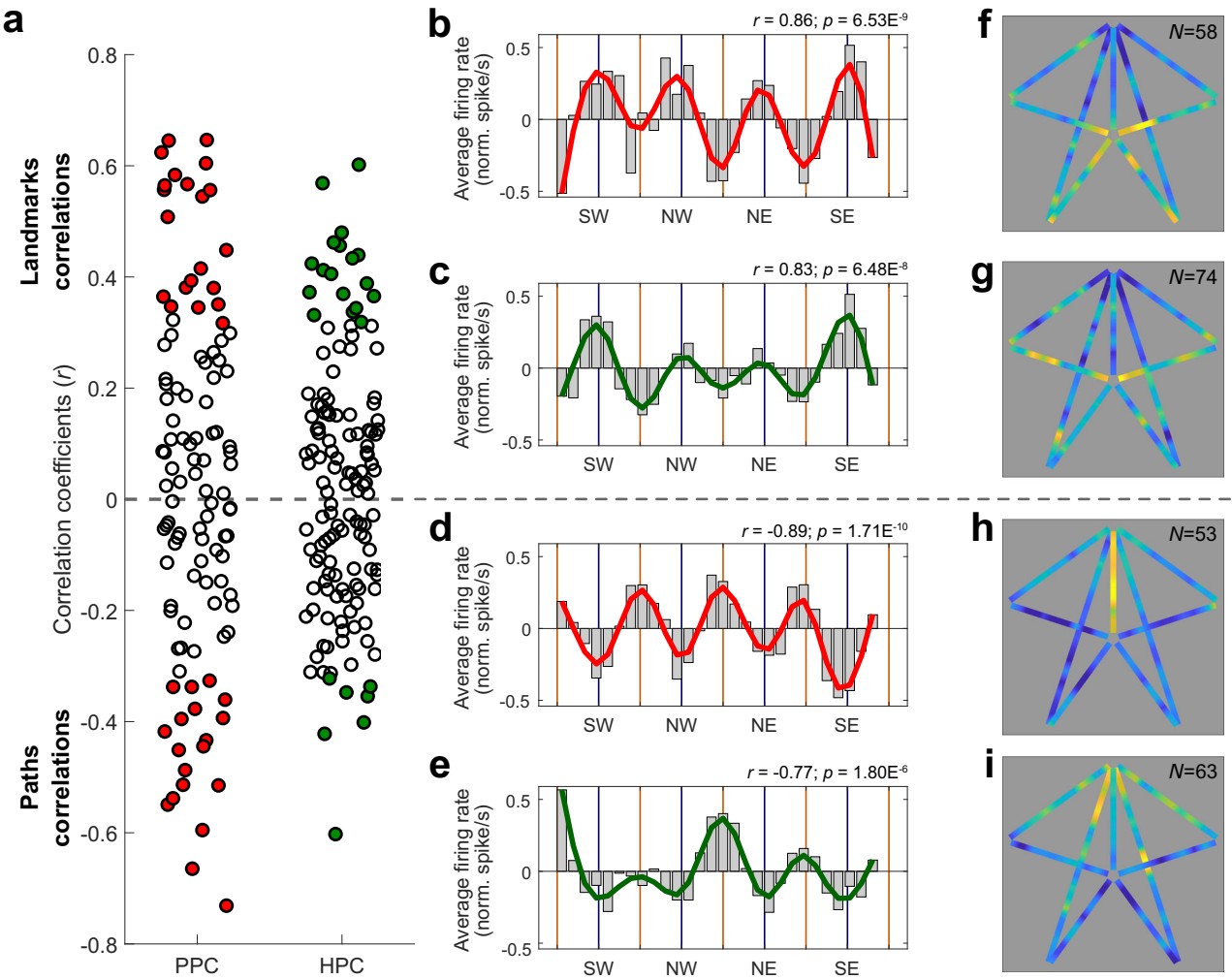

**Fig. 5 | Linking place and view in the posterior parietal cortex and the hippocampus. a** Pearson correlation coefficient of all PPC (left, $N = 111$) and HPC (right, $N = 137$) cells. Source data are provided as a Source Data file. The gaze-related activity of 40/111 (36.0%) of PPC cells (red) and of 25/137 (18.3%) of HPC ones (green) were significantly positively, or negatively correlated with the reference sine-wave (colored markers; one-sided permutation test), and overall the average absolute coefficient was higher in PPC than in HPC ($\mu|r|_{PPC} = 0.26 \pm 0.035$, $\mu|r|_{HPC} = 0.19 \pm 0.023$; Wilcoxon rank sum: $|Z| = 2.65$, $p = 0.0080$). **b, c** Averages of the normalized activity of PPC ($N = 58$, in red) and HPC ($N = 74$, in green) landmark cells, as a function of monkey's gaze position into the visual panorama. Source data are provided as a Source Data file. The activities were significantly positively correlated with the reference sine-wave (PPC: two-sided Pearson correlation: $r = 0.86$, $p = 6.53 \times 10^{-9}$; HPC: $r = 0.83$, $p = 6.48 \times 10^{-8}$). **d, e** Averages of the normalized

activity of PPC ($N = 53$, in red) and HPC ($N = 63$, in green) path cells, as a function of monkey's gaze position into the visual panorama. Source data are provided as a Source Data file. The activities were significantly negatively correlated with the reference sine-wave (PPC: two-sided Pearson correlation: $r = -0.89$, $p = 1.71 \times 10^{-10}$; HPC: $r = -0.77$, $p = 1.80 \times 10^{-6}$). **f, g** Averages of the normalized neural place maps (see Fig. 1) of PPC ($N = 58$, S) and HPC ($N = 74$, T) landmark cells. The two maps were significantly correlated (Pearson correlation: $r = 0.50$, one-sided permutation test: $p = 1.00 \times 10^{-3}$). **h, i** Averages of the normalized neural place maps of PPC ($N = 53$, U) and HPC ($N = 63$, V) path cells. The two maps were significantly correlated (Pearson correlation: $r = 0.24$, permutation test: $p = 0.0070$). In both regions, the place maps of landmark cells were negatively correlated with the path cells one (PPC: Pearson $r = -0.18$, one-sided permutation test: $p = 0.026$; HPC: $r = -0.31$, $p = 1.00 \times 10^{-3}$).

$r_{monkey\ K} = 0.070$, $p_{monkey\ K} = 0.60$; $r_{monkey\ S} = -0.16$, $p_{monkey\ S} = 0.24$), and preventing the clear identification of the neurons of these two regions.

When comparing modulation strength across regions, the absolute correlation coefficients and the modulation coefficient (M; see "Methods") of PPC cells were higher compared to HPC ones (Pearson correlation: $\mu|r|_{PPC} = 0.26 \pm 0.035$, $\mu|r|_{HPC} = 0.19 \pm 0.023$, Wilcoxon rank sum: $|Z| = 2.65$, $p = 0.0080$; Modulation coefficient, $\mu M_{PPC} = 0.13 \pm 0.012$, $\mu M_{HPC} = 0.11 \pm 0.0094$, Wilcoxon rank sum: $|Z| = 3.028$, $p = 0.0025$). When comparing firing rates, a two-way-ANOVA on the neurons' peak activity revealed a significant effect of the neural area ($F_{1df} = 15.5$, $p = 1.08 \times 10^{-4}$) and of the cell "Type" factors (i.e. landmark or path cells; $F_{1df} = 4.72$, $p = 0.031$), and that those two factors tended to interact ($F_{1df} = 2.97$, $p = 0.086$). Precisely, parietal cells ($\mu peak = 9.70 \pm 2.39$ sp/sec) fired more than HPC ones

($\mu peak = 4.73 \pm 0.97$ sp/sec) and landmark cells ($\mu peak = 8.09 \pm 2.11$ sp/sec) fired more than path ones ($\mu peak = 5.74 \pm 1.10$ sp/sec), with the effect tending to be amplified for the parietal landmark neurons (see Supplementary Table 3). In sum, PPC cells responded more strongly to retinal stimulations linked to foveation of the maze's visual panorama (landmarks or paths) than HPC cells, while within HPC, more cells were recruited by processing of landmarks than by paths, but their modulation was generally lower than PPC.

In order to examine the relationship between neurons' responses to view and the virtual position, our approach was two-fold: after having characterized the cells based on their visual selectivities and parted them into two populations depending on their landmark or path preference, as described above, we computed the average neural place map corresponding to these populations. These maps would evidence where in the maze the cells were active, according to the

visual features they preferentially responded to. This classification immediately revealed clear and distinct spatial patterns for landmark or path cells. Both parietal and hippocampal landmark cells fired as monkey were located in the inbounds, but parietal path cells appeared to prefer the rewarded path, while hippocampal ones were more active in beginning of the outbound paths, following the reward delivery. Accordingly, within regions, place maps for landmark and path cells were negatively correlated, in both PPC and HPC (Pearson $r_{PPC} = -0.18$, one-sided permutation test: $p = 0.026$; $r_{HPC} = -0.31$, $p = 1.00 \times 10^{-3}$). Across brain regions, and for both cell types (landmark or path cells), the place maps of PPC and HPC were significantly correlated ($r_{land.} = 0.50$, $p_{land.} = 1.00 \times 10^{-3}$; $r_{path} = 0.24$, $p_{path} = 0.0070$), suggesting shared patterns across regions at a large scale.

We finally assessed how the populations of landmarks and path cells differed across brain regions at a finer scale. To this end, we compared their average activities within segments: the inbound path, the center and the rewarded/outbounds paths. When aligned to the time the monkey reached the center of the maze, the landmark parietal cells displayed a significantly higher activity than the HPC ones near intersections (Supplementary Fig. 11c, see "Methods"). Thus, PPC neurons responded more to the gazing of landmarks at specific key-positions (i.e. following the trial start, and preceding and during choice point, i.e. center), while HPC ones had a more homogeneous response along the paths. Next, the parietal path cells were significantly more active in the rewarded path, compared with the HPC ones (see Supplementary Fig. 11d), while the latter showed a more homogeneous activity, increasing until the reward was reached, and then decreasing

along the outbounds. Again, the parietal cells displayed a higher firing rate in the center compared to the HPC cells. The increased activity in the center of the maze explains the modulation to all fixated maze paths seen on Fig. 5d (see also Supplementary Fig. 6), as camera rotations in the center made the paths visible within the FOV. The results thus suggest that reward expectancy strongly modulated parietal's activity, leading to a higher activity along the specific path leading to reward.

In sum, the distribution of responses depending on animal's fixation of landmarks or paths accounts for a continuous range of selectivity depending on visuo-spatial layout, but parietal cells were generally more strongly modulated by the content of the view than hippocampal ones. Importantly, in both regions, cells were recruited in different virtual positions affording distinct views of landmarks or paths. Notably, we identified substantial differences between parietal and hippocampal position-related activity patterns, suggesting a different task-based processing across regions despite similar views.

### Temporal neural dynamics relative to landmarks appearance reveal anticipatory activities in PPC and HPC

As a cognitive map of the scene may lead to anticipation of landmark appearance, we investigated the temporal dynamics of neurons' activities immediately preceding and following the appearance of the landmarks on screen, i.e. in the animal's FOV (see "Methods"). In the parietal cortex, 46.9% of the cells (52/111) responded significantly to the appearance of landmarks in the FOV (see "Methods"; see examples in Fig. 6a, b). In the hippocampus, the proportion was similar, with 55/

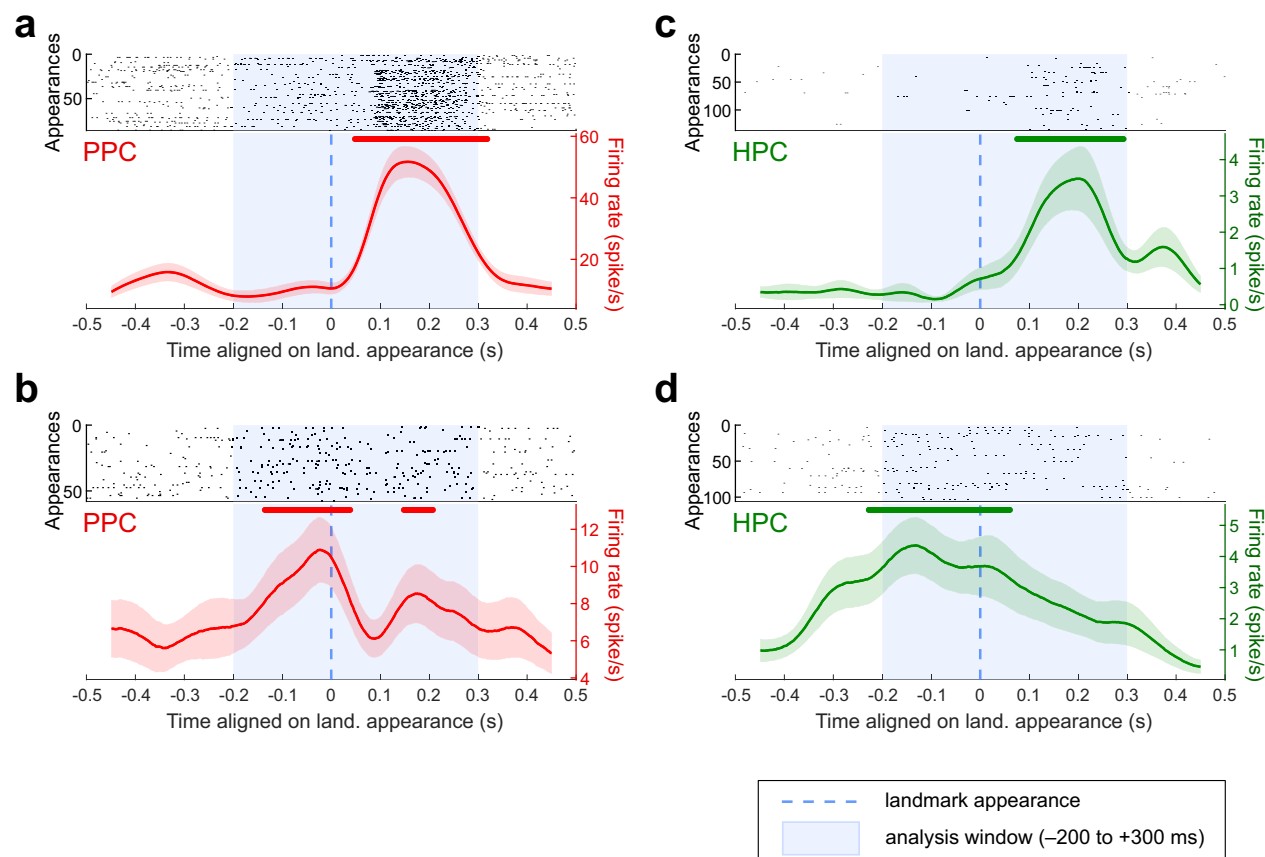

**Fig. 6 | Example of single units' activity as a function of landmark appearance.**
**a**, **b** Raster plot (top) and average activity (bottom, in red) of two example PPC cells showing a significant response when aligned on landmark appearances, for any positions in the maze. The standard errors are indicated in light color. The lines above indicate times for which activity was significantly higher than baseline ((**a**)

from 47 to 319 ms; (**b**) from −137 to 39 ms and from 147 to 208 ms). **c**, **d** Same as (**a**, **b**) for two hippocampus example cells. The activity was significantly higher than baseline from 74 to 293 ms relative to appearance for (**c**) and from −229 to 61 ms for (**d**).

137 cells (40.2%; homogeneity Pearson's Chi-squared with Yates' correction: $X^2_{1df} = 0.87$, $p = 0.35$; Fig. 6c, d). The task unfolded in such a way that most of the time, a landmark appeared on the screen concomitantly to the disappearance of the adjacent one. Thus, to check that the activities of the neurons were not actually a response to these disappearances, we isolated the parts of the maze in which solely one landmark appeared in the FOV, i.e. the beginnings of the outbound paths. Even in these conditions, we still found 48/111 (43.2%) of PPC and 42/137 (30.7%) of HPC cells that significantly responded to landmark appearances.

Then, we determined the selectivity of individual cells depending on side of appearance of the landmarks (left or right), or their identity (north-east, south-east, south-west or north-east). Importantly, the identity factor here comprises both the visual features of the object, and their position in the maze, since for each session, one landmark will still remain at the same location in the spatial layout. Two-way-ANOVAs (see "Methods"; $p \leq 0.05$) revealed that 26/52 (50.0%) of the parietal responsive cells were selective to the side, and 23/52 (44.2%) to the identity, with 18 (34.6%) of them being selective to both. In hippocampus, only 8/55 (14.6%) of the responsive cells were selective for one side over the other, and 9/55 (16.4%) for the landmark identity, 3 of them (5.5%) being selective to both factors. There were as many cells that preferred the left side compared to the right side in PPC (16/26, 61.5%; homogeneity Pearson's Chi-squared with Yates' correction: $X^2_{1df} = 0.31$, $p = 0.58$) and in the hippocampus (4/8, 50.0%). However, at the population level, a two-way-ANOVA with repeated measures on the

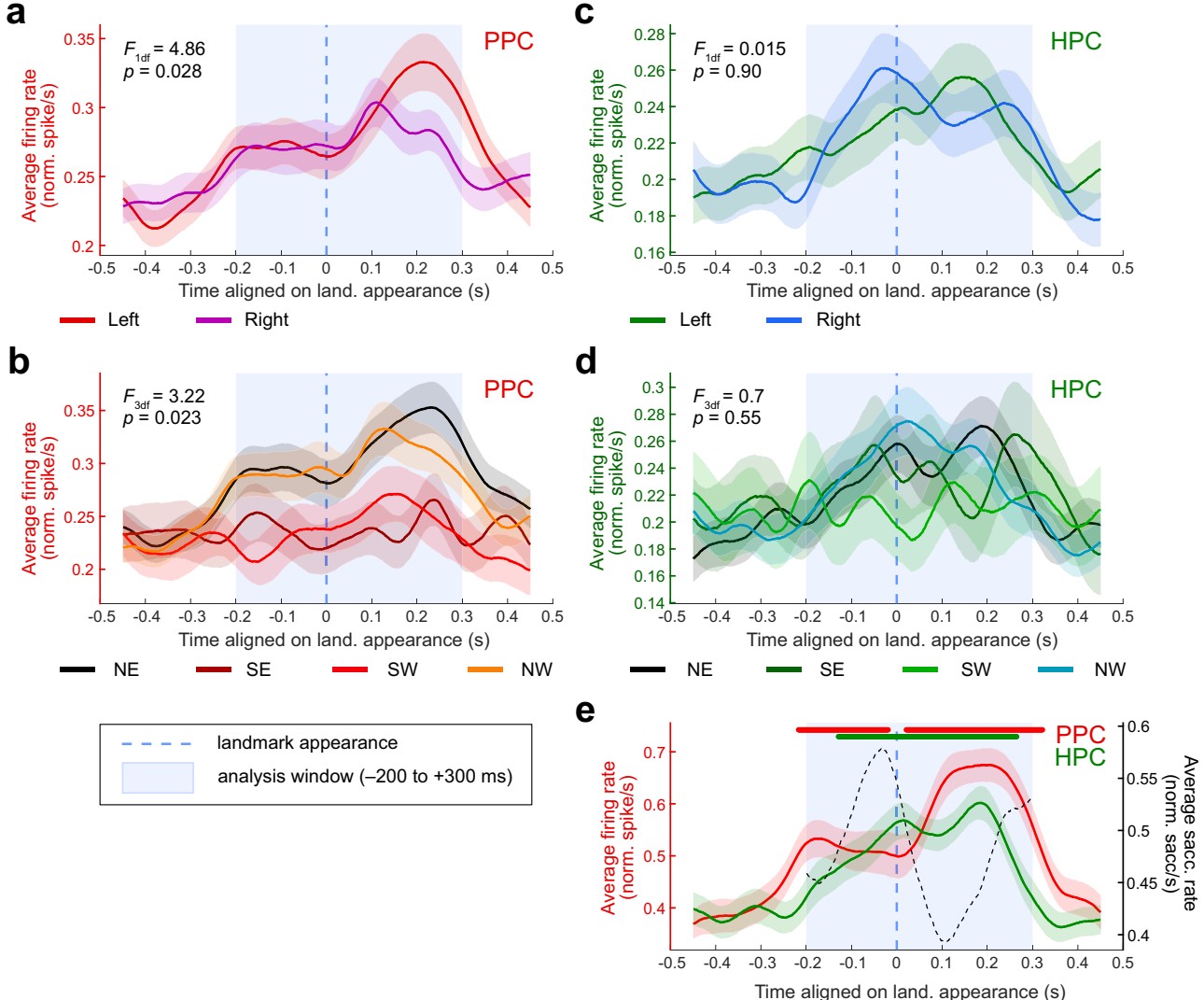

**Fig. 7 | Population activity as a function of landmark features and appearance.**
**a** Averaged normalized activity of the PPC cells responsive to landmark appearance ($N = 52$), aligned on the landmark appearance times. The peri-appearance activity is separated by the side of appearance of the landmarks in the field of view. The standard errors are indicated in light color. The population displayed a significant difference in response rate between the two conditions (3-way-ANOVA: $F_{1df} = 4.86$, $p = 0.028$). **b** Same conventions as in (**a**), for PPC cells responsive to landmark appearance ($N = 52$), with peri-appearance activity separated by the identity of the appearing landmark. The population displayed a significant difference in response rate between the four conditions (3-way-ANOVA: $F_{3df} = 3.22$, $p = 0.023$). **c** Same conventions as in (**a**), for HPC cells responsive to landmark appearance ($N = 55$), with peri-appearance activity separated by the side of appearance of the landmarks

in the field of view. There was no significant difference in the population's response rate when comparing these two conditions (3-way-ANOVA: $F_{1df} = 0.015$; $p = 0.90$). **d** Same conventions as in (**a**), for HPC cells responsive to landmark appearance ($N = 55$), with peri-appearance activity separated by the identity of the appearing landmark. There was no significant difference in the population's response rate when comparing these four conditions (3-way-ANOVA: $F_{3df} = 0.70$, $p = 0.55$). **e** Averaged normalized activity of PPC ($N = 52$, in red, left Y axis) and HPC ($N = 55$, in green, left Y axis) responsive cells aligned on the landmark appearance times. The dotted black line represents the rate of saccade starts (right Y axis). The standard errors are indicated in light color. The lines above indicate times for which activity was significantly higher than baseline (PPC: from −218 to −19 ms and from 20 to 322 ms; HPC: from −130 to 265 ms).

cells' averaged normalized firing rate (fixed factors: side of appearance of the landmarks and identity, see "Methods") revealed that parietal cells were more active when landmarks appeared on the left of the screen ($F_{1df}$ = 4.86, $p$ = 0.028; Fig. 7a). This suggests that the parietal cells that were selective to the left appearance side, were more active among the population than those selective to the right. As the recording were all performed in animal's right hemispheres, the results are consistent with a preference for the contralateral FOV in the parietal cortex. Additionally, the cells were more active when the north-west or the north-east landmarks appeared on screen ($F_{3df}$ = 3.22, $p$ = 0.023; Fig. 7b), which correspond to the landmarks that surrounded the reward location. In the hippocampus, there was no preference, neither for the side ($F_{1df}$ = 0.015; $p$ = 0.90, Fig. 7c), nor the landmark ID ($F_{3df}$ = 0.70, $p$ = 0.55; Fig. 7d).

Finally, we examined the temporal dynamics of the population activity relative to the landmark appearance across regions (Fig. 7e). The averaged activity of the PPC responsive cells was higher than baseline (see "Methods"), from −218 to −19 ms and from 20 to 322 ms relative to landmarks appearance, although the activity peak was post-appearance. Similarly, the hippocampal responsive cells increased their firing from −130 ms pre-appearance, and decreased around 265 ms post-appearance. Hence, the temporal dynamics of both the parietal and the hippocampal cells show an anticipatory activity of the entrance of the visual cues in the field of view, with the parietal cortex slightly preceding hippocampus. It is noteworthy that the rise in PPC activity also anticipates the saccade rates (black dashed line), suggesting a memory guidance of directed saccades.

Overall, our results show that parietal and hippocampal neurons are sensitive to the appearance of meaningful visual cues in the field of view, and encode spatial features, such as their side of appearance or their position (or identity) in the environment. At the population level, anticipation of landmarks appearance, together with a higher activity for landmarks surrounding the reward location in parietal cortex, suggest a representation of the spatial layout in a cognitive map, with objects' saliency established according to the internal agenda.

## Discussion

To understand the neural basis of vision-based navigation, we compared neural activity in intraparietal areas to that of the hippocampus, while two monkeys searched a virtual star-maze for a hidden reward. More parietal neurons were modulated by animals' position in the virtual maze than hippocampal ones, and they expressed more stereotyped patterns than in the hippocampus (Fig. 1). Active visual exploration during navigation was accompanied by saccade-related neural activity; this was expected in the parietal cortex, but was also surprisingly observed to a lesser extent in the hippocampus. Nevertheless, oculomotor activity per se did not solely account for spatial selectivity in neither area (Figs. 2, 3). Parietal activity especially, was strongly linked to the fixation of visual elements of the environment such as landmarks or paths, thereby resulting in specific virtual-position selectivity (Figs. 4, 5), connecting view and task-specific contexts. The analysis of the temporal neural dynamics of cells relatively to landmark appearances in the field of view suggested that both regions anticipated them, but only parietal neurons were modulated by landmark's proximity to the reward (Figs. 6, 7). These results show how parietal cortex and hippocampus support a continuous representation of visual space, through dynamic and successive recruitment of neurons, during action preparation and acquisition of elements in visual space. The results shed light on how seemingly place codes bind views and actions in space with an internal task-based map.

### Interpreting position selectivity

Virtual navigation resulted in a greater recruitment of parietal neurons than hippocampal ones, and further, their selectivity was as strong as in hippocampus – yet qualitatively distinct. Thus, virtual-reality-generated

visual stimulation resulted in parietal spatial selectivity that could erroneously be interpreted as "place" codes akin to those found in rodent hippocampal place cells. An important difference was that parietal neurons displayed repeated patterns of position selectivity, for similar sub-segments of the maze. This echoes results showing that rodent posterior parietal cells demonstrates strong pattern recurrence, for similar action-types or postures, at different locations of a real or virtual environment[6,46,47,65–67]. This suggests that parietal cells are recruited for specific types of sensorimotor patterns, such as oculo-motor movements or self-motion cues, such as optic-flow stimulations, linked to specific navigational context. The interpretation of our data may follow a similar line. For instance, the PPC neurons' enhanced activity before the maze center or at the beginning of the inbound paths (see Figs. 1b, c, d, 5f) may be related to anticipatory or concurrent responses to the onset of rotational or linear movements, as previously observed in rodent PPC[68]. Further, in the primates, including humans, parietal cortex supports visuo-spatial attention and eye-movements directed in space[21,23,24,29,32,33,38,41–43,69], which are functional properties likely recruited during virtual navigation. This could account for the very strong parietal activity observed before the decision point, or in the rewarded path, before the reward delivery, seemingly tiling the space in a function-driven manner. We suggest that the apparent parietal position code can result from visual processing of relevant cues, contributing to a task-relevant representation of visual space, rather than pure *place* code for self-position in Euclidian coordinates. Further, we also extend the latter interpretation for the position selectivity found in hippocampal cells in this task. Indeed, unlike rodents place cells[61] (though recent works have evidenced orientation-modulated responses in hippocampal cells of rodents[1,17,62,63]), cells we recorded displayed small and unstable spatial fields and orientation selectivity in the center of the maze, compatible with an active task-dependent visual processing of the scene, as gaze behavior is highly dynamic and consists of brief saccades and fixations toward visual cues.

Indeed, in both regions, the nature of cues processed during fixations impacted the cells strongly. Neurons displayed a range of responses to landmarks and paths at different retinal positions, associated to different virtual positions (Figs. 4, 5 and Supplementary Fig. 11c, d). Given the environment layout, the nature of the selectivity of the path cells could arise from a peripheral retinal stimulation by the landmarks or from a central stimulation by the path, the latter being consistent with a role of parietal cortex in encoding route affordance[44,70]. Cells responding to landmarks views displayed a higher activity than path cells. This suggests that the direct foveation of cues that are relevant to navigation due to their stable and orientation-informative nature elicits more activities than routes, that geometrically structure the environment in an ambiguous (because symmetrical) fashion. Compared to hippocampal cells, parietal activity was more strongly controlled by visual cues, as expressed by a stronger correlation between rate as a function of point of gaze and the visual scene layout (Fig. 5a), and a higher modulation coefficient. Our results show that the previously identified link between hippocampal activity and view[10,19,71–73] is weaker than in the parietal cortex. Thus, although hippocampus receives input from the parietal cortex[24,74,75], its activity is not merely inherited from the parietal cortex. Rather, it does not simply reflect the presence of a spatially or behaviorally informative object in the visual receptive field, but cells may be sensitive to a combination of visual elements. The hippocampal cells bear similitudes with previously described spatial view cells[76–79], as they can be selective to different views or landmarks. However, we show that they also integrate task-sensitive variables[10,80]. Our interpretation is that unlike spatial view cells, which are position invariant, cells recorded in our task encode the landmark presence from specific position within segments (inbound paths or center or out-bounds). This is consistent with our previous conclusions[10] and recent findings obtained in freely moving marmosets[80] or macaques[20] which show mixed selectivity between view and place.

An interesting picture arose in the two regions, when we computed the average neural place maps for landmark or path cells, revealing how population position-related firing patterns are linked to visual cues and their task-relevant processing (Fig. 5f–i). In line with a role of parietal cortex in encoding animal's motivational state[41,69,81], parietal "path" cells displayed a higher activity in the path leading to reward. By contrast, hippocampal path cells were the most active following the reward delivery, in line with previous reports showing that hippocampus encodes reward outcome in primates[82–87] and reward locations in rodents[88–91]. For landmark cells, the highest modulation occurred in parietal cortex when animals were close to decision points in inbound paths, while hippocampal activity was more homogeneous in those segments (Fig. 5f, g and Supplementary Fig. 11c), suggesting that each region takes part in different processes. Population activity allows to identify key-spatial-points in the parietal cortex, while hippocampal activity has less of a population average signature within segments. The spatial selectivity of parietal cells suggests that it may take part in decision and action-guidance through its role in attention-driven saliency maps[92,93]. On the other hand, the HPC cells may play a role with landmark identification within a specific layout, as different individual cells are recruited for different views. The fact that the HPC population were mostly active during the returns along the outbounds additionally suggests that such landmark selectivity would play a specific role following outcome delivery, taking part in learning processes. Overall, and despite their superficially similar responses to actions such saccades or cues fixations, the general sensori-motor context results eventually in a distinct, complementary task-based representation of the space between the two areas.

## Contextually-modulated saccade-related activity

Saccades, which are characteristic of visual exploration, bring elements of the visual scene into focus. Consequently, they can serve as a proxy for visual attention[94–99] and play a part in the subsequent formation or consolidation of memories[100–104]. Accordingly, monkeys made many saccades when landmarks appeared in the visual field, or before decision points (Fig. 2), and these were followed by fixations of landmarks and paths. This active exploration likely reflects information-seeking that guides navigation[99,105]. Concomitantly to this explorative behavior, we observed a strong recruitment of the intraparietal neurons during saccades. The modest observed pre-saccadic increase was characteristic of saccade-preparation and target-selection activities, described in LIP cells[32,42,69,106]. Further, following a previously described trans-saccadic suppression[107–110], many cells displayed the post-saccadic activity enhancement, generally observed in VIP or LIP when a stimulus enters a foveal or peri-foveal receptive field, concomitantly to saccade end[29,43,111,112]. The proportion of post-saccadic activity was higher than what is typically reported in the literature for LIP. Nevertheless, this is in line with recent work showing that saccade-related activity in LIP was diminished and displayed altered time dynamics in free-behavior conditions[113]. Therefore, the peri-saccadic activity profile does not suffice as a criteria to discriminate LIP from VIP as VIP also displays post-saccadic enhancement[107] and response to stimuli appearing in their receptive field[23,29,35]. Hence, we could not clearly separate LIP from the VIP along the depth of the sulcus using cell's activity profiles (Supplementary Fig. 9b, c). This indicates that the rich visual stimulation of the task recruited cells in both regions, with properties either functionally overlapping or expressed for co-occurring features. In line with this, the task context, embodied in virtual position, influenced both baseline activity and the amplitude of the saccade response itself in the parietal cortex (Fig. 3h and Supplementary Fig. 10a). Moreover, at the population level, there was generally no preference for one part of the maze compared with others, suggesting that decisional, attentional or motivational context weighted on the strength of the baseline signal in which saccades to landmarks were made in a complex way, differing across the cells.

Surprisingly, we found that a small population of hippocampal cells also displayed perisaccadic activity, with a similar distribution to that of parietal perisaccadic responses (Fig. 3f), including pre-, trans-, and mostly post-saccadic preferences. These results are consistent with the previous findings linking hippocampal activity not only to view but also to eye movements[20,71,114–117] and confirm the importance of active visual exploration for hippocampal processing[19,71,72,118]. The results document how a naturalistic task strongly engages neurons within the intraparietal areas and hippocampus, and how their activities reflect the impact of task-context on oculo-motor control, during goal-directed navigation.

## Processing and anticipating landmarks across regions

Our results showed that, in accordance with the existence of a self-centered reference frame previously described in literature, parietal neurons responded to the entrance of a moving object in the field of view[28,29,119]. While individual cells preferred the left or right side of appearance, the whole population did show a preference for the contralateral hemifield, as often presented in the literature[29,120,121]. While such contra-lateral selectivity is expected, parietal cells preferred landmark closer to reward, in line with the notion that posterior parietal cortex processes the motivational relevance of cues and contributes to representing the visual environment through a saliency map[32,33,81,122]. As we previously showed[10], this preference was not carried over to hippocampal cells, which appeared to represent landmarks more evenly as a population, despite selectivity being observed at the individual level. This observation is congruent with an encoding of landmarks in the HPC as topological cues that structure the environmental layout, and separately provide orientation-relative information. We may hypothesize that, while PPC would rather encode the *position* of the relevant visual cues neighboring reward, the hippocampus would more likely be selective to the *visual identity* of the objects, defined by their qualitative features. Our results are consistent with the hypothesis that parietal cells encode the saliency or relevance of landmarks with respect to the current task goal, while the hippocampus maintains a global internal map of space.

Interestingly, we also found that both parietal and hippocampal neurons anticipated the appearance of a landmark in the FOV. While we previously identified this anticipation in the hippocampus[10], the current results illustrate how parietal activity displays a similar pattern (Fig. 7), likely consistent with exploratory saccades targeting upcoming landmarks, and with the role of PPC in such target selection[37,39–42]. It has been previously proposed that hippocampus receives an indirect efference copy of the eye movement from the superior colliculus[115,117,123,124]. The temporal dynamics of the peri-saccadic response are compatible with this hypothesis, with the presence of some hippocampal pre-saccadic neurons (Fig. 3). The hippocampal temporal dynamics are also in line with the notion that the hippocampus serves as a predictive map[118,125], likely anticipating the upcoming landmark. The nature of the parietal temporal dynamics relative to landmark appearance supports the existence of such a predictive coding in parietal cortex as well, allowing controlling exploratory behavior towards anticipated targets continuously linking timely action and objects in space. The phenomenon of *predictive remapping*, through which LIP neurons begin to respond to stimuli presented in their future receptive field prior to saccade execution[38,40,43,126], implies that the neurons have access at any time to visual information covering the whole FOV, giving an impression of "anticipation". In addition, the tonic activity of LIP neurons that maintains the location of a recently extinguished stimulus in working memory[38,39,111,127] suggests that they can also represent non-visually-present cues within the FOV. Our results demonstrate that, further, PPC neurons may hold a representation of the environment that extend beyond the FOV. This may be achieved through top-down

connections from HPC to PPC[50,51,74,75], that would provide the latter with information extracted from the cognitive map held in memory.

Altogether, the complementary spatial activity maps of the two regions suggests that, despite the superficial similarity of their responses to actions such as saccades or to sensory events such as landmark appearances, their selectivity to specific contexts results in a task-based discrimination of animal's position in its environment and its orienting towards the goal. Ultimately, those spatial responses allows for the linking of action and objects in space and memory. Future work should further explore the impact of the environment familiarity on such spatial encoding, and its evolution through learning stages.

## Methods

### Ethics statement
Our study involved two nonhuman primates. All experimental procedures were approved by the animal care committee (Department of Veterinary Services, Health & Protection of Animals, permit no. 69 029 0401) and the Biology Department of the University Claude Bernard Lyon 1, in conformity with the European Community standards for the care and use of laboratory animals (European Community Council Directive No. 86–609). Further, our procedures were examined by CELYNE, the local ethics board, which approved the in vivo methods used in the laboratory. We minimized animal suffering and maintained their well-being by using anesthetics and pain management during surgeries for recording chamber implantation. During the experiments, animals' behavior and well-being was monitored.

### Animal subjects
The study was conducted on two male rhesus macaques (*Macaca mulatta*), aged 5 to 6 years. Both were born in captivity (F2 generation or more), in France.

### Behavioral methods and setup (Fig. 1)
Animals were head restrained and placed in front of a large screen (152 × 114 cm), at a distance of 101 cm, with a horizontal FOV of 74°. They were further equipped with active shutter glasses (Nuvision), coupled to the computer for 3-D projection (DepthQ projector, Infocus) of a virtual world (Monkey3D, Holodia). The projection parameters were calibrated to render objects' size real by calibrating disparity using the actual interpupillary distance of the monkeys (3.1 cm for monkey K and 3.0 cm for monkey S). We confirmed the animals perceived images with the depth of stereoscopic projection by measuring vergence, as a small object moved from an apparent 50 cm in front of the screen, to 150 cm behind the screen. To this end, two small infrared cameras were mounted above each eye, and the movement of the pupils of each eye was monitored (ASL). The cameras further allowed monitoring the animal's gaze through the task. Animals learned to navigate *via* the joystick towards a reward hidden at the end of one of the star-maze paths (Fig. 1a). The star-maze had a radius of 16 m, and speed of displacement was 5 m per second. This velocity was chosen to optimize the number of rewarded trials in a session, and prevent the animals from getting too impatient.

During a shaping period that lasted 6 months, animals learned to find the reward targets whilst operating a joystick that controlled a sphere on the screen. Once they had mastered this task, they were introduced to a 3-D-version of this task. Then, they were trained in a simple Y-maze, in which they had to move the joystick to approach the sphere. Next, landmarks were introduced along the Y-maze, and animals were trained with the sphere in presence of the landmarks. Then, the sphere was removed and animals were rewarded when they went toward the end of the arm where the sphere was last. To this end, they had to use the landmarks. At this point, they were finally introduced to the full star-maze. For one animal that would not go to the end of an arm if a sphere was not there, a different strategy was adopted: we replicated the sphere five times and changed the rules such that there was a sphere at each end, but the animal had to find "the one" which would give a reward and blink when approached. Once this step was learned, the spheres were removed for him as well. Finally, animals were trained to learn new landmark arrangements every day.

We used a star-shaped environment rather than using an open field to ensure multiple passes through the same trajectories, and to avoid locations with too sparse data. Each day, the animals had to locate a new position of the reward with respect to new landmarks. Each trial began with the animal facing the maze from one inbound path end. The joystick allowed the animal to move to the center and turn left or right to choose and enter one path. Once the animal reached the end of the path, it was given a liquid reward only if correct, and then brought to a randomly chosen new start, whether the trial was correct or incorrect. Figure 1a presents the sequence of a trial from above (top panel) and from the animal's perspective (bottom panel).

### Mapping the animal's point of gaze in the allocentric reference frame (Fig. 4)
We computed the point of gaze in an allocentric frame, wherein objects (landmarks or paths) or positions in space towards which the monkey gazed were mapped. To this end, we projected the point of gaze on a cylindrical wall intersecting landmarks and paths (Fig. 4a), using orientation of the camera as a function of its position, and combined with the X and Z eye position in the field of view. The points of regard were mapped onto a vertical circular wall enclosing the landmarks; this wall was then cut and flattened into a visual panorama collapsing the vertical dimension. This panorama represents where the animal is gazing in the spatial scene, not the craniocentric eye position. The coordinates obtained were then used to compute the firing rate of the cells as a function of the animal's point of regard (Fig. 4b–m; see *Correlation between gaze activity and reference sine-wave*).

### Electrophysiological recordings
For a period of approximately 6 months, each animal underwent daily recording sessions, during which laminar U-probes (Plexon®) were lowered to the target areas along the orthogonal stereotaxic axis (see Supplementary Fig. 1 for PPC, and[10] for the HPC), both located in the animal's right hemispheres. Recordings began if individual cells were present on the contact electrodes, and the task was then started. Individual cells were pre-sorted online and re-sorted offline (Offline Sorter, Plexon Inc.), and only cells whose waveforms possessed reliable signal-to-noise ratios (two-thirds of noise), high enough activity rate (activity peak superior to 1 spikes/sec), and whose activity was stable in time for at least 10 trials per starting inbound paths (40 trials in total) were included in the database. One hundred and eleven cells recorded in the intraparietal sulcus were kept for place- and saccades-related analyses, and 142 cells recorded in CA3, CA1 or the dental gyrus of hippocampus. However, some of the eye-tracking data were not reliably recorded for a few HPC cells, and therefore, 10 of them were removed from the pool for the gaze-related activity analyses.

### Data analysis
All data were analyzed with custom Matlab scripts. The normality condition of the samples being not always respected, we preferred to use non-parametric statistical tests for most of the analyses, allowing a greater robustness. For all tests, the α risk was set to 5%. Analysis were performed only on the correct trials data.

**Wilcoxon tests.** As the normality was not respected for most of our samples, and in a concern of higher robustness, we used a Wilcoxon rank sum test when the medians of two unpaired samples had to be compared, and matched signed-rank in the case of paired samples. We used the exact method to compute *p*-values when $\min(N_{sample\ 1}, N_{sample\ 2}) < 10$ and $N_{sample\ 1} + N_{sample\ 2} < 20$ for unpaired samples, and

when $N_{sample} < 15$ for paired samples. For larger samples, we used a normal approximation.

**Saccade detection.** Saccade starts and ends were respectively defined as instants when eye velocity exceeded or went under a fixed speed threshold of 50°/sec.

**Neural place maps & saccades maps.** We computed each cell's mean firing rate for each spatial bin, by simply dividing the number of spikes recorded in that bin by the total time spent in it. Only bins comprising at least four successful trials were kept. For display (Figs. 1b–m, and Supplementary Fig. 2, 3), a smoothing procedure was applied: the instantaneous firing activity was smoothed with a Gaussian kernel ($SD = 100$ ms) before computing the map. The temporal smoothing was chosen to suit HPC cells, which have often low firing rates (<10 Hz), and for which we adopted a temporal binning of 100 ms per position bin. This spike rate series was then smoothed with a Gaussian kernel of width 10 ms for visualization purposes only. When comparing spaces, no smoothing was used and bin sizes were adjusted so that each map contained a similar number of bins ($N_{bin} = 113$). The same protocol was used to produce saccades maps, except that instead of the neuron's spikes, the animals' saccades were counted, using the time of their ends. Because a slight variability in saccade density was noticed across sessions, the correlations tests have been performed between the neural place maps and the saccades maps, both computed for each individual cell. As several cells were recorded on a single session, different cells could share the same saccades map. To study the similarity between two maps, either place or saccades ones, a one-sided permutation test was performed: surrogated data were created by randomly shifting one of the two 113-bins-long-matrix 999 times, and two-sided Pearson correlation tests were performed for each of the surrogated data, as well as for the actual one. The rank of the actual correlation coefficient among the set of 1000 (actual + 999 surrogate ones) was used to extract a statistical $p$-value (bilateral test). To generate mean maps of a population of cells, the matrices were normalized using standardization method, before being averaged.

**Information Content (IC).** For each individual cell, we iteratively adjusted the spatial resolution of their place map to get as close to 200 valid bins as possible. Bins were considered valid if they included more than 400 ms of time in successful trials. We computed the information content in bits per spike with the following formula[128]:

$$I = \sum_i \frac{\lambda_i}{\bar{\lambda}} log_2 \left( \frac{\lambda_i}{\bar{\lambda}} \right) p_i$$

where $\lambda_i$ is the firing rate in the spatial bin $i$, $\bar{\lambda}$ is the mean firing rate, and $p_i$ is the fraction of the time spent by the animal in bin $i$. IC is zero for a homogeneous firing over the $M$ bins; it is equal to $log_2(M)$ when a single bin contains all the spikes and the animal spends an equal amount of time visiting each bin. To avoid potential bias, we normalized the IC by subtracting from it the mean IC of the 999 surrogate datasets: we first created 999 surrogate data sets, in which we divided the recording time into chunks of 5 s, that we randomly shifted. This procedure decorrelated the spikes from the animal's behavior while essentially preserving the structure of spike trains (e.g. spike bursts). The analyses were run on actual and surrogate data, and for any tested variable, the rank of its actual value among the set of 1000 (actual + 999 surrogate ones) was used to extract a statistical $p$-value (bilateral test). The ICs were compared across areas *via* a one-way-ANOVA with repeated measures, with the area as fixed factor and the 3 values of IC (inbounds, rewarded path, outbounds) as repeated factor.

**Sparsity index (S).** Following standard procedures[129,130], we estimated sparsity by the ratio of L1 norm over L2 norm and defined as sparsity index:

$$S = \left( M - \frac{\left( \sum_{i=1}^{M} \lambda_i \right)^2}{\sum_{i=1}^{M} \lambda_i^2} \right) / (M - 1)$$

where $M$ is the number of spatial bins and $\lambda_i$ the firing rate in bin $i$ as above. The sparsity index S is 1 for a homogeneous firing map and 0 when a single bin contains all the spikes. The sparsity indices were compared across areas and maze segments via a two-way-ANOVA with repeated measures, with the area and the maze segment as within-subjects fixed factors, the latter being also measured within subjects.

**Depth of tuning Index (DT).** The depth of tuning index was computed as a classic selectivity index:

$$DT = \frac{(FR_{max} - FR_{min})}{(FR_{max} + FR_{min})}$$

where $FR$ is the firing rate of the neuron at the different spatial bins or for the different virtual head orientations. A depth of tuning index DT is 0 when the amplitude of $FR$ is null, i.e. when there is no modulation between conditions, and 1 when the neuron responds only in certain conditions, but is silent otherwise. The DTs were compared across areas and maze segments using the same method as for the sparsity indexes, described above.

**Characterization of spatial fields.** Spatial fields were defined as contiguous place bins with neuronal activity above the threshold of mean + 2 SD of the activity over the whole space. Distances between spatial fields were computed in "virtual meter unit" coordinates, where 1 bin equals 1.6 units. The inbound paths were divided into 10 bins (16 units), short returns into 12 bins (19.2 units), and long outbounds into 19 bins (30.4 units).

**Assessment of spatial stability.** To assess the stability of the cells' spatial modulation through time, we computed for each of them a cross-correlation between the mean neural place maps of the first and the second halves of the session. The significance of the correlation was determined by one-sided permutation test, with 999 iterations of surrogates. Then, we aimed to explain the stability of the neuron's response as a function of other electrophysiological factors, that are the number of spatial fields of the neuron, the summed size of these spatial fields, and the neuron's peak firing rate. To do so, we performed two-sided Pearson correlation tests between the stability correlation coefficients of the neuronal population, computed previously, and the three predictors.

**Neuronal activity aligned on position.** To create the position rasters, we first determined on the neural place map which maze segment contained the peak activity of the neuron. As the time spent by the animal in the center and the outbound arms could greatly vary across trials, depending on the identity of the starting positions, we only computed the raster on this segment where the cell was most active. In addition, trials for which the animal took a break in the middle of a segment were discarded. The peri-event spike histograms were computed by aligning the activity of the neuron on the time of the segment start, with a time-resolution of 1 ms. To trace the mean activity, we first averaged the spike counts for all the trials to obtain a raw mean segment-start-relative activity, and finally smoothed it using a Gaussian kernel ($SD = 40$ ms).

**Orientation-related activity.** To assess the dependence of the neurons to the animal's virtual orientation for a same position in the maze, we only used data corresponding to the center, which was the only

position for which the orientation varied. The activity of the neuron was computed as a function of the orientation, smoothed (with a Butterworth low-pass filter set to 2.5% of the sample rate), and the significance of the modulation was tested against a pool of 1000 surrogated data, with the threshold fixed as the mean ± 2.5 SD of these surrogated data.

**Symmetry of place-related activity.** For each cell significantly modulated by position (see *Information Content*), we compared their mean FR for the right part of the maze with the one of the left part with a two-sided Wilcoxon test, considering activity only in inbound paths, outbound paths or both. For the population, we performed a two-sided Wilcoxon test, comparing the left *versus* right activity of all cells, in inbound paths, outbound paths or both.

**Kruskal–Wallis for saccades frequencies and durations.** For each maze segment (inbound paths, center, rewarded path, outbound paths) of each trial, the number of saccades detected was normalized by the time spent in the segment, to obtain a saccades rate in saccades/seconds per segment, for each trial. The median saccades rates of the 4 segments were compared using a Kruskal–Wallis test, followed by a multiple comparison test, using critical values from Student's *t*-distribution, adjusted for multiple comparisons with the Bonferroni method. The same method was applied to compare saccades durations. Saccades durations were computed by subtracting the time of start from the time of end, for each saccade of each maze segment.

**Neuronal activity aligned on saccade ends (perisaccadic activity).** The ends of saccades were used instead of starts because literature showed that they were more informative and were associated to a finer tuning compared to starts[43]. Saccades were classified depending on their horizontal direction, either toward the left (negative horizontal vector), or toward the right (positive horizontal vector). The spikes of each cell were aligned on each saccade end and distributed in 1 ms-long time-bins, so we obtained two 1900ms-by-$N_{saccades}$-long matrices (one for leftward saccades, on for rightward ones), with the first time-bin corresponding to −899 ms relative to saccade ends, and the last one to +1000 ms. Then, the spike counts for all the saccades were averaged to obtain a raw mean saccade-relative activity per saccade direction, and finally smoothed using a Gaussian kernel ($SD = 40$ ms). Using a conservative criterion, a cell was considered as saccade-responsive when, considering the leftward or rightward saccades from the whole maze, the maximum or the minimum of the smoothed activity in the response window (−250 to +150 ms relative to saccade ends) was superior or equal, or inferior or equal, respectively, to the mean ± 3.5 SD of the baseline (−899 to −250 ms) for at least 30 consecutive ms. We specifically chose a higher threshold (1) to exclude spurious correlations, since saccades were very frequent events, and (2) because activity in response to saccade was more stereotyped, and thus displayed lower intra-cell variation. All of the following analyses regarding saccade-related activities were performed using each cell's response to its preferred saccade direction (i.e. the response with the higher amplitude modulation, that could be either positive, negative or mixed). When determining the response timing of a cell, we used its maximum activity peak in case of an excitatory or a mixed cell, and its minimum activity in case of suppressive cell.

**ANOVA on the saccadic responses rate and on their amplitude.** To compare the response rate of the neurons to saccades performed from different portions of the maze, for each cell, its raw activity (corresponding to its favorite direction) over the whole response window (−250 to +150 ms relative to saccade ends) was averaged for each saccade of the maze portion. This resulted in 4 matrices (1 for inbound paths, 1 for center, 1 for rewarded path, 1 for outbound paths) containing 1 value of global activity per saccade. Within each cell, the mean of those 4 matrices were compared using a one-way-ANOVA, and critical values from Student's *t*-distribution, adjusted for multiple comparisons with the Bonferroni method for post-hoc analysis. To compare the amplitude of the neurons' responses to saccades, for each cell, and for each maze segment, its minimal and maximal activity within the response window was identified from its smoothed activity matrix. The mean minimal response for each saccade was the averaged smoothed activity of a window of −25 to +25 ms around the minimal response, and the same protocol was used for the mean maximal response. Finally, for each saccade, the amplitude of the response was determined by subtracting the mean minimal response to the mean maximal response. This resulted in 4 matrices containing 1 value of amplitude per saccade. The mean of those 4 matrices were compared using a one-way-ANOVA with a Bonferroni-corrected Student's *t*-distribution-based post-hoc. To determine whether the segment in which a cell exhibited its highest activity coincided with the segment where the animal executed the longest saccades, we assessed, at the population level, whether the observed probability of both the preferred segment and the longest saccade duration occurring in the same segment exceeded chance (chance level = 4.5).

**Neurons depth in IPS.** The coordinate of each cell was computed by combining the placement of the electrode on the cranial implant with the location of the recording contact on the probe. The inclination of the intraparietal sulcus was determined using the anatomical MRI of the animals, for each frontal slice. Then, each cell's depth relative to the anatomical dorso-ventral axis was converted to coordinate relative to the sulcus inclination. The zero of the depth axis thereby created correspond to the bottom extremity of the sulcus.

**Correlation between gaze activity and reference sine-wave.** The visual space was split in 36 bins of 10° each, centered on the goal, and the point of gaze of the animal was computed (see *Mapping the animal's point of gaze in the allocentric reference frame*) for each spike of the neuron. After a temporal normalization, those spikes were distributed along the visual bins to obtain a firing rate as a function of animal's gaze position in the visual panorama. These data were smoothed using a normal probability density function, with a mean of 0, and a standard deviation of 1.2. As only the correct trials were used for the analyses, the monkey never faced the southern landmark. Thus, we removed the corresponding 8 bins out of the 36-bin-long matrix of firing rate, to be left with a 28-bin-long matrix. Further, to avoid biased firing rate calculation, we removed bins for which the cumulated time was inferior to 2 s. Each of the 4 landmarks was separated from the others by 7.2 bins (i.e. 72° of the FOV), and that was also the case for the 5 paths. A reference wave was created, as a sinus function whose period was 7.2 bins, and whose peaks were centered on the landmarks positions. To assess whether the neurons significantly responded to the gazing of landmarks or of paths, a one-sided permutation test was performed: surrogated data were created by randomly shifting the cut, smoothed gaze-related activity matrix 10,000 times, and two-sided Pearson correlation tests were performed between each of the surrogated data and the reference sine-wave, as well as for the actual one. The rank of the actual correlation coefficient among the set of 2000 (actual + 1999 surrogate ones) was used to extract a statistical *p*-value (bilateral test). If the correlation coefficient (*r*) was positive, the cell was considered as a "landmark cell", and if it was negative, the cell was considered as a "path cell".

**Modulation coefficient (*M*) and offsets with reference wave.** The modulation coefficient (*M*) was here computed as the magnitude of

the signal of the smoothed gaze-related activity at the frequency of the reference sine-wave, divided by the sum of the magnitudes at all frequencies found in the smoothed gaze-related activity, determined by Fast Fourier transform. In other words, it assesses how strongly the gaze-relative activity is oscillating at the frequency of our reference sine-wave, relatively to all of the other frequencies present in the signal, i.e. how much the neurons' activities were driven by the gazing of the landmarks.

**ANOVAs comparing the gaze populations' FR.** For each cell, we computed their maximal firing rate from their smoothed gaze-related activity. Then we tested the effect of area of recording (PPC or HPC) and cell "Type" (landmark or path cell) on the firing rate by performing a two-way-ANOVA, with interaction factor. If a *p*-value was significant, multiple comparison tests were subsequently performed, using Tukey's honestly significant difference criterion.

**Comparison of place activity in the different maze segments.** In the same way as for the histogram aligned on the start of the neuron's preferred maze portion (see *Neuronal activity aligned on position)*, the activity of each cell was aligned on two events, separately, with a time-resolution of 1 ms. First, the activity was aligned on the time the monkey reached the maze center, and computed from the first push of the joystick, at the inbound path start, to the end of the first rotation in the maze center. Second, it was aligned on the time the reward was reached, and computed from the beginning of the rewarded path, to the end of the outbound ones. Note that between reaching the reward location and the start of the returns along the outbounds, there was a stationary period, during which the monkey was given the liquid reward. To avoid excessive variation across trials, the duration of each maze segment was evaluated for all trials of all cells. Trials with durations that were too high or too low compared to the total sample (mean $\pm$ 3 SD) were discarded. The resulting averaged durations of the segments were: $1.31 \pm 0.03$ s for the inbounds, $1.16 \pm 0.04$ s for the center, $1.21 \pm 0.002$ s for the rewarded path, $1.21 \pm 0.01$ s for the reward-delivery delay, and $2.56 \pm 0.02$ s for the outbounds. Once the individual activities were obtained for each trial, they were averaged, smoothed (Gaussian kernel *SD* = 40 ms) and normalized (Max-Min method). Finally, the activities of all cells were averaged together to obtain the population mean activity. A two-sided Wilcoxon rank sum test was used to compare the activity of the PPC and the HPC, for each time-bin.

**Neuronal activity aligned on landmarks appearances.** The same protocol was used than the one used to align spikes on saccade ends, except that the beginnings of appearance times were used. As the relationship between the camera orientation and position in the maze was consistent across trials and sessions, we first determined which positions corresponded to the entrance of a landmark in the FOV, on average (the different objects could slightly vary in size). Then, the appearance times were defined as the timestamps at which the camera was at those positions. For information, the average duration took by a landmark to fully appear on screen was the following, depending on the position of the animal in the maze: on the short outbound paths, the first landmark to enter the FOV (that could be the 2 North ones) appeared in ~0.6, and the second one (the 2 South ones) did in ~0.3 s; on the long outbound paths, the landmarks (the 2 North ones) appeared in ~0.8 s. In the maze center, the duration of appearance depended on how long the monkey took to push the joystick to initiate the rotation: the landmark never fully appeared at once, since when the camera was aligned on a path, only half of two landmarks were visible on the right and the left of the screen. A cell was considered as responsive to landmarks foveations when the smoothed activity in the response window (−200 to +300 ms relative to appearance) was superior or equal to the

mean + 2.5 SD of the baseline (−800 to −200 ms and +300 to +800 ms) for at least 30 consecutive ms. After that, to create the population activity, each cells' raw mean activity was at first convoluted with a Gaussian kernel (*SD* = 40 ms), then normalized using the Max-Min normalization, before being altogether averaged. The population activity was considered as significantly responsive to landmarks appearance for each time-bin of the response window (−200 to +300 ms relative to appearance) whose value was superior to the mean + 2.5 SD of the baseline (−800 to −200ms and +300 to +800 ms).

**ANOVA on the response rate to landmark appearance.** To compare the response rate of the neurons to landmark appearance, we proceeded the same way as when we compared the saccades-response rate (see *ANOVA on the saccadic responses rate and on their amplitude*), at the population level. The average rates of the raw activity in the whole response window (−200 to +300 ms relative to saccades appearance) of every responsive cell were compared by the use of a two-way-ANOVA with repeated measures, including the side of appearance factor (i.e. left or right) and the landmark identity (i.e. north-east, south-east, south-west or north-west), both analyzed as between- and within-subjects factors.

**Reporting summary**
Further information on research design is available in the Nature Portfolio Reporting Summary linked to this article.

## Data availability
The data generated in this study have been deposited in the OSF database, without any restrictions [https://osf.io/e974a/]. Source data are provided with this paper.

## Code availability
The custom codes generating results that are deemed central to the conclusions of this study have been deposited in the OSF database, without any restrictions [https://osf.io/9jn7k/].

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

## Acknowledgements

This project was supported by LABEX-CORTEX of the University Claude Bernard Lyon 1, grant number ANR-11-LABEX-0042/ ANR-11-IDEX-0007 to S.W. and JR.D., by the Fondation pour la Recherche Médicale, grant number FDT202204015196 to M.V., and by the Agence Nationale de la Recherche, grant number ANR-17-CE37-0015 to S.W. We thank Serge Pinède for assistance with the experimental setup, Olivier Zitvogel for the code for the virtual reality task, and Aurélie Planté for help with animal behavioral training.

## Author contributions

S.W. and JR.D. conceived the study and the task. S.W. supervised the project. M.V. analysed the data and wrote custom codes, with P.B. and S.W. contributing significant additional implementations. M.V. wrote the initial draft of the manuscript. S.W., P.B. and JR.D. edited the manuscript.

## Competing interests

The authors declare no competing interests.

## Additional information

**Supplementary information** The online version contains Supplementary Material available at https://doi.org/10.1038/s41467-024-54736-7.

