## [Transparent Peer Review file · Nature Communications]

Organizing space through saccades and fixations between primate posterior parietal cortex and hippocampus

Corresponding Author: Dr Sylvia Wirth

Version 0:

Reviewer comments:

Reviewer #1

(Remarks to the Author)

Verice et al. report on recordings from the primate posterior parietal cortex (sub-regions VIP and LIP) and the hippocampus (sub-regions unspecified) during performance of a star maze virtual reality task (restrained with joystick control of movement). At issue is the responses of neurons to either maze locations or to landmarks and the comparison of posterior parietal cortex (PPC) and hippocampus (HPC). The authors report that more neurons in PPC exhibited spatial selectivity and that such selectivity was driven by visual responses to maze pathways or to landmarks.

While the responses of some neurons are indeed interesting and spatially structured, in some way, with relation to the animal's experience and behavior in the environment, the data is based on a small population of neurons. The analysis and characterization of the spatial responses do not adequately explain the responses across different maze locations and the contrasts between PPC and HPC are of minor to moderate strength. As a result, the manuscript fails to build a novel conceptualization of the role of PPC versus HPC in task performance and the visual responses are less rigorously characterized than one would normally expect (if the experiment and results were analyzed primarily from an object and visual field location perspective). The manuscript could be much stronger in its message with the addition of much more data and with attention to analyses that better parse different definitions of location as well as movement variables (as in rodent work on mapping of environmental spaces).

1. The dataset is simply too small to reach conclusions and generalizations about the properties of PPC and HPC neurons and, certainly, the extent to which they differ between these regions. Based on the small differences (even if statistically significant), one would be hard-pressed to state that HPC and PPC are qualitatively different in any way. Perhaps that is the message itself, but showing this would require a much more detailed and larger-N assessment of spatial firing characteristics in the spaces defined by the environment or by the trajectories along paths that undoubtedly accompany task performance and which can be dissociated from environmental space.

2. As it stands, the primary method for assessing spatial firing comes primarily from the calculation of spatial information, an approach that is flawed in that it actually speaks only to the distribution of firing rates across all locations. The definition of spatial firing as that above randomized data constitutes a low bar for defining neurons as tuned to maze location. Certainly, the example neurons show some sign of this, but the nature of the full dataset remains in question. The analysis should include measurements of reliability in the firing of neurons at specific locations as well as spatial coherence in firing. The measure of spatial information utilized here obtains the exact same result whether higher firing rates are observed across adjacent regions of the environment or whether they are distributed randomly. Further, this coarse measure leads the authors to confuse "place cells" with cells that have multiple fields with field distribution organized relative to the maze's visual appearance or relative to trajectories through the maze.

3. It would help for the authors to consider examining neural activity according to trajectories (e.g., as in path-equivalence – Frank and Wilson, 2000), according to head orientation, according to optic flow, and according to the control of movement with the joystick. For a manuscript that seeks to characterize PPC and HPC firing properties during complex behavior in a complex environment, such examinations are expected and can help to determine whether it is more appropriate to explain the observed structured firing on the maze as reflective of visual responses in egocentric coordinates or true spatial responses relative to the distribution of landmarks. Sampling across these variables is necessary and manipulation of landmarks (rotations, deletions, etc.) is needed.

4. The saccade differences across paths seem more similar than different (figure 2B) and the colormap of figure 1A is left undefined.

Reviewer #2

(Remarks to the Author)

This manuscript by Vericek and colleagues touches upon an important issue, i.e., the differential role of primate hippocampus (HC) and posterior parietal cortex (PPC) for (virtual) navigation. The authors had trained two macaque monkeys in a virtual navigation task in which monkeys had to find and travel toward a target in a 5-arm maze. The task could only be solved by processing self-motion induced optic flow and considering the spatial location of landmarks. The authors recorded neural activity from HC and PPC concurrently with eye movements. The structure of the task was different from typical short-trial based tasks, in which the visual input is well-controlled due to fixation of the experimental animals. Hence, the major challenge in this study was to develop new suitable analyses. The manuscript summarizes numerous statistically significant findings. Overall, the study shows links between the encoding of place and view, key features which had been reported before for HC and para-HC neurons, modulated by active navigation.

This study is exciting and timely. At the same time, it differs from typical studies in the field of visual neuroscience and to my taste some additional analyses are required to link these new and thrilling data to the existing literature. Overall, the following remarks (in chronological order) might help to further improve the manuscript. Unfortunately, the manuscript did not come with line numbers. So, I must refer to page numbers.

Major concerns

Page 3, last paragraph: a mean firing of 6.2 Sp/s appears rather low to me. Does this value refer to spontaneous activity. Then it would be low but okay. Yet, if this refers to stimulus driven activity, this would worry me. The authors must resolve this issue.

Page 4, last paragraph. The authors decided to focus on saccades and fixation. Yet, it is known from the literature that (visually simulated) self-motion leads to optokinetic like eye movements (e.g., Lappe et al., J Neurophysiol, 1998; Bremmer et al., Nature Comms, 2017). The authors should go beyond their current analyses and quantify periods of OKN and related neural activity.

Page 5 and following: as detailed above, I understand that the experimental approach differed from classical approaches in visual neuroscience. Yet, to be able to link the current results with existing data: do the authors have any data concerning the location and size of visual receptive fields of the neurons under study? If yes, these must be mentioned.

Page 6 and following: As correctly pointed out by the authors, neurons in area VIP are known for e.g., their response to (visually simulated) self-motion. I missed analyses which aimed to quantify the neural responses with respect to ongoing self-motion, either translation or rotation. As an example: could, what the authors show e.g., in figure 1F be optic flow (OF) responses?

Page 7 and following: if I understand it correctly, the authors only considered the horizontal component of the eye movements for their analyses? What influence did the vertical components have (see e.g., Noel et al., eLife, 2022)?

Page 8 and more general: why did the authors not use a six-arm maze? That would have been compatible with the periodicity of grid cells. So, what was the benefit of 5 over 6 arms?

Page 11: how sure were the authors to not record from area MIP? Did the authors check for hand movement related neural activation?

Figure 2 and related to my previous comment: how sure are the authors they did not record from area 7a? Panel 1G shows that the peak of saccade related discharges of most parietal neurons occurred post-saccadically. Yet, following Barash et al., J Neurophysiol., 1991, on average the peak should occur pre-saccadically.

Figure 2, continued: I am worried by the average saccade length of 93 ms. I doubt this can be correct. Such a large value would imply average saccade sizes of 30 - 40 degrees or even more, which I consider unlikely.

Minor concerns

Page 5 and following: I appreciate the level of detail of statistical analyses. Yet, incorporating all the statistics in the main text hinders reading. I suggest to somehow condense the statistics provided in the main text and provide some supplementary information with all the results displayed in detail.

Page 13, first paragraph: I find the statement a bit weird that the authors could not differentiate between LIP and VIP based on their activity profile. Cells in these two areas have remarkably distinct response features.

Figure legend on Page 25 and Methods: the authors should provide the size of the projection screen also in degrees of visual angle.

Page 35: I consider a Gaussian Kernel with $\text{std}=100\text{ms}$ to be extremely large. The authors must comment on this selection.

Page 36: I did not understand “within a distance of 5.6 units”.

Page 38: from the anatomical sections I was wondering about the recording approach. Was the electrode track parallel to the sulcus. Or did the authors insert the electrodes vertically in stereotactic coordinates?

Page 38: I guess the reference is somehow corrupted “determined by Fast Fourier transform (M.Sc. Eng. Hristo Zhivomirov, 2014). “

Page 40: are the authors sure about the labels of their anatomical sections? The form of the IPS at “AP 1” looks more posterior to me than the shape at “AP -1”. Are all panels shown at the same scale? Compared to the other panels, the IPS at “AP 8” looks rather deep to me, at least deeper than to be expected at this rather anterior position. And in relation to one of my above comments, a few recording sites shown in this panel (AP 8) could be in MIP rather than LIP/VIP.

Reviewer #3

(Remarks to the Author)

The present paper examined neural activity in the posterior parietal cortex and hippocampus during two macaque monkeys performed vision-based navigation in a virtual maze. The authors compared position-related activity, saccade-related activity, gaze-related activity and landmark-appearance activity between the two areas. While both areas substantially show task-related activity, the activity in the parietal cortex was related to anticipation of reward or necessity of choice compared with the hippocampus. It is very timely to investigate neural process of the primate hippocampus and parietal cortex in terms of “place” and “view.” The present paper challenged an important question in cognitive neuroscience including both perception and memory particularly for navigation. I agree this study is valuable, but I found several unclear points that should be addressed as following.

1. Authors examined only correct trials for analyses of neural activity and saccades. Because this procedure made asymmetry between the reward path (center to peripheral) and other four inbound paths (peripheral to center) in animals' virtual movement, this point (the usage of only correct trials for analysis) should be noted explicitly in Result Section. Authors should also display the asymmetry of data structure clearly. I may suggest adding arrows indicating direction of movements to Figs. 1B and 2A.
2. “We first examined whether parietal or hippocampal cells were modulated by animal's virtual position, i.e. the camera's location within the environment, independent from its orientation” (Page 3): What does the phrase “independent from its orientation” mean? The animal's head direction in the virtual space did not change at each position in the inbound paths and the reward path (except for center).
3. Please show the behavioral data related with saccade for each animal (Figs. 2A and 2B).
4. “the absolute values of correlation coefficients were significantly higher in PPC ($\mu|r|=0.11\pm 0.015$) than in HPC ($\mu|r|=0.095\pm 0.012$; tailed-Wilcoxon rank sum: $Z=1.65$, $p=0.049$), showing that parietal cells tended to be more positively or negatively saccade-responsive than HPC ones.” (Page 5): The correlation between virtual position neural map and saccade frequency map can be modulated by common factors such as an attention effect. It is important to remove those confounding factors from the analysis if the authors want to suggest “saccade-responsive” beyond the co-variation.
5. “we first determined which positions corresponded to the entrance of a landmark in the FOV, on average (the different objects could slightly vary in size). Then, the appearance times were defined as the timestamps at which the camera was at those positions.” (Page 39): Does the “entrance” indicate appearing of the first pixel, center or whole of a landmark (on average)? How long did it take for the appearance of a landmark from its first pixel to its entire shape?
6. “More parietal neurons were modulated by animal position in the virtual maze than hippocampal ones” (Page 10): Because the present experimental and analytical conditions contain several confounding factors (visual stimuli, active vs passive, reward), the above sentence might be misleading as a summary / conclusion.
7. Was there any difference in response patterns among recording sites in the hippocampus (e.g., CA1 - CA3, anterior-posterior)?
8. “Taken together, these findings shed light on the neural processes that link place and view.” (Page 1): Please explain this sentence more by specifying the linking between place and view in the present study.

Reviewer #4

(Remarks to the Author)

Visual information is key to spatial navigation in humans and non-human primates. The aim of this study is to better understand the role of the posterior parietal cortex (PPC), an important area for high-level visual information processing in human and non-human primates, in goal-directed navigation. Vericel et al. used tetrode recordings in the intraparietal sulcus of the PPC and in the hippocampus while head-fixed monkeys navigated a virtual environment (5-arm star maze) to find uncued locations associated with rewards. These recordings were combined with video monitoring of eye movements to analyze saccades and fixations in particular. They found that the activity of the majority of PPC cells was modulated by position, but in a stereotyped way according to the ongoing action. The activity of a significant proportion of HP cells was also modulated by position in a more distributed way (several place fields per cell). As virtual navigation triggered a high rate

of saccades to probe relevant landmarks in the environment (probably increased by the fact that the animals were head-fixed), the authors then analyzed the modulation of neuronal activity by saccades. The activity of the majority of PPC cells and a significant proportion of HP cells was modulated (increased or decreased) around saccades. As the saccade rate was variable in different locations, the authors then analyzed the spatial distribution of saccades (saccade spatial map) and how it correlated with neuronal activity maps. The two spatial distributions were correlated only for a minority of cells, so that saccade-induced modulation of activity alone cannot explain the modulation of PPC and HP cells by position. Neurons in the PPC were highly sensitive to the fixation of visual landmarks compared to paths, especially for landmarks close to the reward, and were active before the appearance of the landmark, suggesting that they have access to a spatial map of the environment. In conclusion, both PPC and HC cells show some modulation by position, but PPC activity is more likely to reflect a task-relevant representation of visual space.

Overall, this manuscript is a technical tour de force. The authors were able to train monkeys to perform a goal-directed navigation task in virtual reality and record activity in two key regions for spatial navigation (hippocampus and posterior parietal cortex) together with eye movements (to assess saccades and fixation). Importantly, the authors report convergent and divergent coding schemes in these two areas. Most interestingly, they observed a strong modulation of hippocampal activity by saccades, classically associated to PPC cells, and conversely, PPC activity shows some anticipatory activities usually associated with hippocampal spatial mapping. Altogether the results suggest a cooperation between the two structures allowing visually guided goal directed navigation.

Major concerns:

1- In rodents, the modulation of the firing rate of hippocampal cells by position is often estimated over several trials. To be considered a place cell, a pyramidal neuron must discharge reliably at the same position over time. On average, how many trials were used to assess spatial selectivity? Could you show the trial-by-trial modulation of hippocampal and PPC cell activity as in Figure 2 C and D for saccades? How consistent was the spatial modulation across trials (spatial stability)?

2- The different behavioral phases (entry path to the center, exit path to the reward and passive return) are associated with different motivational and behavioral states (active versus passive; before versus after reward consumption) that could lead to different hippocampal spatial maps. Could you analyze the location selectivity of HPC and PPC activity separately for the different phases (inbound path to the center, outbound path to the reward or return path)?

3- HPC cells have previously been described as 'spatial view cells', but it is unclear from the present results whether the HPC cells recorded have this property. Were HPC cells modulated by where the animal was looking at in a manner reminiscent of spatial view cells? Could you discuss this point?

4- Previous work in freely moving monkeys has shown that HPC activity is strongly modulated by the direction the animal is looking at (in 3D space) and by head tilt. However, in the present experiment, several head movements are restricted by head fixation. The authors should discuss how this behavioral constraint might affect the results they observed.

5- It is unclear whether the sampling of different locations by the animals was uniform or biased. The authors report two experimental phases each day: one in which the animals learn the location of the reward location and presumably sample each arm of the maze in search of the reward, and a phase in which the reward location is learned and the animals presumably mostly visit the rewarded arm. Could the authors show examples of trajectories made by the animal at different stages of training? When was the modulation of PPC and HC cells by location assessed? If there was a sampling bias, could this have affected the reported results?

6- Along the same lines, animals could use different navigational strategies at different stages of learning the reward location. For example, navigation could be based more on the external landmark at the beginning of the learning (place strategy) and on movement sequences in later phases (when animals only use landmarks to know the starting location then use only sequences of movements to reach the correct arm). Could the authors compare the occurrence of saccades and fixations before versus after the learning of reward location? Similarly, could the authors compare the modulation of PPC and HC cells by location, saccades, landmarks and path before versus after the learning of reward location? This is important as HC cells might be more modulated by position in an allocentric reference frame early during learning, but more modulated by position in an egocentric reference frame late during learning.

Minor:

1- Could you show the recording sites for hippocampal recordings as well as in Fig. S1?

2- Fig. S1 G rightmost example, it seems to me that this cell is symmetrical for outgoing paths?

3- Were there any changes in theta rhythm activity between incoming (active) and outgoing (passive) pathways?

4- Typo in P14, first paragraph, sentence beginning with "Our results are consistent with "this" hypothesis" should be "the" hypothesis.

5- Could you report the information content sparsity index and depth of tuning for each illustrated cell in main and supplementary figures.

Jérôme Epsztein

Version 1:

Reviewer comments:

Reviewer #1

(Remarks to the Author)

The revised manuscript by Vericel introduces few new analyses and no new data. As a result, I find myself less convinced than before that the work can deliver a well-evidenced message that reveals something new about hippocampal (HPC) and posterior parietal cortex (PPC) function in navigation.

As before, there are too few neurons recorded in PPC and HPC to reach meaningful comparisons between the 2 regions. The authors point to technical reasons for low neuron numbers, but none stand as valid reasons to accept a less than reasonably complete dataset. To add to this, the observed differences are not large even where they are statistically significant and they sometimes concern values (correlations or information) that indicate less than robust tuning of spiking activity. Combined, these reasons make it impossible to make strong statements about complementary function between the two regions.

The observed information values are low. The 1st-2nd half reliability correlations for spatial firing patterns are low, especially as compared to data from track environments in rodents for both PPC and HPC. The authors chose not to invoke a coherence analysis to complement the information metric despite and cite incorrect reasons as to why it cannot be carried out. The correlations of neural activity for different orientations to the path versus landmark sinusoid are low. Saccade and landmark (equivalent to object) responses of primate PPC neurons are not novel findings. The manuscript lacks a coherent message that is strongly evidenced.

Line 160 – In a variety of studies (e.g., Johnson/Redish J. Neurosci.), orientation-specific responses of place-specific neurons have been observed. The statement here is incorrect.

Reviewer #2

(Remarks to the Author)

The authors have revised their manuscript substantially along the points raised by the reviewers. Speaking for those points raised in my review: the authors have addressed all of them carefully and convincingly. I have no further comments.

Reviewer #3

(Remarks to the Author)

The authors addressed all my concerns appropriately. They did a quite good job, and the manuscript deserves for the publication.

Reviewer #4

(Remarks to the Author)

In their revised manuscript, Vericel et al. have included several new analyses that support their results and strengthen the manuscript. In particular, they better characterized the reliability of the spatially modulated firing of HPC and PPC cells by plotting rasters, and several examples convincingly show that the activity of these cells is indeed spatially modulated, as they consistently discharge when animals pass through the same virtual locations. They also correlated the spatial stability of firing (assessed by correlating neural maps between the first and second half of the session) with the number of fields, field size, and firing rate peaks, and found correlations consistent with higher stability of more spatially modulated cells (cells with a low number of place fields and high firing rates).

However, I found the interpretation of the results as presented in the discussion unclear P12 Ln 475 “We suggest that the apparent parietal position code can result from visual processing of relevant cues, contributing to a task-relevant representation of visual space, rather than pure place code for self-position in Euclidian coordinates. Further, we also extend the latter interpretation for the position selectivity found in hippocampal cells in this task. Indeed, unlike rodents place cells (Andersen et al. 2006, but see Acharya et al. 2016; Jercog et al. 2019; O’Keefe and Dostrovsky 1971; Rubin et al. 2014), cells we recorded displayed small and unstable spatial fields and orientation selectivity in the centre of the maze, compatible with an active task-dependent visual processing of the scene.”

It is unclear to me why small and unstable fields would be more compatible with active task-dependent visual processing of the scene than, say, large and stable place fields. To me, active and task-dependent visual processing indicates that the animals are engaged in the task, and this should be associated with higher and more stable spatial modulation in the hippocampus, as recently shown in rodents (Pettit et al., Nat Neurosci 2022, doi: 10.1038/s41593-022-01050-4.).

The authors also included a more detailed analysis of the activity of PPC and HPC cells in different paths of the track and found that landmark cells are more recruited in the inbound paths and path cells are mostly active during the rewarded path and the outbound paths. This much more detailed analysis adds significant new information to the manuscript.

Overall, the authors have performed new analyses and further discussed their results in relation to the previous literature.

To conclude the authors have addressed my concerns with new analyses and discussion and the manuscript has been strengthened as a result.

Minor:

P9 Ln348 first sentence. The sentence is confusing. First, it is unclear what the "partition in two" means, and we have to refer to the Figure 3S-V legend to figure it out, which is not convenient. Also, it is not clear to me how plotting landmark and path responsive cells separately will tell us about the relationship between view response and virtual position.

We thank reviewers for their valuable comments. We address the comments below, in purple font, and the changes in the manuscript are indicated in blue.

Reviewer #1 (Remarks to the Author):

Vericel et al. report on recordings from the primate posterior parietal cortex (sub-regions VIP and LIP) and the hippocampus (sub-regions unspecified) during performance of a star maze virtual reality task (restrained with joystick control of movement). At issue is the responses of neurons to either maze locations or to landmarks and the comparison of posterior parietal cortex (PPC) and hippocampus (HPC). The authors report that more neurons in PPC exhibited spatial selectivity and that such selectivity was driven by visual responses to maze pathways or to landmarks.

While the responses of some neurons are indeed interesting and spatially structured, in some way, with relation to the animal's experience and behavior in the environment, the data is based on a small population of neurons. The analysis and characterization of the spatial responses do not adequately explain the responses across different maze locations and the contrasts between PPC and HPC are of minor to moderate strength. As a result, the manuscript fails to build a novel conceptualization of the role of PPC versus HPC in task performance and the visual responses are less rigorously characterized than one would normally expect (if the experiment and results were analyzed primarily from a object and visual field location perspective). The manuscript could be much stronger in its message with the addition of much more data and with attention to analyses that better parse different definitions of location as well as movement variables (as in rodent work on mapping of environmental spaces).

We thank the reviewer for their evaluation of the manuscript which provides valuable feedback. In order to address the reviewers' points, we brought additional analysis to characterize the definition of "location", head orientation and movement variables. We modified the first paragraph to convey that our point is not to describe "place" like activity, but to understand the variables that could explain the activity of some cells in specific locations. For example, if a cell's activity plays a role in oculomotor control to support eye movement and that animal's make saccades at specific locations, this would be manifest as a place-like activity, although this would not represent place encoding per se. We tried to clarify this in the manuscript and hope to convince the reviewer that we bring an interesting new result. To our knowledge, this manuscript represents the only manuscript which analyses neural activity as a function of saccades, position of gaze, in a landmark based spatial navigation task, comparing activity in the hippocampus to that in the intra-parietal sulcus. Our results put forward that place like encoding can emerge from a combination of sensorimotor and cognitive processes co-occurring in specific locations. While behaviour and maze processing is the same for parietal and hippocampus, the analysis of their respective activity reveals recruitment at different locations, hence, likely associated with different functions. This complements our previous work showing that hippocampal activity recorded in this task, did not appear exactly as encoding place as in rodent place cells (Wirth *et al.* 2017; Baraduc *et al.* 2019). The present results show how, in a region that controls saccades and fixations, activity is not a simple reflection of oculomotor dynamics but carries a task-relevant representation of space, that is different and complementary to that of the hippocampus. We hope that the reviewer will agree that the overall quality of the manuscript is improved and the paper provides a solid new take on the construction of task relevant representations through eye movements uncovering how important features of the environments are processed.

1.1. The dataset is simply too small to reach conclusions and generalizations about the properties of PPC and HPC neurons and, certainly, the extent to which they differ between these regions. Based on the small differences (even if statistically significant), one would be hard-pressed to state that HPC and PPC are qualitatively different in any way. Perhaps that is the message itself, but showing this would require a much more detailed and larger-N assessment of spatial firing characteristics in the spaces defined by the environment or by the trajectories along paths that undoubtedly accompany task performance and which can be dissociated from environmental space.

We agree with the reviewer that the number of cells appears small relative to conventional recordings with tetrodes performed in the rodents and more recent techniques resting on Neuropixels. However, in contrast to electrophysiology performed in rodents, acute recordings in the monkeys proceeds differently when single electrodes are used, as the experimenter selects single cells online whose waveforms can be well identified. These contrasts with recordings performed in the rat or mice in which cells are identified as the animal is at rest, and then brought in the open field or the maze. While we understand that ideally such method should be used in the monkey, it is not feasible with acute recordings as the electrode is lowered in the chamber as the animal is awake. Further, the position of the hippocampus as a deep structure in the primate poses specific technical challenges that do not exist in the rodent. Last, there are limitations about how long the animal can tolerate to be maintained with the head fixed while in the lab. One of the advantages is that one could argue that all cells examined here are recruited by the task. Further, we discarded for the analysis many cells whose activity was below 100 spikes per session to avoid spurious statistics, and for which the monkey did not perform a minimum of 40 correct trials. While one probably would augment the number of cells using new recording methods developed more recently, these were not available at the time this experiment was conducted, hence the smaller number. Finally, while in the rodent it is possible to add another rat to the sample, there are limits to the number of monkeys used in an experiment. We would like to point out that many functional properties of the lateral and intraparietal sulcus in the primate (a subpart of the posterior parietal sulcus, which also comprises MIP, 7a, 7m) have been identified with such low or even lower number of cells. So, in sum, while these numbers contrast greatly with the usual numbers in the rodent, they are perfectly in line with classic electrophysiological recordings performed in the monkey. We nevertheless agree that with the current developments of Neuropixels and generalization of linear arrays, these standards will probably get closer to the rodent's in the future.

1.2. As it stands, the primary method for assessing spatial firing comes primarily from the calculation of spatial information, an approach that is flawed in that it actually speaks only to the distribution of firing rates across all locations. The definition of spatial firing as that above randomized data constitutes a low bar for defining neurons as tuned to maze location. Certainly, the example neurons show some sign of this, but the nature of the full dataset remains in question. The analysis should include measurements of reliability in the firing of neurons at specific locations as well as spatial coherence in firing. The measure of spatial information utilized here obtains the exact same result whether higher firing rates are observed across adjacent regions of the environment or whether they are distributed randomly. Further, this coarse measure leads the authors to confuse “place cells” with cells that have multiple fields with field distribution organized relation to the maze’s visual appearance or relative to trajectories through the maze.

We agree with the reviewer that if a cell's activity is modulated by the position variable, this, however, does not suffice to qualify it as a place cell. In fact, we actually hope that the present work helps supporting this very idea. We apologize for the lack of clarity in the use of the formulation “spatial

modulation”. On the contrary to concluding on the presence of a place code, we aimed to convey in the manuscript that the cells responded in some position/locations, likely because the animal made a saccade at a specific place or fixating a specific object that is relevant for successful navigation. However, the position variable is the first one that we attempted to characterize because of its prevalence in the rodent literature. What is interesting is that this first characterization of a coarse spatial modulation reveals that cells in the lateral and ventro-lateral bank of the parietal sulcus (with mostly gaze-related activity) are much more recruited during the landmark-based navigation than hippocampal place cells were. From there, we aimed to understand what drove this seemingly spatially modulated activity and analysed the activity of the cells as a function of saccades and fixations. However, to provide a better account of the spatial modulation of the cells and to clarify that these should not be mistaken for place cells, we now have added several analyses below, such as characterizing the stability of the spatial modulation as a function of number of fields, field size and peak activity (see also response to reviewer 4.2 for the full stability-related analysis). Overall, while 56.0% of PPC cells and 40.6 % of HPC display stable spatial modulation, many cells which have many place fields of small size with a low peak activity are unstable and would not qualify as place cells (Figure S1G-I). We also characterized the activity of cells with respect to orientation, as this could be done in the centre of the maze and showed that many spatially modulated cells are also modulated by orientation (see response to R1.3 hereafter). Overall, this suggests that, in this maze navigation task, the majority of the cells do not encode the current position of the animal stably throughout the session, but may be driven by other variables, which could be motor (joystick movement, eye movements), visual (fixations, optokinetic nystagmus) or motivational (reward or attention to landmarks) which we characterized and linked with positions. We present the new measures below:

First, to assess the stability of the cells’ spatial modulation through time, we computed a cross-correlations of the place maps (see also response to reviewer #4, point 1) between the first and second half of the session, testing the significance of the correlation against spike permutations. We showed that in the PPC and in the HPC, the neurons with a lower number of place fields displayed more stability through time (i.e. a higher correlation coefficient) than the ones with multiple fields (Pearson correlation: $r=-0.30$, $p=0.0043$ for PPC, and $r=-0.46$, $p=2.44 \times 10^{-4}$ for the HPC, Figure 1.1, below).

Figure R1.2A: Scatter plot of each place-modulated cell’s number of spatial fields (X axis) as a function of their stability correlation coefficient (Y axis), for the PPC (N=91; left, in red) and the HPC (N=64; right, in green). Filled markers represent cells with a significant positive correlation between first and second halves of the session (Pearson correlation: $r \geq 0$ and permutation test: $p \leq 0.05$). The equation of the regression line is indicated on the top right corner.

Next, we examined the link between spatial stability and the size of spatial fields (SFs). We found that neurons with larger SFs were more stable, in both PPC (Pearson correlation: $r=0.51$, $p=2.81 \times 10^{-7}$) and HPC ($r=0.65$, $p=1.43 \times 10^{-8}$) compared to cells with smaller spatial fields (Figure 1.2.2, below).

Figure R1.2B: Scatter plot of each place-modulated cell's total size of spatial fields (X axis) as a function of their stability correlation coefficient (Y axis), for the PPC (N=91; left, in red) and the HPC (N=64; right, in green). Filled markers represent cells with a significant positive correlation between first and second halves of the session (Pearson correlation: $r>0$ and permutation test: $p \leq 0.05$). The equation of the regression line is indicated on the top right corner.

Finally, when examined the link between spatial stability and peak firing rate, we found that the neurons with higher peaks rates were also more stable, in both PPC (Pearson correlation: $r=0.38$, $p=2.13 \times 10^{-4}$) and HPC ($r=0.41$, $p=8.12 \times 10^{-4}$, Figure 1.2.3) suggesting that the strength of the recruitment of the cell was linked with a higher stability.

Figure R1.2C: Scatter plot of each place-modulated cell's peak firing rate (X axis) as a function of their stability correlation coefficient (Y axis), for the PPC (N=91; left, in red) and the HPC (N=63; right, in green). Filled markers represent cells with a significant positive correlation between first and second halves of the session (Pearson correlation: $r>0$ and permutation test: $p \leq 0.05$). The equation of the regression line is indicated on the top right corner.

Finally, because of the star shape nature of the maze, we decided to not compute spatial coherence, akin to what is classically done in the literature, as the linearity would not enable us to correlates across

enough neighbouring bins, and yield spurious statistics depending on positions (intersections vs centre and straight segments).

However, we added a section to present the results relative to the stability and field size, number, peak in the main text page 4, line 142, and in Figure S1G-I (see also response to reviewer #4.1):

“Stability of the spatial modulation was related to the number of fields and to their size: cells with fewer and larger fields were more stable through time than cells with numerous and smaller fields (Pearson correlation between spatial correlation coefficient and number of SFs: $r_{PPC}=-0.30$, $p_{PPC}=0.0043$; $r_{HPC}=-0.46$, $p_{HPC}=2.44 \times 10^{-4}$; Figure S1G; correlation with SFs size: $r_{PPC}=0.51$, $p_{PPC}=2.81 \times 10^{-7}$; $r_{HPC}=0.65$, $p_{HPC}=1.43 \times 10^{-8}$; Figure S1H). Further, neurons with higher peaks rates were more stable than others, in both PPC ($r=0.38$, $p=2.13 \times 10^{-4}$) and HPC ($r=0.41$, $p=8.12 \times 10^{-4}$, Figure S1I). These results suggest that, in addition to cells presenting a stable modulation, many displayed brief spatial modulation unlike rodent place cells. Next, we examined whether position-modulated cells were sensitive to head orientation. This analysis could only be performed in the centre of the maze, since in other locations, place and direction covaried (see Methods).”

We as well added more details on the properties of the SFs, notably the distance separating them, within cells, page 4, line 130:

“In both areas, cells often expressed multiple spatial fields (SFs; see Methods), but they didn’t differ across regions by SFs number ($\mu N_{PPC}=3.65 \pm 0.33$ SFs, $\mu N_{HPC}=3.66 \pm 0.34$ SFs; Wilcoxon rank sum: $|Z|=0.067$, $p=0.95$) or Euclidean distance separating them ($\mu Dist_{PPC}=12.68 \pm 0.86$ units, $\mu D_{HPC}=11.88 \pm 1.12$ units in the HPC; $|Z|=0.85$, $p=0.40$).”

See the respectively associated Method sections, page 36, line 1253:

“Assessment of spatial stability. To assess the stability of the cells’ spatial modulation through time, we computed for each of them a cross-correlation between the mean neural place maps of the first and the second halves of the session. The significance of the correlation was determined by permutation test, with 999 iterations of surrogates. Then, we aimed to explain the stability of the neuron’s response as a function of other electrophysiological factors, that are the number of spatial fields of the neuron, the summed size of these spatial fields, and the neuron’s peak firing rate. To do so, we performed Pearson correlation tests between the stability correlation coefficients of the neuronal population, computed previously, and the three predictors. “

and page 36, line 1248:

“Characterization of spatial fields. Spatial fields were defined as contiguous place bins with neuronal activity above the threshold of mean + 2SD of the activity over the whole space. Distances between spatial fields were computed in “virtual meter unit” coordinates, where 1 bin equals 1.6 units. The inbound paths were divided into 10 bins (16 units), short returns into 12 bins (19.2 units), and long outbounds into 19 bins (30.4 units).”

To accommodate this insertion, we also modified the result section slightly, stressing out how the cells described in the present study differ from the rodent place cells:

- p.4 line 134: “Because the spatial properties of the recorded neurons may echo rodent place cells (Aronov and Tank 2014; O’Keefe 1979), we conducted additional analyses in order to comprehend to which extent they resemble or differ from place cells.”
- p.4 line 158: “The results suggest that the spatially-modulated PPC or HPC cells were not akin to classic rodent place cells, which encode the animal’s position per se and are invariant to head orientation (Andersen et al. 2006, but see Acharya et al. 2016; Jercog et al. 2019; O’Keefe and Dostrovsky 1971; Rubin et al. 2014). Rather, many cells displayed brief and transient spatial modulation and selectivity to orientation in addition to position, which may be driven by other variables than position itself such as direction or view.”
- p.5 line 194: “These spatially-modulated neurons in the HPC and the PPC displayed distinct properties than these of the classical place cells.”

We finally modified the discussion to account for the interpretation of place encoding in our set-up, page 12 line 475:

“We suggest that the apparent parietal position code can result from visual processing of relevant cues, contributing to a task-relevant representation of visual space, rather than pure place code for self-position in Euclidian coordinates. Further, we also extend the latter interpretation for the position selectivity found in hippocampal cells in this task. Indeed, unlike rodents place cells (Andersen et al. 2006, but see Acharya et al. 2016; Jercog et al. 2019; O’Keefe and Dostrovsky 1971; Rubin et al. 2014), cells we recorded displayed small and unstable spatial fields and orientation selectivity in the centre of the maze, compatible with an active task-dependent visual processing of the scene.”

1.3. It would be help for the authors to consider examining neural activity according to trajectories (e.g., as in path-equivalence – Frank and Wilson, 2000), according to head orientation, according to optic flow, and according to the control of movement with the joystick. For a manuscript that seeks to characterize PPC and HPC firing properties during complex behavior in a complex environment, such examinations are expected and can help to determine whether it is more appropriate to explain the observed structured firing on the maze as reflective of visual responses in egocentric coordinates or true spatial responses relative to the distribution of landmarks. Sampling across these variables is necessary and manipulation of landmarks (rotations, deletions, etc.) is needed.

We agree that determining the egocentric or allocentric nature of the visual responses is of great importance. Ideally, it would have been relevant to compute the activity for the same trajectory (i.e. a movement to the left) in different locations, to interpret whether the activity reflects a specific sequence of movement independent of the location (Frank et al. 2000; Nitz 2006; Shelley and Nitz 2021), but the layout of the maze we used does not fully allow this, as the goal location is spatially invariant through the trials. We performed several alternate analyses to overcome this limitation.

First, we examined the activity of the cells as a function of head orientation in the centre, which is the only position with multiples head orientation which do not covary with the position. We reasoned that if cells encoded the centre as an allocentric position, then, these cells should not exhibit any head orientation selectivity. The results are detailed a few paragraphs below.

Second, we compared the same movement (Left or Right joystick turns) for different trajectories. This is not exactly alike to Frank and Wilson, because our trajectories consist of 1 or 2 successive movements to the right or the left. However, by aligning the activity on the movements, this should distinguish self-

centred movements from other variables such as positions or views. This section is detailed in response to reviewer #2, point 7. Briefly, results show that cells did not selectively respond to self-centred left or right movements independently of the trajectory, but encoded something relative to elements present in the field of view or to the progression along a trajectory. Further, the absence of self-centred hand-movement encoding also fits the recording location which is targeting VIP and LIP rather than MIP, in which reaching and grasping hand movements have been described (Grefkes and Fink 2005).

Ideally, we acknowledge that measures of the visual receptive fields would have enriched characterization of the cells. Enough time for a proper test of the visual and saccadic responses of these neurons in space was nevertheless not available, in addition to the navigation sessions. An indirect measure stemming from the analysis of the exploration behaviour could be hoped for. Yet, in most segments of the maze, animals moved their eyes at the same time as they progressed in the maze, leading to changes in the visual scene and objects entering and exiting the FOV at the same time. This greatly challenges a fine evaluation of the visual receptive fields, which are known to be skewed to the contralateral side, yet fovea-centred (Barash *et al.* 1991a; Ben Hamed *et al.* 2001; Blatt *et al.* 1990; Viswanathan and Nieder 2017). However, we noted that in the parietal cortex, the population responded more to landmarks entering the field of view on the animal's left, which is in line with a stimulation of a receptive contralateral to the recording site which, in this case, would be egocentric coordinates. We discussed this page 15 line 582:

*“Our results showed that, in accordance with the existence of a self-centred reference frame previously described in literature, parietal neurons responded to the entrance of a moving object in the field of view (Bremmer *et al.* 2002b; Bremmer *et al.* 1997; Duhamel *et al.* 1998). While individual cells preferred the left or right side of appearance, the whole population did show a preference for the contralateral hemifield, as often presented in the literature (Duhamel *et al.* 1998; Sereno *et al.* 2001; Wardak *et al.* 2002).”*

What may be interesting is that while there is a higher representation of contralateral FOV, as animal's move their eyes, the resulting map isn't asymmetrically biased, but rather biased for meaningful and salient landmarks, such as the ones neighbouring the reward (page 11, line 413).

We present below the results in detail here for the analysis as a function of head orientation, which we summarized in the main text (p.4, line 153), while the analysis as a function of joystick turns can be found in response to reviewer #R2.7, and in the manuscript page 5 line 184. The analysis of the neurons' optic-flow-related properties are detailed in response R2.4.

Activity of the cells as a function of head orientation: To determine whether cells were sensitive to the orientation of the camera in the virtual maze, we examined their activity in the centre as a function of the virtual orientation of the animal (Figure R1.3A-B, see more on Figure S1J-K). Cells were considered as significant if their tuning curve was significantly different from what would be expected by chance. Many more of the parietal cells 89/111 (80.2%) were selective to the HD compared to hippocampal ones 73/142 (51.4%). If only considering cells with significant spatial IC, 76/91 (83.5%) of PPC cells and 38/64 (59.4%) of HPC ones were orientation-selective. This suggests that the majority of the parietal cells and a large proportion of HPC cells expressed different properties than place cells because they are, among other properties, selective to the orientation of the monkey. Interestingly, although more cells responded significantly differently to the head orientation in the parietal cortex compared to the hippocampus, the depth of tuning (DT, see Methods) was not significantly different between the two areas ($\mu\text{DT}_{\text{PPC}}=0.705\pm 0.044$, $\mu\text{DT}_{\text{HPC}}=0.68\pm 0.047$; Wilcoxon rank sum: $|Z|=0.18$, $p=0.86$). These suggest a similar sensitivity to visual cues in both regions for the active cells.

Figure R1.3A. Polar (top) and linear (bottom) plots of two example PPC cells showing a significant modulation as a function of the camera orientation, when the animal is in the centre of the maze. The North corresponds to an orientation of 0° . For these cells, the South was never faced in correct trials. The lines above the linear plot indicate orientations for which the response activity was significantly higher (dark) or lower (light) than threshold (mean of surrogate data $\pm 2.5SD$).

Figure R1.3B. Same as in A, for two example HPC cells.

We incorporated these results in the manuscript page 4, line 153:

“Amongst the cells previously identified as modulated by spatial position, a large proportion of the PPC (76/91, 83.5%) and the HPC cells (35/61, 57.4%), were also significantly modulated by the camera’s orientation in the centre of the maze (Figures S1J-K). The depth of tuning for orientation was not significantly different between the two areas ($\mu DT_{PPC}=0.58\pm 0.047$, $\mu DT_{HPC}=0.60\pm 0.065$; Wilcoxon rank sum: $Z=0.39$, $p=0.69$) suggesting a similar sensitivity to visual cues as a function of orientation in both regions.”

We as well added the corresponding method section, page 37, line 1272:

“Orientation-related activity. To assess the dependence of the neurons to the animal’s virtual orientation for a same position in the maze, we only used data corresponding to the centre, which was the only position for which the orientation varied. The activity of the neuron was computed as a function of the orientation, smoothed (with a Butterworth low-pass filter set to 2.5% of the sample rate), and the significance of the modulation was tested against a pool of 1000 surrogated data, with the threshold fixed as the mean $\pm 2.5SD$ of these surrogated data.”

Overall, in light of the additional results, we have substantially modified the first paragraph of the result section to clarify that the fact that cells displayed a modulation as a function of the position does not imply that they are place cells, rather that the cells are recruited for other variables that appear in specific position. This is in line with our previous work, in which we showed that hippocampal cells responded to a visual landmark as a function of the position, direction and side of the landmark appearance in the field of view which suggested that landmarks were encoded as a function of some egocentric viewpoint (Wirth *et al.* 2017). The current results deepen this interpretation with an analysis of cells in the intraparietal sulcus as a function of visual variables including fixations and saccades, appearance side and relevance taken in their spatial viewing context.

1.4. The saccade differences across paths seem more similar than different (figure 2B) and the colormap of figure 1A is left undefined.

All the analysis relative to the saccade related activity were recomputed completely after adjustment of the saccade detection algorithm (see also response to reviewer #2, point 9). The results were confirmed again, as we extended the analyses to each monkey separately, and although they are small, these differences are highly significant (page 6, line 214). Thus, in line with previous findings in primates (Hayhoe and Ballard 2005; Henderson 2003; Zhu *et al.* 2022; Corrigan *et al.* 2017; Lakshminarasimhan *et al.* 2020), saccades are not uniformly distributed over space and time, but rather act as a proxy for attention and action planning during goal-directed behaviours. It then makes sense that animals increase their visual exploration rate in location immediately preceding a choice, for instance. In the same way, saccades performed during the outbound paths, as landmarks are present at the two extremities of the screen, display longer durations than the rest of the maze.

For more clarity, colour bars have been added to Figures 1B-M and 2A.

Reviewer #2 (Remarks to the Author):

This manuscript by Vericel and colleagues touches upon an important issue, i.e., the differential role of primate hippocampus (HC) and posterior parietal cortex (PPC) for (virtual) navigation. The authors had trained two macaque monkeys in a virtual navigation task in which monkeys had to find and travel toward a target in a 5-arm maze. The task could only be solved by processing self-motion induced optic flow and considering the spatial location of landmarks. The authors recorded neural activity from HC and PPC concurrently with eye movements. The structure of the task was different from typical short-trial based tasks, in which the visual input is well-controlled due to fixation of the experimental animals. Hence, the major challenge in this study was to develop new suitable analyses. The manuscript summarizes numerous statistically significant findings. Overall, the study shows links between the encoding of place and view, key features which had been reported before for HC and para-HC neurons, modulated by active navigation. This study is exciting and timely. At the same time, it differs from typical studies in the field of visual neuroscience and to my taste some additional analyses are required to link these new and thrilling data to the existing literature. Overall, the following remarks (in chronological order) might help to further improve the manuscript. Unfortunately, the manuscript did not come with line numbers. So, I must refer to page numbers.

We thank the reviewer for this valuable review and their enthusiasm for the manuscript. We performed new analyses to characterize the cells as a function of optic flow and optokinetic movements, camera orientation. We are particularly grateful to the reviewer for pointing out potential inaccuracies in our computation of saccade parameters. As a result, we refined the algorithm to process the eye movements particularly to make sure that there were no artefacts due to up-sampling of the eye data with respect to the actual sampling rate. Therefore, we modified our signal processing routines and recalculated the saccades and gaze positions completely throughout the manuscript. We describe the new methodology in point 2.9.

We agree with the reviewer that because of the unconventional testing method, the manuscript falls short with respect to specific measures for optic flow and optokinetic movements in absence of concomitant visual stimulation. We described below the newly performed analysis, but have added in the manuscript only results for which no confounding factor prevented a clear conclusion. Therefore, the scope of the paper remains mainly on the comparison between parietal and hippocampal activity given saccades and fixations during navigation, although we have added analysis on the effect of hand movements and left or right optic flow resulting from the concomitant camera orientation.

Major concerns

2.1. Page 3, last paragraph line 109 : a mean firing of 6.2 Sp/s for the PPC appears rather low to me. Does this value refer to spontaneous activity. Then it would be low but okay. Yet, if this refers to stimulus driven activity, this would worry me. The authors must resolve this issue.

Indeed, the values describing the neurons firing rates referred to their overall mean activity, and not to the stimulus-driven activity. We now included the peak activity of the cells which are 14.96 ± 3.46 spikes/s for the PPC and of 7.54 ± 1.35 spikes/s for the HPC (Wilcoxon rank sum: $p=1.41 \times 10^{-5}$; page 3, line 112). Also, these number refer to peak in a “position place field”, which is different from the way parietal cells are generally described as a function of eye movements for which the peak can much higher (see Figure S1P).

2.2. Page 4, last paragraph. The authors decided to focus on saccades and fixation. Yet, it is known from the literature that (visually simulated) self-motion leads to optokinetic like eye movements (e.g., Lappe et al., J Neurophysiol, 1998; Bremmer et al., Nature Comms, 2017). The authors should go beyond their current analyses and quantify periods of OKN and related neural activity.

Indeed, some components of the task may lead to optokinetic nystagmus eye movements (OKN) during self-motion, and it was previously shown that regions such as 7a or VIP participates to OKN (Bremmer *et al.* 2002b). To test whether cells in the intraparietal sulcus were modulated during the OKN, a slow movement component of OKN was detected when the eye velocity exceeded $4^\circ/\text{s}$ for at least 100ms; to exclude any large saccadic eye movement, a maximal velocity threshold was fixed at $75^\circ/\text{s}$ (a fairly high threshold as there were ample evidence for microsaccadic eye movements during the slow pursuit component). As the origin of OKNs only existed during the centre motions and in the outbound paths, we performed the analyses only for the corresponding task epochs. Given the continuity of animals' behaviour between saccade and OKN in the VR environment, we cannot discriminate between the fast component of the OKN and a saccade per se. Thus, we combined the latter under the *saccade* label.

The figures below show examples of eye traces with the portions corresponding to detected OKNs displayed in colour. The left column shows instances when the animal turns in the centre, and the right column corresponding to instances when the animal is on the outbounds.

Figure R2.2A. Examples of horizontal and vertical eye traces in which OKN are displayed in colour, when occurring in the centre (left column) or the outbound (right column) segments of the maze. The figure illustrates how OKN events are intertwined with saccades and fixations.

The frequency of OKN behavioural events varied significantly across maze segments, with significantly higher rates in the centre compared to the outbound paths ($\mu f_{\text{centre}} = 2.056 \pm 0.071$ OKN/sec, $\mu f_{\text{outbound}} = 1.66 \pm 0.092$ OKN/sec; Wilcoxon rank sum: $|Z| = 5.89$, $p = 3.92 \times 10^{-9}$; see Figure R2.2B below).

The duration of the OKNs were also higher in the centre compared with the outbound paths ($\mu \text{dur}_{\text{centre}} = 0.27 \pm 0.011$, $\mu \text{dur}_{\text{outbound}} = 0.24 \pm 0.0091$; $|Z| = 2.98$, $p = 0.0029$).

Figure R2.2B. Scatter plots of the OKN rates (top) and durations (bottom) of each session ($N=71$), depending on the monkey's position in the maze segments where camera rotations occurred. The OKN rates and durations were significantly higher in the centre compared with the outbound paths (rates: Wilcoxon ranks sum: $|Z|=5.89$, $p=3.92 \times 10^{-9}$, $p=1.36 \times 10^{-51}$; durations: $|Z|=2.98$, $p=0.0029$).

Next, we characterized the activity of neurons depending on the OKN. When aligning the activity of the cells on the OKN start (response window: -150 to +250 ms, using the preferred eye movement direction, similarly to the method used for saccades-related activities), we identified 62/91 (68.1%) of the PPC cells as OKN-responsive. Fifty-five of them (89%) belonged to the 73 saccade-responsive cells that we previously evidenced (see Figure 2). In the HPC, 32/128 (25.0%) neurons were significantly responsive to the OKNs, of which 14 cells (31%) were part of the 45 saccade-responsive cells. Thus, there are only 7/91 (7.7%) of the PPC cells and 18/128 (14.1%) of the HPC ones clearly OKN-driven. For the majority of cells that respond to both OKNs and saccades, interpreting the results is challenging. Indeed, when aligning the saccades rates on the time of OKN start (see top panel of Figure R2.2C, below) or end (bottom panel), we observed that, as expected, the two ocular phenomena were tightly related. Indeed, as visible on an example session (saccade starts on top raster and black solid line on bottom panel; saccade ends on bottom raster and grey dashed line on bottom panel), the saccades surrounded the OKN in a narrow time window (the top panel also highlights the difficulty to identify the exact end of a saccade when it is immediately followed by the slow pursuit component of the OKN). Thus, because OKN followed or were followed closely by saccades, it is challenging to conclude on the specificity of the response of a neuron to OKN, since its response could also be related to saccades, and conversely.

Figure R2.2C. Raster plots (top and middle) and average activities (bottom) of an example session showing the saccadic activity of the animal aligned on OKN starts (top panel) and ends (bottom panel), indicated by the blue solid lines. The saccades starts are plotted on the top raster, and in black solid line on the averaged activities, and their ends on the middle raster and in black dotted line. The standard errors of the mean are indicated in light colour. The analysis window (-150 to +250ms relative to OKN starts) is represented in a blue shaded area.

Given this close relationship, we next separated the saccades depending on whether they were accompanied by OKN events or not. Specifically, we considered saccades as unrelated to OKN if no OKN occurred within the neural response window, i.e. -250 to +150ms relative to saccade ends.

Across maze segments, and congruently with the results presented above, the saccades accompanied by an OKN were on average more frequent in the centre ($53.74 \pm 0.017\%$) than in the outbound paths ($46.26 \pm 0.014\%$; Wilcoxon rank sum: $|Z|=5.34$, $p=9.36 \times 10^{-8}$). Within each segment, saccades neighboured by an OKN represented $84.51 \pm 0.016\%$ of the saccades made within the centre and $78.44 \pm 0.031\%$ of the saccades within the outbound paths, and the difference between the two proportion

was significant ($|Z|=2.08$, $p=0.037$). So, in line with what described above, the OKN-related saccades occurring in the centre and the outbound paths represent the great majority (>78%) of the saccades performed there.

Finally, we assessed to what extent a modulated response of a neuron to the OKN could explain its modulation to saccades depending on the maze segment (see Figure 2J, S2O). We previously found that 18 cells of the PPC differed by response amplitude to saccades depending on the maze segment. By separating the saccades linked with or isolated from the OKNs, we found that 12/73 (16.4%) saccades-responsive cells were significantly modulated by the presence of an OKN. Out of them, 6 (50.0%) belonged to the pool of 18 neurons whose saccade response was modulated by the maze segment. For these 6 cells, their response to OKN corresponded to the maze segment in which they also had the highest saccade related response. In other words, the cells that had a higher response for saccades neighbored by OKNs, also had a higher saccade-related activity in the centre or the outbound paths, while the cells responding to saccades without OKNs were more responsive in the inbounds or the rewarded paths. In conclusion, the analysis of OKN-related activities identified a potential explanation to the saccade-response modulation of 6/18 PPC neurons.

Because the analysis of the OKN does not shed light of the activity of parietal neurons as they co-occur with saccades and only a very small fraction of cells appeared responsive to OKN in absence of saccades, we chose to not present these results in the manuscript.

2.3. Page 5 and following: as detailed above, I understand that the experimental approach differed from classical approaches in visual neuroscience. Yet, to be able to link the current results with existing data: do the authors have any data concerning the location and size of visual receptive fields of the neurons under study? If yes, these must be mentioned.

In retrospect, we agree with the reviewer that, a clear characterisation of the receptive fields should have been performed for each cell. Unfortunately, such characterization was not included in this experiment because electrodes were lowered in the hippocampus and in the parietal cortex, and cells were not always well isolated before we started recording. While we do not have these receptive fields, we do have the position of the gaze, which informs us on activity being elicited by a foveally centred landmark (landmarks cells) or peripheral landmark (path cells).

2.4. Page 6 and following: As correctly pointed out by the authors, neurons in area VIP are known for e.g., their response to (visually simulated) self-motion. I missed analyses which aimed to quantify the neural responses with respect to ongoing self-motion, either translation or rotation. As an example: could, what the authors show e.g., in figure 1F be optic flow (OF) responses?

To quantify the effect of optic-flow (OF) variations on the activities of the neurons, we used general linear regression model. As the design of the task was not originally built to dissociate the effects of the OF-related factors, their independence was not maintained through all the maze segments, provoking rank-deficiencies in the analyses. To get around this issue, we used separated models on different data samples to account for the different ways the OF could modulate the cells' activities:

- first, we analysed the neurons FRs as a function of the translation speed and of the translation direction, that was dissociated in 3 components: the orientation of the flow (from -180 to +180°), its directionality (forward vs backward), and the interaction between the two. This analyse was performed using the data of the inbound, the rewarded, and the outbound paths.

- secondly, we analysed the FRs as a function of the rotation speed and of the rotation direction (clockwise vs counter-clockwise). This analyse was performed using data of the centre and the outbound paths.

We found that, in the PPC, 48/111 (43.2%) cells were significantly modulated by the translation speed, 33 (29.7%) by the translation orientation, 67 (60.4%) by the translation direction, and 16 (14.4%) by the interaction between the two latest components. Regarding the rotational dimension of the OF, 59 (53.2%) cells were modulated by its speed, and 57 (51.4%) by its direction. In the HPC, 46/142 (32.4%) cells were significantly modulated by the translation speed, 26 (18.3%) by the translation orientation, 54 (38.0%) by the translation direction, and 20 (14.1%) by the interaction between the two latest components. Fifty-five (38.7%) cells were modulated by the OF rotational speed, and 43 (30.3%) by its direction.

If performing the same analysis, only considering the position-modulated cells, to try to explain their spatial activity patterns, we found that, in the PPC, 44/91 (48.4%) cells were modulated by the translation speed, 31 (34.1%) by the translation orientation, 62 (68.1%) by the translation direction, and 13 (14.3%) by the interaction between the two latest components. Regarding the rotational dimension of the OF, 52 (57.1%) cells were modulated by its speed, and 54 (59.3%) by its direction. In the HPC, 28/64 (43.8%) cells were significantly modulated by the translation speed, 14 (21.9%) by the translation orientation, 34 (53.1%) by the translation direction, and 11 (17.2%) by the interaction between the two latest components. Thirty-nine (60.9%) cells were modulated by the OF rotational speed, and 26 (40.6%) by its direction.

Thus, the OF-components appeared to have an effect on a large proportion of the PPC, but also of the HPC neurons. However, it should be acknowledged that there were several confounding factors that could overlap with that of the OF patterns. Notably, the parcelling of the maze depending on the type of OF pattern that can be found there corresponds to the different maze segments (inbounds, centre, rewarded path, outbounds), across which many other variables, such as motivation, attention, saccadic activity or visual scene also vary. As the progression of the task is mostly fixed, the factor co-variate. For instance, the rotations in the centre also correspond to moments where landmarks appear and disappear of the FOV. During the passive returns along the outbound paths, the acceleration of the camera is fixed, so its variation will also correspond to the same time of landmark appearance, or position on the paths. Finally, the difference between leftward and rightward optic-flow could also be explained by the appearance of the landmarks in the FOV from one side of the screen to the other.

While the result that optic flow modulates parietal cells confirms previous knowledge on VIP functional properties (Bremmer *et al.* 2013; Chen *et al.* 2011; Bremmer *et al.* 2002a; Sunkara *et al.* 2016), the fact that it also modulated hippocampal cells is a new discovery. This could match the fact that hippocampal place cells are expressed only in moving animals, rather than static. However, we feel that the optic flow analysis needs behavioural control to be correctly interpreted in our task, due to the aforementioned confounds. This is why we will not include these analyses, because the task design doesn't allow to disentangle the OF factor from others. Nevertheless, we mentioned in the discussion that optic flow should also be a factor taken into account to explain cell activity, page 12, line 462:

“This suggests that parietal cells are recruited for specific types of sensorimotor patterns, such as oculomotor movements or self-motion cues, such as optic-flow stimulations, linked to specific navigational context.”

2.5. Page 7 and following: if I understand it correctly, the authors only considered the horizontal component of the eye movements for their analyses? What influence did the vertical components have (see e.g., Noel et al., eLife, 2022)?

We took into consideration vertical and horizontal components when studying saccades by themselves in the section 2 (saccades are defined by changes in eye velocity on both axes). We only selectively focused on the horizontal component in the section 3, when studying the gaze-position-related activity, because we chose to focus on the gazing of the elements of the environment, whose positions are discriminated by their horizontal position on screen rather than their vertical positions. This reduced considerably the number of variables included in the analysis. Further, studies show that during the visual exploration of virtual, naturalistic landscapes, the distribution of fixations follows the horizon of the scene (Bischof *et al.* 2020). Therefore, we chose to concentrate on the changes associated with the horizontal visual panorama.

2.6. Page 8 and more general: why did the authors not use a six-arm maze? That would have been compatible with the periodicity of grid cells. So, what was the benefit of 5 over 6 arms?

While it is true that grid cells display a hexa-directional modulation, to demonstrate such modulation, one needs to sample in an alignment to the 60° direction as well as in other directions. Indeed, in the rodents, grid cells' peaks are aligned when the rat travels in one direction, but the periodicity can only be evidenced if the cells are sampled in different alignments for which the activity is then lower, enabling to contrast the activity between the peaks and off peaks. Therefore, using a 5 arms starmaze is more suitable if one attempts to test for entorhinal grid cells. We computed a gridness score (Methods by Killian *et al.* 2012) for hippocampal cells and previously found that no cell reliably displayed grid-like activity above what would be expected by chance (data not published). Further, previous work showed that grid cells lose their 6-fold periodicity and are more path-equivalent when rodents are tested in alleys-like environments (Derdikman *et al.* 2009). Thus, to properly test for gridness, one should characterize the activity of cells in a 2D arena and possibly contrast their firing to that obtained in maze like environments to understand the transformation across the two geometries.

The choice of the 5-branches-starmaze actually came from practical reason, as we wanted enough activity per position. Previous work with virtual 2D environments (Furuya *et al.* 2014) showed that it is difficult to get the animal to explore a virtual open field as a rodent would do it, just by foraging. Further, when a monkey forages virtually, it then, quickly stops to pay attention to the environments and just chases visible objects. As we aimed to study the activity as animals used landmarks, we chose maze in which the animal needed to make landmark based decisions. The initial training of the animal was done using a Y maze which allowed to “physically” stop him at the choice point, allowing him to make movements to the right or the left. This geometry allowing to force a stop in the middle and a path decision was then enriched with two additional paths.

2.7. Page 11: how sure were the authors to not record from area MIP? Did the authors check for hand movement related neural activation?

We were attentive to remove any unit that would be anatomically ambiguous (or not well isolated) hence a rather limited number of neurons. MIP being located on the other bank of the IPS, it was easy to discriminate neurons as part of it or not, based on anatomical coordinates. Further, the activity was

monitored online during the lowering of the electrodes, and in principles, cells only included those which passed the sulcus.

Nevertheless, to test for the presence of motor related activation in our data, we analysed the activity of each cell at the time at which the animal made a right or left joystick move in the centre of the maze. We reasoned that if cells were recruited during left or right joystick push, then, this should be independent from the orientation of the animal (i.e. what the animal is facing as the animal will make 1 movement or a series of 2 movements in a row). First, we identified that 39/111 (35.1%) parietal and 57/142 (40.1%) hippocampal neurons displayed significant hand movement related activation (response window: -200 to +100ms relative to movement; baseline: -800 to -200ms and +100 to 800ms; activity threshold: +2.5SD). for the spatially-modulated neurons, these proportions were 29/91 (31.9%) and 19/64 (29.7%) for the parietal and hippocampal cells respectively. Next, we used a two-way-ANOVA, (with the direction of movement and the virtual orientation of the animal as fixed factors, with the orientation nested into the movement direction) to test for movement selectivity independently of head orientation. We showed that in total, only 6/111 parietal cells (5.41%) and 10/142 hippocampal cells (7.0%.) responded to joystick movements while being selective to its the right or left direction, independently of the orientation. When only considering the place-modulated cells, the proportions are similar, with 5/91 (5.5%) PPC cells and 5/64 (7.8%) HPC ones that met the criteria.

These are very low proportions of the population, and very close to the chosen statistical alpha level, notably in the PPC, which makes the probability of recording in the MIP unlikely.

We integrated to the manuscript a part of these new information about motor-related activity in the results section, page 5, line 184:

“Finally, we tested whether cells were modulated by left or right joystick moves, which could explain some of the asymmetries. Briefly, 29/91 (31.9%) parietal and 19/64 (29.7%) hippocampal spatially-modulated neurons displayed significant hand-movement-related activation (see Methods), but only 5/91 (5.5%) PPC and 5/64 (7.8%) were significantly selective to left or right movements independent from the orientation of the animal at the moment of the motion. This excluded the hypothesis of purely motor-driven neurons in both areas, rather suggesting that the cells were mainly modulated by sensory inputs.”

and the associated conclusion, page 5, line 199:

“In this context, we sought to determine whether these spatial selectivities were primarily attributed to bottom-up visual information or to the top-down systematic exploratory behaviour of the animals, rather than to motor events, with respect to the visual maze elements.”

2.8. Figure 2 and related to my previous comment: how sure are the authors they did not record from area 7a? Panel 1G shows that the peak of saccade related discharges of most parietal neurons occurred post-saccadically. Yet, following Barash et al., J Neurophysiol., 1991, on average the peak should occur pre-saccadically.

Indeed, neurons in area LIP tend to be active pre-saccadically, participating to the behavioural control of these eye movements (Barash *et al.* 1991b; Ben Hamed *et al.* 2001; Bendiksbj and Platt 2006; Bremmer *et al.* 2009; Leathers and Olson 2012; Mazzoni *et al.* 1996; Paré and Wurtz 1997; Platt and Glimcher 1999; Rao *et al.* 2012; Steenrod *et al.* 2013; Zhou *et al.* 2018). However, it has been demonstrated that some cells rather express trans- or post-saccadic responses, that can start up to 200ms after the saccade (Barash *et al.* 1991b; Ben Hamed *et al.* 2001; Munuera and Duhamel 2020; Zhou *et*

al. 2018). Another relevant argument is that in undirected free saccadic behaviour, it was recently shown that the response of the LIP neurons would be delayed (Johnston *et al.* 2022). This delay would result in fewer pre-saccadic neurons in our naturalistic setup, therefore, compatible with the response profile of the cells presented in Figure 2G.

In addition to these points, we would like to emphasize that we recorded from lateral VIP, in which, as in other visual areas, neurons can display saccadic-suppression, followed by post-saccadic enhancement (Bremmer *et al.* 2009). Further, such response has also been documented following the appearance or the motion of a stimulus into their RF (Colby and Duhamel 1991, Duhamel *et al.* 1997; Duhamel *et al.* 1998), as would normally occur when an object enters the RF following a saccade. In this framework, the proportion of post-saccadic parietal neurons, whose size is indeed larger than what expected from the LIP literature, could actually encompass units belonging to both LIP and VIP.

We added a small paragraph discussing these results in the main manuscript, page 14, line 556:

“The proportion of post-saccadic activity was higher than what is typically reported in the literature for LIP. Nevertheless, this is in line with recent work showing that saccade-related activity in LIP was diminished and displayed altered time dynamics in free-behaviour conditions (Johnston et al. 2022). Therefore, the peri-saccadic activity profile does not suffice as a criteria to discriminate LIP from VIP as VIP also displays post-saccadic enhancement (Bremmer et al. 2009) and response to stimuli appearing in their RF (Colby and Duhamel 1991; Duhamel et al. 1997; Duhamel et al. 1998).”

For more clarity, we also modify the population maps of Figure 2G-H, to separately indicate the negative or positive activity peaks of the suppressed, mixed or enhanced per-saccadic neurons. The corresponding results can be found in the manuscript page 6, line 235.

2.9. Figure 2, continued: I am worried by the average saccade length of 93 ms. I doubt this can be correct. Such a large value would imply average saccade sizes of 30 - 40 degrees or even more, which I consider unlikely.

We apologized for omitting the detail of the methods in the previous version of the manuscript. As mentioned above, following the reviewer’s concern, we reanalysed the data and recomputed saccades on a newly processed eye data. We now detail that examination of the eye traces prompted us to first apply a smoothing on the raw eye-tracking data, in order to correct a few inconsistencies generated both by the digital-to-analog conversion inside the eye tracker and its later digital sampling at a higher frequency. Then we averaged the signal of the left and right eyes, properly dealing with pupil signal loss when it was limited to a single eye. Next, to identify the saccades starts and ends, we applied a fixed speed threshold-based method (limit of 50°/sec), which is more conservative than the adaptive acceleration threshold we previously employed. As a results of these modifications, we corrected the values of saccades metrics. We also modified the method to assess saccades-responsive cells, by discriminating activity as a function of right or left saccade directions (Katz *et al.* 2022).

The figure below shows example horizontal and vertical eye traces during the task, with the detected saccades (beginning indicated by blue lines, ends by red lines, duration indicated on top, in blue). These figures are now presented in Figure S2A. As a conclusion, and in light of our examination of eye traces, we conclude that our method is accurate and indeed, there appears to be large saccades in our task.

Figure R2.9. Three example samples of 3-second horizontal (left, top) and vertical (left, bottom) eye-traces. The blue and orange vertical lines respectively mark the start and the end of the saccades detected by our algorithm. For each of them, their duration is indicated on the top of the left panel, in blue. The right panels represent the same eye-traces, plotted in two dimensions.

We examined saccade durations and frequency as a function of the maze segment in which the animal was located. The frequency of saccades significantly varied with segment (Kruskal-Wallis: $X^2_{3df}=59.58$,

$p=7.23\times 10^{-13}$): it was 2.51 ± 0.11 saccade/sec in the inbound paths, 3.19 ± 0.20 in the centre, 2.64 ± 0.10 in the reward path, and 2.40 ± 0.11 in the outbound paths, for a total average of 2.68 ± 0.076 saccades/sec. Their duration also varied with segments ($X^2_{3df}=40.78$, $p=7.27\times 10^{-9}$): it was 76.45 ± 1.31 ms in the inbound paths, 77.46 ± 1.69 in the centre, 74.40 ± 1.37 in the reward path, and 82.99 ± 2.02 in the outbound paths, for a total average of 77.83 ± 0.89 ms.

For the reviewer’s appreciation, we provide a table of these measures for each monkey separately. While the pattern differed slightly across monkeys, the saccade durations were the longest in the outbound paths for both animals, suggesting that animals made large eye movement when being passively allocated from the reward position to a new start. The significant variation of saccades frequency (Monkey K: $X^2_{3df}=65.14$, $p=4.69\times 10^{-14}$, Monkey S: $X^2_{3df}=48.17$, $p=1.96\times 10^{-10}$; see Figure 2A, bottom panel) and duration (Monkey K: $X^2_{3df}=36.76$, $p=5.16\times 10^{-8}$, Monkey S: $X^2_{3df}=37.29$, $p=4.00\times 10^{-8}$; see Figure 2B, bottom panel) across maze segments held true in each animal, taken individually.

		Monkey K	Monkey S	Total
Frequency (sacc/s)	inbounds	2.38±0.12	2.74±0.18	2.51±0.11
	center	2.76±0.13	3.98±0.34	3.19±0.20
	rewarded	2.79±0.11	2.36±0.16	2.64±0.10
	outbounds	2.22±0.094	2.73±0.20	2.40±0.11
	Total	2.54±0.066	2.95±0.17	2.68±0.076
Duration (ms)	inbounds	78.49±1.58	72.70±1.49	76.45±1.31
	center	80.86±1.81	71.22±1.64	77.46±1.69
	rewarded	75.28±2.01	72.78±0.98	74.40±1.37
	outbounds	85.06±2.80	79.19±1.80	82.99±2.02
	Total	79.92±1.16	73.97±0.96	77.83±0.89

Table R2.9. Contingency table indicating the average frequency (in saccades/sec) and duration (in ms) of saccades depending on the maze segments where they were performed (rows) and on the animal’s identity (columns).

The description of the saccade rates is actualized in the manuscript, page 6, line 205.

The fact that averaged saccades duration were higher than what is usually described in the literature (40-50ms; Rayner 2009; Rayner and Pollatsek 1992) can be explained by the naturalistic task we used, during which, animals performed unrestricted eye movements, and extracted spatial information from their 74-degrees-large visual environment. This is unlike previous studies in which saccades were performed in a simple and controlled stimulus-response tasks (Fuchs 1967; Bahill *et al.* 1975; Dick *et al.* 2004; Garbutt *et al.* 2001). Previous works evidenced that the dynamics of oculomotor behaviour were significantly modified during the visual exploration of a virtual environment compared with simple oculomotor tasks, with notably slightly slower saccades in the former case (Corrigan *et al.* 2017). Thus, in our settings, animals tended to make large saccades, scanning the screen from one extremity to the other, especially during the motions along the outbound paths, when landmarks were appearing on the screen edge (as visible on Figure 2B, bottom panel, showing higher durations during outbounds). Fuchs (1967) described a duration of about 60 ms for a 38-degrees-large saccade in monkeys (his Fig. 4), and more recently, van Opstal and Kasap (2019) showed main sequence data (their Fig. 2) in which saccades of 40° executed with no head movement lasted ~75 ms. Additionally, Corrigan *et al.* (2017) showed that monkeys made slower saccades during exploration compared to ones directed to rewarded targets, and

Zhang *et al.* (2022) evidenced that, if visually-guided saccades of about 10° reached a duration of 60ms, memory-guided ones (as it can be the case in our task, when landmarks appearances are anticipated) could reach 100ms for some animals (see their Fig. 5). Thus, given the other arguments listed above, the durations we report seem in line with the literature on the monkey. We added a sentence page 6 line 206 to describe what account to unusually large saccades:

*“The long duration of saccades can be explained by the structural layout of the task maze, in which animals freely moved eyes from one part of the screen the other, as elements (landmarks, paths) of the environment entered or exited the field of view (see also Corrigan *et al.* 2017; Zhang *et al.* 2022).”*

The rest of the analyses focusing on the saccade-related properties of the neurons, were also updated, on pages 6-8, accounting for the updates saccade detection method. The conclusions of the results remained unchanged, except for the points discussed further below:

Notably, in order to enable comparison with previous work, we chose to improve the detection of saccades-responsive cells. It is frequent that parietal (Platt and Glimcher 1999; Gnadt and Andersen 1988; Heiser and Colby 2006; Hagan *et al.* 2012; Barash *et al.* 1991a), but also HPC (Katz *et al.* 2022) cells respond preferentially to one saccade direction. Therefore, we computed the cells' responses to left- *versus* right-directed saccades, and based the data analysis on the cells' the strongest response (i.e. highest amplitude). The description of these new results is on page 6, lines 223 and following, and the updated Methods is on page 37 line 1292 (section *“Neuronal activity aligned on saccades ends (perisaccadic activity)”*). Briefly, with this method, the percentage of cells detected as modulated by saccades in the parietal and in the hippocampus increased.

We also realized that it would make sense to compute the correlations between the neural maps and the saccades maps only for the saccade-responsive cells, that are the cells whose spatial activity could be explained by the oculo-motor dynamics. Thus, we recomputed this analysis, and integrated it page 7, line 247:

“We found that, out of the saccade-responsive cells, 17/73 (23.3%) of PPC cells and 13/45 (28.9%) of HPC ones showed a significant positive or negative correlation between their neural and saccades maps (Figure 2I; permutation test, $p \leq 0.05$; see Methods). Overall, the absolute values of correlation coefficients were not different between the PPC ($\mu|r|=0.11 \pm 0.020$) and the HPC ($\mu|r|=0.11 \pm 0.0026$; Wilcoxon rank sum: $|Z|=0.31$, $p=0.76$), showing that hippocampal saccade-responsive cells tended to covariate with saccadic activity as much as parietal ones. In sum, the behavioural saccade rate may explain the position-related neural activity, but only for a modest fraction of cells, from both areas.”

The conclusions held true, showing that a minority of cells displayed significant correlations, in PPC and HPC. The only difference is that we show now that the neural activity of the HPC-saccade-responsive cells co-variated with the animals' saccades rates just as much as that of the PPC.

When investigating whether the segments for which the 73 saccade-responsive cells' peri-saccadic activity was the higher were uniformly distributed across the maze (i.e. inbounds, centre, rewarded path, outbounds), we now found that there was significantly less cells preferring the outbounds in the PPC. The proportions were nevertheless similar between the 3 other segments for the PPC, and between the 4 segments for the HPC. The new results are actualized page 7, line 274:

“While, in PPC, fewer cells responded in the outbound paths compared to the other segments (5/53, 9.4%; Chi-squared goodness-of-fit on a uniform theoretical distribution: $X^2_{3,df}=7.91$, $p=0.048$), they distributed

equally between inbound (13, 24.5%), centre (18, 34.0%) and reward paths (17, 32.1%; $X^2_{3df}=0.88$, $p=0.65$; Figure S2R). In the HPC, the four maze segments were homogeneously distributed among the cells, with 9/26 (34.6%) cells more active in inbound paths, 5 (19.2%) in centre, 4 (15.4%) in rewarded path and 8 (30.8%) in outbound paths ($X^2_{3df}=2.62$, $p=0.45$). Thus, cells in both areas responded differently depending on the ongoing task context. In PPC, general attentional or motivational processes might have had an impact on the whole population, notably because the passive motion along the outbound paths may require less cognitive resources for the animal. However, the homogenous repartition among the rest of the maze suggests that each neuron responds to a specific sensori-motor context.”

Finally, following the reprocessing of the eye data for saccades, we took the opportunity to re-run our analyses on gaze-related activity patterns (Results section 3, pages 8-10), and also reorganized the section for more clarity. In that process, we noticed that the computation of the p-values of the permutation tests were erroneously too stringent. Correcting the p-values resulted in a higher number of cells with an activity significantly correlated with gazing at landmarks or path (Figure 3N; manuscript page 9, line 323). The reprocessing of the eye data also resulted in minor changes with respect to the previous version of the manuscript, such that the value of the correlations between the activity as a function of the point of gaze and the sinewave changed marginally for some of the cells. Therefore, the figures changed slightly, but the results conclusion did not. We also decided that the comparison of the firing rates of the neurons between areas and cell types should be done using their peak activity, rather than their mean activity. The conclusions remained identical, and the data were actualized page 9, line 338:

“When comparing firing rates, a two-way-ANOVA on the neurons’ peak activity revealed a significant effect of the neural area ($F_{1df}=15.5$, $p=1.08\times 10^{-4}$) and of the cell “type” factors (i.e. landmark or path cells; $F_{1df}=4.72$, $p=0.031$), and that those two factors tended to interact ($F_{1df}=2.97$, $p=0.086$). Precisely, parietal cells ($\mu_{peak}=9.70\pm 2.39$ sp/sec) fired more than HPC ones ($\mu_{peak}=4.73\pm 0.97$ sp/sec) and landmark cells ($\mu_{peak}=8.09\pm 2.11$ sp/sec) fired more than path ones ($\mu_{peak}=5.74\pm 1.10$ sp/sec) and the effect tended to be amplified for the parietal landmark neurons (see Table 3).”

as well as in the associated Table 3.

The only significant changes resulting from the reprocessing of the eye data and the calculation of the point of gaze were, firstly, that in the HPC only, more cells responded significantly to the gazing of landmarks compared to paths (see results page 8, line 326, and associate conclusion page 9, line 344), and secondly changes regarding the analysis of the offsets between point of gaze and neural activity. Indeed, using the previous point of gaze calculation, we had described that activity of hippocampal landmark cells as a function of the point of gaze had an offset compared to point of gaze (as if cells were more active when the animal looked at the edge of the landmark in hippocampus only). This strange effect became marginally significant when we reconducted the analysis with the newly computed eye data. Therefore, we removed the corresponding plots, which allows to make Figure 3 less dense.

The analysis accounting for the differences between the PPC and HPC landmarks and paths cells’ place maps was also updated for more clarity, and in response to reviewer 4, point 1.

Minor concerns

2.9. Page 5 and following: I appreciate the level of detail of statistical analyses. Yet, incorporating all the statistics in the main text hinders reading. I suggest to somehow condense the statistics provided in the main text and provide some supplementary information with all the results displayed in detail.

We agree that the statistics hinder the readability of the manuscript, but we followed the journal's convention. For some of the new results, we incorporated statistics in a table (see Table 1 & Table 2).

2.10. Page 13, first paragraph: I find the statement a bit weird that the authors could not differentiate between LIP and VIP based on their activity profile. Cells in these two areas have remarkably distinct response features.

In line with the reviewer, and upon initial assessment of the literature, we also expected the division between LIP and VIP to be evident based on their respective functional properties. However, the majority of the literature describing the functional differences between LIP and VIP neurons relied upon simple behavioural tasks, designed to elicit and isolate a single stimulation or a single behaviour. Typically, monkeys were presented with flashed or moving light bars or dots patterns on a dark screen, and/or performed a single saccade from a fixation point to a target (Barash *et al.* 1991b; Duhamel *et al.* 1992; Ben Hamed *et al.* 2001; Duhamel *et al.* 1998; Bremmer *et al.* 2002a; Sunkara *et al.* 2016). Notably, a memorized-delayed-saccade task would confidently identify LIP neurons, and a passive exposition to random dot stimuli simulating optic-flow in different directions would evidence VIP neurons. Unfortunately, these tests were not systematically performed for each of the recording cells. We also want to point out that such controlled test only allows to positively label LIP cells that fit the criteria but may not enable to characterize others that display other properties and are still in the region. Indeed, the literature describes that while a wide range of 17% (Gnadt and Andersen 1988) to 88% (Mazzoni *et al.* 1996) of the cells display response during the delay-period (Ben Hamed *et al.* 2001; Gnadt and Andersen 1988; Barash *et al.* 1991b; Ipata *et al.* 2009; Mazzoni *et al.* 1996; Paré and Wurtz 1997), the rest of the population is left unlabelled.

In contrast, the task used in the current study was designed in order to approximate naturalistic conditions as closely as possible. This came with the caveat that the response dynamics of the neurons belonging to the two areas may appear as overlapping, making their discrimination challenging. Specifically, as previously developed in point 2.8, the suppressive pre- and trans-saccadic response and the excitatory post-saccadic activity displayed by LIP neurons, can be mixed up with VIP cells showing saccadic-suppression and post-saccadic enhancement in response to the appearance or the motion of a stimuli brought into their RF. In addition, recent works evidenced that the encoding of saccade direction by the LIP neurons, and more generally the global saccade-related activity of the area, were diminished and displayed altered time dynamics in free-behaviour conditions (Johnston *et al.* 2022). This emphasizes the difficulty to transfer the knowledge of what considered as the “classical” functional properties of the cells, defined in constrained tasks, toward naturalistic context, and may be relevant to the interpretation of our results.

Moreover, it has been evidenced that LIP neurons adapt their response to the task the animal has been trained to, probably in link with the attentional and motivational processes. For instance, despite the area being largely recognized as visual, LIP neurons can start developing an encoding of auditory stimuli direction in specific tasks (Grunewald *et al.* 1999). Other studies evidenced that LIP neurons displayed a preference for stimuli categorized as moving toward a direction or another, whether saccades were involved in the task (Swaminathan *et al.* 2013) or not (Rao *et al.* 2012). These examples demonstrate how LIP can flexibly exploit the signals it receives from various modalities, as soon as they become relevant for the behavioural control of saccades or for the success of the task. Another study even evidenced how LIP neurons were selective to stimuli motion direction during a passive, non-rewarded fixation task (Fanini and Assad 2009). All of these properties make the discrimination between neurons belonging to this area and other surrounding ones such as VIP difficult in stimuli-rich goal-directed tasks.

We acknowledge this difficulty in the discussion, from page 14, line 562:

“Hence, we could not clearly separate LIP from the VIP along the depth of the sulcus using cell’s activity profiles (Figure S2O). This indicates that the rich visual stimulation of the task recruited cells in both regions, with properties either functionally overlapping or expressed for co-occurring features.”

2.11. Figure legend on Page 25 and Methods: the authors should provide the size of the projection screen also in degrees of visual angle.

This has been corrected and added to the Figure legend, page 26, line 968, as well as in Methods section, page 33, lines 1117.

2.12. Page 35: I consider a Gaussian Kernel with SD=100ms to be extremely large. The authors must comment on this selection.

We apologize for the lack of clarity. The temporal smoothing was chosen to suit HPC cells, which have often low firing rates (< 10 Hz), and for which we adopted a temporal binning of 100 ms per position bin. This spike rate series was then smoothed with a Gaussian kernel of width 10 ms for visualization purposes only. For consistency, and to allow a fair comparison, the same method was applied to PPC cells. We agree that in the latter case, better resolved maps could have been obtained with a narrower kernel.

We clarified this in the methods page 35, line 1196.

2.13. Page 36: I did not understand “within a distance of 5.6 units”.

Distances between spatial fields were computed in “virtual meter unit” coordinates which correspond to 1 bin = 1.6 units. We divided the inbound paths in 10 bins (16 units), short returns were 12-bins-long (19.2 units), and long outbounds were 19-bins-long (30.4 units). We revised our method to identify the place fields of the cells, axing it on the continuity of the field rather than on the distance separating the responsive spatial bins. Spatial fields are now defined as any contiguous bins whose neuronal activity was above the threshold of mean + 2SD of the activity over the whole maze.

These changes are implemented in the manuscript page 36, line 1248 and a detailed response relative to the description of place fields is also provided in response to reviewer #1 point 2.

2.14. Page 38: from the anatomical sections I was wondering about the recording approach. Was the electrode track parallel to the sulcus. Or did the authors insert the electrodes vertically in stereotactic coordinates?

The rectangular chamber was implanted in orthogonal stereotaxic coordinates to facilitate recording from both the intraparietal sulcus and the hippocampus. This appeared to be aligned with the sulcus for monkey K only.

We now clarify this in the Method section of the manuscript, page 34 line 1165:

“For a period of approximately 6 months, each animal underwent daily recording sessions, during which laminar U-probes (Plexon®) were lowered to the target areas along the orthogonal stereotaxic axis (see Figure S1A-B for PPC, and Wirth et al. 2017 for the HPC), both located in the animal’s right hemispheres.”

2.15. Page 38: I guess the reference is somehow corrupted “determined by Fast Fourier transform (M.Sc. Eng. Hristo Zhivomirov, 2014).

Thank you for noticing this mistake. We deleted the reference and now specify that we used a Fast Fourier transform in the method section (see page 39, line 1360):

“The modulation coefficient was here computed as the magnitude of the signal of the smoothed gaze-related activity at the frequency of the reference sine-wave, divided by the sum of the magnitudes at all frequencies found in the smoothed gaze-related activity, determined by Fast Fourier transform.”

2.16. Page 40: are the authors sure about the labels of their anatomical sections? The form of the IPS at “AP 1” looks more posterior to me than the shape at “AP -1”. Are all panels shown at the same scale? Compared to the other panels, the IPS at “AP 8” looks rather deep to me, at least deeper than to be expected at this rather anterior position. And in relation to one of my above comments, a few recording sites shown in this panel (AP 8) could be in MIP rather than LIP/VIP.

Thank you for noticing this error. We apologize for the inversion of the labels and made the appropriate corrections on Figure S1. Panels are on the same scale and now labelled from anterior to posterior, with positive numbers designating what is anterior (frontal) from inter-aural line, and negative numbers what is posterior from inter-aural line. As pointed before, the properties of the neurones with respect to hand movement, are not compatible with MIP.

Reviewer #3 (Remarks to the Author):

The present paper examined neural activity in the posterior parietal cortex and hippocampus during two macaque monkeys performed vision-based navigation in a virtual maze. The authors compared position-related activity, saccade-related activity, gaze-related activity and landmark-appearance activity between the two areas. While both areas substantially show task-related activity, the activity in the parietal cortex was related to anticipation of reward or necessity of choice compared with the hippocampus. It is very timely to investigate neural process of the primate hippocampus and parietal cortex in terms of “place” and “view.” The present paper challenged an important question in cognitive neuroscience including both perception and memory particularly for navigation. I agree this study is valuable, but I found several unclear points that should be addressed as following.

We thank the reviewer for their enthusiasm for the manuscript. We clarified methods with respect to analysis related to the modulation of neurons by to position, saccades and landmark appearance. We also would like to refer the reviewer to responses to reviewer #1 and #4, with respect to the interpretation of position related activity. With respect to the interpretation of the findings, we clarified that attention processes can be intrinsically linked to visual exploration behaviour. We adapted the results and discussion to account for this.

3.1. Authors examined only correct trials for analyses of neural activity and saccades. Because this procedure made asymmetry between the reward path (centre to peripheral) and other four inbound paths (peripheral to centre) in animals’ virtual movement, this point (the usage of only correct trials for analysis) should be noted explicitly in Result Section. Authors should also display the asymmetry of data structure clearly. I may suggest adding arrows indicating direction of movements to Figs. 1B and 2A.

The indication that only correct trials were used for analysis was indeed only given in the Methods section (page 35, line 1182), and was omitted from the Results section. This point has been corrected (page 3, line 105; see also response R4.5):

“For this study, only the correct trials were selected for analyses. In this way, the path chosen by the animal was always the reward path, oriented toward the North on our maps.”

While error trials would be extremely interesting to analyse, there were not enough trials to exploit for proper analysis.

Also, thank you for the suggestion about the arrows for movement direction. We have now added them on the Figure 1B and 2A.

3.2. “We first examined whether parietal or hippocampal cells were modulated by animal’s virtual position, i.e. the camera’s location within the environment, independent from its orientation” (Page 3): What does the phrase “independent from its orientation” mean? The animal’s head direction in the virtual space did not change at each position in the inbound paths and the reward path (except for centre).

You are right, position and orientation were co-depending variable in this task, apart from the centre of the maze, when considering only correct trials. To avoid confusion, we removed “independent from its orientation”, page 3, line 108.

3.3. Please show the behavioral data related with saccade for each animal (Figs. 2A and 2B).

Example traces are shown in response to reviewer #2, point 9. We now included these eye traces in the supplementary material (Figure S2A). Further, we now document the saccades rates and durations data for each animal separately on Figure 2A-B, also described in response 2.9.

Figure 3.3. Scatter plots of the saccade rates (top) and durations (bottom) of Monkey K (N=46 sessions; circles) and S (N=25; triangles), for each session (total N=71), depending on the animals' position in the maze. The saccades rates and durations were significantly different between the 4 maze segments in each monkey (rates: Monkey K: Kruskal-Wallis: $X^2_{3df}=65.14$, $p=4.69 \times 10^{-14}$, Monkey S: $X^2_{3df}=48.17$, $p=1.96 \times 10^{-10}$; durations: Monkey K: $X^2_{3df}=36.76$, $p=5.16 \times 10^{-8}$, Monkey S: $X^2_{3df}=37.29$, $p=4.00 \times 10^{-8}$).

We changed the text to describe the patterns across both monkeys, on page 6, line 213, and the associate Table 2:

“This inhomogeneity was supported by a significant difference in saccade frequency in each monkey (see Figure 2B, top panel; Monkey K: Kruskal-Wallis: $X^2_{3df}=65.14$, $p=4.69 \times 10^{-14}$, Monkey S: $X^2_{3df}=48.17$, $p=1.96 \times 10^{-10}$) and duration (see Figure 2B, bottom panel; Monkey K: $X^2_{3df}=36.76$, $p=5.16 \times 10^{-8}$, Monkey S: $X^2_{3df}=37.29$, $p=4.00 \times 10^{-8}$) between the four portions of the maze (see also Table 2).”

3.4. “the absolute values of correlation coefficients were significantly higher in PPC ($\mu|r|=0.11 \pm 0.015$) than in HPC ($\mu|r|=0.095 \pm 0.012$; tailed-Wilcoxon rank sum: $Z=1.65$, $p=0.049$), showing that parietal cells tended to be more positively or negatively saccade-responsive than

HPC ones.” (Page 5): The correlation between virtual position neural map and saccade frequency map can be modulated by common factors such as an attention effect. It is important to remove those confounding factors from the analysis if the authors want to suggest “saccade-responsive” beyond the co-variation.

Thank you for the comment. We agree that the term “saccade-responsive” may be an over-interpretation of simple covariation of saccades map with neural map. As we could not characterize saccade in the dark, or in absence of any contextual information, in the manuscript, we replaced “*cells tended to be [...] positively or negatively saccade-responsive*” by “*saccade-responsive cells tended to covariate with saccadic activity*”, on page 6, line 252.

3.5. “we first determined which positions corresponded to the entrance of a landmark in the FOV, on average (the different objects could slightly vary in size). Then, the appearance times were defined as the timestamps at which the camera was at those positions.” (Page 39): Does the “entrance” indicate appearing of the first pixel, centre or whole of a landmark (on average)? How long did it take for the appearance of a landmark from its first pixel to its entire shape?

We considered as “entrance” the moment at the first pixel of the landmark appeared on the side of the FOV. The total duration of appearance of a landmark depended on the location of the animal in the maze. In the centre, it depended on the time the monkey took to push the joystick to initiate the rotation. The landmark never fully appeared all at once, since when the camera was aligned on a path, only the half of two landmarks were visible on the right and the left of the screen. During returns along the short outbound paths, the first landmark (that would be the 2 North ones) was taking approximately 0.6 sec to fully appear. The second landmark (the 2 South ones) was taking about 0.3 sec to appear. Finally, during the long returns, the landmarks (the 2 North ones) appeared in about 0.8 sec.

We added these values in the Method section page 40, line 1398:

“For information, the average duration took by a landmark to fully appear on screen was the following, depending on the position of the animal in the maze: on the short outbound paths, the first landmark to enter the FOV (that could be the 2 North ones) appeared in ~0.6, and the second one (the 2 South ones) did in ~0.3 sec; on the long outbound paths, the landmarks (the 2 North ones) appeared in ~0.8 sec. In the maze centre, the duration of appearance depended on how long the monkey took to push the joystick to initiate the rotation: the landmark never fully appeared at once, since when the camera was aligned on a path, only half of two landmarks were visible on the right and the left of the screen.”

We also added a few lines describing the method used to compute and test the variations of the population, page 40, line 1408, as it was missing:

“After that, to create the population activity, each cells’ raw mean activity was at first convoluted with a Gaussian kernel (SD=40ms), then normalized using the Min-max normalization, before being altogether averaged. The population activity was considered as significantly responsive to landmarks appearance for each time-bin of the response window (-200 to +300ms relative to appearance) whose value was superior to the mean + 2.5SD of the baseline (-800 to -200ms and +300 to +800ms).”

3.6. “More parietal neurons were modulated by animal position in the virtual maze than hippocampal ones” (Page 10): Because the present experimental and analytical conditions contain

several confounding factors (visual stimuli, active vs passive, reward), the above sentence might be misleading as a summary / conclusion.

In addition, in addressing comments of reviewer #1.2 and #4.2, we substantially remodelled the first paragraph in order to clarify that the position-related activity cannot be directly related to rodent place cells. These changes can be found:

- p.4 line 134: *“Because the spatial properties of the recorded neurons may echo rodent place cells (Aronov and Tank 2014; O’Keefe 1979), we conducted additional analyses in order to comprehend to which extent they resemble or differ from place cells.”*
- p.4 line 158: *“The results suggest that the spatially-modulated PPC or HPC cells were not akin to classic rodent place cells, which encode the animal’s position per se and are invariant to head orientation (Andersen et al. 2006, but see Acharya et al. 2016; Jercog et al. 2019; O’Keefe and Dostrovsky 1971; Rubin et al. 2014). Rather, many cells displayed brief and transient spatial modulation and selectivity to orientation in addition to position, which may be driven by other variables than position itself such as direction or view.”*
- p.5 line 194: *“These spatially-modulated neurons in the HPC and the PPC displayed distinct properties than these of the classical place cells.”*

3.7. Was there any difference in response patterns among recording sites in the hippocampus (e.g., CA1 - CA3, anterior-posterior)?

We do not feel we can reliably separate the two fields anatomically with the resolution given by the MRIs. However, when taking recordings that could only belong to either field because they were distant enough from the medio-lateral midline, we did not find any significant difference. These null findings are not reported in the manuscript.

3.8. “Taken together, these findings shed light on the neural processes that link place and view.” (Page 1): Please explain this sentence more by specifying the linking between place and view in the present study.

Indeed, “Places” are defined by a combination of factors, mostly deriving from the current, past, and future field of view. We now clarify this in the abstract, line 32:

“Taken together, these findings highlight the neural processes that make up place, combining active visual exploration of objects in space with memory-driven actions.”

Reviewer #4 (Remarks to the Author):

Visual information is key to spatial navigation in humans and non-human primates. The aim of this study is to better understand the role of the posterior parietal cortex (PPC), an important area for high-level visual information processing in human and non-human primates, in goal-directed navigation. Vericel et al. used tetrode recordings in the intraparietal sulcus of the PPC and in the hippocampus while head-fixed monkeys navigated a virtual environment (5-arm star maze) to find uncued locations associated with rewards. These recordings were combined with video monitoring of eye movements to analyze saccades and fixations in particular. They found that the activity of the majority of PPC cells was modulated by position, but in a stereotyped way according to the ongoing action. The activity of a significant proportion of HP cells was also modulated by position in a more distributed way (several place fields per cell). As virtual navigation triggered a high rate of saccades to probe relevant landmarks in the environment (probably increased by the fact that the animals were head-fixed), the authors then analyzed the modulation of neuronal activity by saccades. The activity of the majority of PPC cells and a significant proportion of HP cells was modulated (increased or decreased) around saccades. As the saccade rate was variable in different locations, the authors then analyzed the spatial distribution of saccades (saccade spatial map) and how it correlated with neuronal activity maps. The two spatial distributions were correlated only for a minority of cells, so that saccade-induced modulation of activity alone cannot explain the modulation of PPC and HP cells by position. Neurons in the PPC were highly sensitive to the fixation of visual landmarks compared to paths, especially for landmarks close to the reward, and were active before the appearance of the landmark, suggesting that they have access to a spatial map of the environment. In conclusion, both PPC and HC cells show some modulation by position, but PPC activity is more likely to reflect a task-relevant representation of visual space.

Overall, this manuscript is a technical tour de force. The authors were able to train monkeys to perform a goal-directed navigation task in virtual reality and record activity in two key regions for spatial navigation (hippocampus and posterior parietal cortex) together with eye movements (to assess saccades and fixation). Importantly, the authors report convergent and divergent coding schemes in these two areas. Most interestingly, they observed a strong modulation of hippocampal activity by saccades, classically associated to PPC cells, and conversely, PPC activity shows some anticipatory activities usually associated with hippocampal spatial mapping. Altogether the results suggest a cooperation between the two structures allowing visually guided goal directed navigation.

We thank the reviewer for their appreciation of the work required to gather the data and the scientific advance we made. In response to the important questions and suggestions, we provided many additional analyses that help refine and interpret the data.

Major concerns:

4.1. In rodents, the modulation of the firing rate of hippocampal cells by position is often estimated over several trials. To be considered a place cell, a pyramidal neuron must discharge reliably at the same position over time. On average, how many trials were used to assess spatial selectivity? Could you show the trial-by-trial modulation of hippocampal and PPC cell activity as in Figure 2 C and D for saccades? How consistent was the spatial modulation across trials (spatial stability)?

We now provide additional details and invite the reviewer to read our response to reviewer #1.2, which provides detailed information on place fields and their stability as a function of the place field sizes and number of fields.

On average, 87.83 ± 3.97 correct trials were used for analysis for each cell which enabled us to assess spatial selectivity with about 20 laps for each inbounds and out-bound and 80 for the reward path. We now include position rasters for the cells in Figure 1 (see Figure R4.1 below and Supplementary Figures S1E-F). In this figure presented below, to provide the relevant maze segment for the raster, for each cell, we determined the location of its peak FR, and aligned the activity as a function of the animal's virtual position in the corresponding maze segment. In many cells, the spatial modulation within a segment can be appreciated by visual inspection.

Figure R4.1A. Raster plot (top) and average activity (bottom) of the 6 PPC example neurons presented in Figure 1B-G, aligned on their preferred segment start (black solid line). The standard errors of the mean are indicated in light colour. The identity of the preferred segment is indicated on top of each plot.

Figure R4.1B. Same as in Figure R4.1A, for the 6 HPC example cells, corresponding to the neurons in Figure 1H-M.

We added the corresponding information in the Methods section, page 37, line 1262:

“Neuronal activity aligned on position. To create the position rasters, we first determined on the neural place map which maze segment contained the peak activity of the neuron. As the time spent by the animal in the centre and the outbound arms could greatly vary across trials, depending on the identity of the starting positions, we only computed the raster on this segment where the cell was most active. In addition, trials for which the animal took a break in the middle of a segment were discarded. The peri-event spike histograms were computed by aligning the activity of the neuron on the time of the segment start, with a time-resolution of 1ms. To trace the mean activity, we first averaged the spike counts for all the trials to obtain a raw mean segment-start-relative activity, and finally smoothed it using a Gaussian kernel ($SD=40ms$).”

Then, we analysed the stability of the position-dependant FR pattern of the cells, i.e. of their spatial modulation, as a function of number of fields, field size and firing rate peaks (see response to reviewer #1.2, partially reproduced below).

To assess the stability of the cells’ spatial modulation through time, we computed a cross-correlation of the mean neural place maps between the first and second half of the session, testing the significance of the correlation against correlation computed on 999 spike permutation surrogates. We found that 51/91 (56.0%; $\mu r=0.23\pm 0.040$) of the PPC cells and 26/64 (40.6%; $\mu r=0.16\pm 0.049$) of the HPC cells with a significant information content for position presented a correlation coefficient between first and second half above that expected by chance. The spatial modulation was more stable in the PPC than in the HPC (Wilcoxon rank sum: $Z=2.12$, $p=0.034$). In both regions, the neurons with a lower number of place fields were more stable through time than the ones with multiple fields (Pearson correlation: $r=-0.30$, $p=0.0043$ for PPC, and $r=-0.46$, $p=2.44\times 10^{-4}$ for the HPC, Figure R1.2A). Moreover, the cells with a larger receptive field (Figure R1. 2.B) and a higher peak rate (Figure R1.2C), were also more stable.

Taken together, the results suggest that these spatially modulated cells are not akin to classic rodent place cells which encode the animal's position. Rather many cells displayed volatile brief spatial modulation which may be driven by other variables than encoding position *per se*.

We now modified the text to account for these results (page 4, lines 137):

“First, 51/91 (56.0%) of PPC position-modulated cells and 26/64 (40.6%) of HPC ones displayed a stable pattern of spatial modulation across the first and second half of the recording session (see Methods). The spatial stability of the example cells of the Figure 1B-M can be appreciated on the corresponding position rasters presented in Figure S1E-F. These rasters were created by aligning the neurons' activity as a function of the animal's virtual position in the maze segment containing its peak FR (see Methods). Stability of the spatial modulation was related to the number of fields and to their size: cells with fewer and larger fields were more stable through time than cells with numerous and smaller fields (Pearson correlation between spatial correlation coefficient and number of SFs: $r_{PPC}=-0.30$, $p_{PPC}=0.0043$; $r_{HPC}=-0.46$, $p_{HPC}=2.44 \times 10^{-4}$; Figure S1G; correlation with SFs size: $r_{PPC}=0.51$, $p_{PPC}=2.81 \times 10^{-7}$; $r_{HPC}=0.65$, $p_{HPC}=1.43 \times 10^{-8}$; Figure S1H). Further, neurons with higher peaks rates were more stable than others, in both PPC ($r=0.38$, $p=2.13 \times 10^{-4}$) and HPC ($r=0.41$, $p=8.12 \times 10^{-4}$, Figure S1I). These results suggest that, in addition to cells presenting a stable modulation, many displayed brief local modulation unlike rodent place cells. Next, we examined whether position-modulated cells were sensitive to head orientation. This analysis could only be performed in the centre of the maze, since in other locations, place and direction covaried (see Methods).”

We also modified the discussion to account what the interpretation of place encoding in our set-up, page 12 line 475:

“We suggest that the apparent parietal position code can result from visual processing of relevant cues, contributing to a task-relevant representation of visual space, rather than pure place code for self-position in Euclidian coordinates. Further, we also extend the latter interpretation for the position selectivity found in hippocampal cells in this task. Indeed, unlike rodents place cells (Andersen et al. 2006, but see Acharya et al. 2016; Jercog et al. 2019; O'Keefe and Dostrovsky 1971; Rubin et al. 2014), cells we recorded displayed small and unstable spatial fields and orientation selectivity in the centre of the maze, compatible with an active task-dependent visual processing of the scene.”

We also would like to bring to the attention of the reviewer that we included in the supplementary figures (Figure S3B-C) a time-aligned population activity as a function of the different segments for the path and landmark gazing cells. The supplementary analysis allows to quantify the differential recruitment of parietal and hippocampal cells as a function of the position of the animal in the maze, and allows for a much higher resolution than the previous analysis, that used the position bins. The new results are described page 9, line 359:

“We finally assessed how the populations of landmarks and path cells differed across brain regions at a finer scale. To this end, we compared their average activities within segments: the inbound path, the centre and the reward/outbounds paths. When aligned to the time the monkey reached the centre of the maze, the landmark parietal cells displayed a significantly higher activity than the HPC ones near intersections, (Figure S3B, see Methods). Thus, PPC neurons responded more to the gazing of landmarks at specific key-positions (i.e. following the trial start, and preceding and during choice point, i.e. centre), while HPC ones had a more homogeneous response along the paths. Next, the parietal path cells were significantly more active in the reward path, compared with the HPC ones (see Figure S3C), while the latter showed a more homogeneous activity, increasing until the reward was reached, and then decreasing along the outbounds. Again, the parietal cells displayed a higher FR in the centre compared to the HPC cells. The increased

activity in the centre of the maze, explains the modulation to all fixated maze paths see on Figure 3Q (see also Figure S1J-K), as camera rotations in the centre made the paths visible within the FOV. The results thus suggest that reward expectancy strongly modulated parietal's activity, leading to a higher activity along the specific path leading to reward."

The associate method is described page 39, line 1374:

"Comparison of place activity in the different maze segments. In the same way as for the PETH aligned on the start of the neuron's preferred maze portion (see Neuronal activity aligned on position), the activity of each cell was aligned on two events, separately, with a time-resolution of 1ms. First, the activity was aligned on the time the monkey reached the maze centre, and computed from the first push of the joystick, at the inbound path start, to the end of the first rotation in the maze centre. Second, it was aligned on the time the reward was reached, and computed from the beginning of the rewarded path, to the end of the outbound ones. Note that between reaching the reward location and the start of the returns along the outbounds, there was a stationary period, during which the monkey was given the liquid reward. To avoid excessive variation across trials, the duration of each maze segment was evaluated for all trials of all cells. Trials with durations that were too high or too low compared to the total sample (mean \pm 3SD) were discarded. The resulting averaged durations of the segments were: $1.31\pm 0.03s$ for the inbounds, $1.16\pm 0.04s$ for the centre, $1.21\pm 0.002s$ for the rewarded path, $1.21\pm 0.01s$ for the reward-delivery delay, and $2.56\pm 0.02s$ for the outbounds. Once the individual activities were obtained for each trial, they were averaged, smoothed (Gaussian kernel $SD=40ms$) and normalized (Max-Min method). Finally, the activities of all cells were averaged together to obtain the population mean activity. A two-sided Wilcoxon rank sum test was used to compare the activity of the PPC and the HPC, for each time-bin."

Finally, we also added a paragraph in the discussion to better highlight the main results brought by these new figures, page 13, line 525:

"Population activity allows to identify key-spatial-points in the parietal cortex, while hippocampal activity has less of a population average signature within segments. The spatial selectivity of parietal cells suggests that it may take part in decision and action-guidance through its role in attention-driven saliency maps (Bisley and Mirpour 2019; Levichkina et al. 2017). On the other hand, the HPC cells may play a role with landmark identification within a specific layout, as different individual cells are recruited for different views. The fact that the HPC population were mostly active during the returns along the outbounds additionally suggests that such landmark selectivity would play a specific role following outcome delivery, taking part in learning processes."

4.2. The different behavioral phases (entry path to the centre, exit path to the reward and passive return) are associated with different motivational and behavioral states (active versus passive; before versus after reward consumption) that could lead to different hippocampal spatial maps. Could you analyze the location selectivity of HPC and PPC activity separately for the different phases (inbound path to the centre, outbound path to the reward or return path)?

We agree with the reviewer's suggested interpretation of the findings. To refine and clarify the take home message relative to the differential recruitment of parietal and hippocampal cells, we performed a finer analysis across regions that allows to examine cells activity within segments. The new analysis, which has a finer resolution than the previous version, is presented in Figure S3C-D and replaces the previous Figures S3C-D. Specifically, we showed that when taking into account view selectivity for paths or landmarks, parietal and hippocampal cells are recruited in different locations in the maze

(Figure 3W-Z). Specifically, landmarks cells of both areas were recruited in the inbound paths, with the parietal ones more active when the animal was located towards the centre of the maze, and the paths cells were mostly active during the rewarded path and the outbounds, with the parietal cells more active in the rewarded path.

Further, to address the difference in recruitment within segments in parietal and hippocampal spatially-modulated cells as the reviewer suggested, we computed their information content (IC), depth of tuning index (DT) and sparsity index (S) for each segment (inbound, reward path, outbound paths), separately. Then, we separately conducted a 2-way-ANOVA on each of the computed indexes, with the neural area as fixed factor and the maze segments as repeated measures. Because of inequality of sizes across times per bin between the maze segments, we used DT and S indexes only to compare across segments. Thus, to analyse the IC, we only used the area factor.

The results showed that, when using the 2-way ANOVA, we confirmed the first result presented in the first section of the manuscript, as there was no significant effect of the area factor and thus there was no difference in the overall selectivity between the PPC and the HPC spatially-modulated cells, whether comparing their IC ($\mu IC_{PPC}=0.34\pm 0.048$, $\mu IC_{HPC}=0.45\pm 0.094$; $F_{1df}=2.15$, $p=0.14$), DT (DT: $\mu DT_{PPC}=0.76\pm 0.031$, $\mu DT_{HPC}=0.75\pm 0.039$; $F_{1df}=0$, $p=0.96$) or S (S: $\mu S_{PPC}=0.76\pm 0.022$, $\mu S_{HPC}=0.73\pm 0.035$; $F_{1df}=0.75$, $p=0.39$) indexes (see Methods in main manuscript for indexes calculation).

On the other hand, we found that both DT and S varied significantly according to maze segments (DT: $F_{2df}=327.29$, $p=9.74\times 10^{-77}$, S: $F_{2df}=166.33$, $p=1.60\times 10^{-49}$): selectivity was higher in the inbound ($\mu DT=0.86\pm 0.030$, $\mu S=0.69\pm 0.035$) and outbound ($\mu DT=0.87\pm 0.028$, $\mu S=0.69\pm 0.034$) paths than in the reward one ($\mu DT=0.54\pm 0.043$, $\mu S=0.85\pm 0.024$) for both the PPC and HPC cells. Thus, cells displayed a stronger modulation of amplitude as a function of position in the inbound and outbound paths than in the reward path. This suggests that the different views offered by the varying positions along the inbound and outbound paths triggers selective responses in the neurons, by contrast to the reward path, in which the view is pretty poor.

Finally, there was a significant interaction between the area and the maze segment factors for the DT variable only ($F_{2df}=3.31$, $p=0.038$), yet, the post-hoc comparisons did not reach significance level when comparing the two regions indexes in the rewarded path ($\mu DT_{PPC \times reward}=0.52\pm 0.051$, $\mu DT_{HPC \times reward}=0.56\pm 0.074$; $p=0.87$).

In sum, taken together with the data presented in Figure 3 and Figure S3, these results show that, while the PPC and the HPC don't are both as selective within the maze segments, both areas vary in selectivity across them. This fits the results described in result section 3, in which we showed that both PPC and HPC cells were modulated by the gazing of the different element of the environment available from different positions of the inbound or outbound paths.

We clarified results presented in Figure 3, by presenting the time-aligned population activity as a function of position in Figure S3, page 9 line 359 (see previous response R4.1).

We replaced the previous comparison between the spatial selectivity indexes of the two areas by this new combined analysis on selectivity across area and segments in the result section, page 4, line 118:

“ANOVAs with repeated-measures (see Methods), on various selectivity measures, such as the Information Content (IC; $\mu IC_{PPC}=0.34\pm 0.048$, $\mu IC_{HPC}=0.45\pm 0.094$; $F_{1df}=2.15$, $p=0.14$), sparsity index (S; $\mu S_{PPC}=0.76\pm 0.022$, $\mu S_{HPC}=0.73\pm 0.035$; $F_{1df}=0.75$, $p=0.39$), or depth of tuning index (DT; $\mu DT_{PPC}=0.76\pm 0.031$, $\mu DT_{HPC}=0.75\pm 0.039$; $F_{1df}=0.0025$, $p=0.96$) of the spatially modulated cells, showed that parietal neurons were as selective to virtual position as hippocampal cells. On the other hand, both S and DT (the analysis could not be performed on IC, since the index varies according to sample size) varied significantly according to maze segments (S: $F_{2df}=166.33$, $p=1.60\times 10^{-49}$, DT: $F_{2df}=327.29$,

$p=9.74 \times 10^{-77}$): selectivity was higher in the inbound ($\mu S=0.69 \pm 0.035$, $\mu DT=0.86 \pm 0.030$) and outbound ($\mu S=0.69 \pm 0.034$, $\mu DT=0.87 \pm 0.028$) paths than in the reward one ($\mu S=0.85 \pm 0.024$, $\mu DT=0.54 \pm 0.043$) for both the PPC and HPC cells. Thus, cells displayed a stronger modulation of amplitude as a function of position in the inbound and outbound paths than in the reward path.”

and page 5, line 195:

“Finally, varying positions in the inbound and outbound paths triggered stronger selectivities in both areas, compared with the reward path, where the visual scene was poorer.”

We as well included new paragraphs in the Methods sections pages 36-37 to describe how we assessed spatial stability and selectivity, computed neural activity on position and compared them across different maze segments.

We also wanted to clarify that, thanks to reviewer #2's comments, we re-processed the eye data and updated our algorithm for detecting saccades and fixations. In addition to noticing that our cut-off was too stringent for assessing correlations between target of gaze and firing rates, the results of individual correlations changed marginally as a result of the re-analysis, hence some of the figure exact dot to dot do not match, although the overall results with respect to point of gaze are the same (for more details, see response R2.9).

4.3. HPC cells have previously been described as 'spatial view cells', but it is unclear from the present results whether the HPC cells recorded have this property. Were HPC cells modulated by where the animal was looking at in a manner reminiscent of spatial view cells? Could you discuss this point?

Spatial view cells have been discovered in square environments (3x3 room) with the same floor surface and floor rendering across different positions in the room. In our virtual star maze, different virtual positions often bear different views except on inbound arms. But even then, the floor changes as the animal moves forward towards the centre. Combining our different analysis, our results show that 1) cells are selective to different views (landmarks or paths, Figure 3) and specific landmarks identity, (Figure 4), and 2) that cells are selective to different portions of the maze (Figure 3W-Z, and S3C-D). So, while, we cannot apply the exact same kind of analysis as in Georges-François *et al.* (1999), we can conclude that cells are sensitive to the landmarks viewing in line with our previous findings (Wirth *et al.* 2017) and that this activity is sensitive to the position from which the landmark is viewed. Our global interpretation is that cells encode the landmark presence from specific locations within segments (inbound paths, centre, outbounds). We discussed previously that the specific task demand differing in Rolls' laboratory and our VR task, may lead to the differences between the spatial view and our view cells (Rolls and Wirth 2018), but that in contrast to rodent place cells, primate hippocampal view cells encode what the animal views rather than where it is (Rolls and Wirth 2018; Wirth 2023). Our finding is nuanced with respect to Rolls interpretation of pure view cells, but we should however note, that modulation of spatial view was also previously shown to be modulated by the position even in Rolls room set-up (Rolls and O'Mara 1995) and that recent findings in the freely moving marmoset corroborate ours (Piza *et al.* 2024) and show that hippocampal cells are sensitive to a combination of place and view. We shall stress out that in the current paper, the added value is the comparison of hippocampal cells with the cells in the lateral-ventral bank of the intraparietal sulcus. The results show that while parietal and hippocampal cells share many features such as a recruitment by saccades, and fixations, the locations in the maze in which the cells are recruited differs, which suggests that these cells fulfil different functions.

We now further discuss the similitude and difference with spatial view cells, page 13, line 496:

“Our results show that the previously identified link between hippocampal activity and view (Corrigan et al. 2023; Doucet et al. 2020; Hoffman 2013; Jutras et al. 2013; Wirth et al. 2017) is weaker than in the parietal cortex. Thus, although hippocampus receives input from the parietal cortex (Kravitz et al. 2011; Rockland and Van Hoesen 1999; Selemon and Goldman-Rakic 1988), its activity is not merely inherited from the parietal cortex. Rather, it does not simply reflect the presence of a spatially or behaviourally informative object in the visual receptive field, but cells may be sensitive to a combination of visual elements. The hippocampal cells bear similitudes with previously described spatial view cells (Georges-François et al. 1999; Robertson et al. 1998; Rolls 1999; Rolls and Wirth 2018), as they can be selective to different views or landmarks. However, we show that they also integrate task sensitive variables (Piza et al. 2024; Wirth et al. 2017). Our interpretation is that unlike spatial view cells, which are position invariant, cells recorded in our task encode the landmark presence from specific position within segments (inbound paths or centre or outbounds). This is consistent with our previous conclusions (Wirth et al. 2017) and recent findings obtained in the freely moving marmoset (Piza et al. 2024) or macaques (Mao et al. 2021) which show mixed selectivity between view and place.”

and place cells, page 12 line 475:

“We suggest that the apparent parietal position code can result from visual processing of relevant cues, contributing to a task-relevant representation of visual space, rather than pure place code for self-position in Euclidian coordinates. Further, we also extend the latter interpretation for the position selectivity found in hippocampal cells in this task. Indeed, unlike rodents place cells (Andersen et al. 2006, but see Acharya et al. 2016; Jercog et al. 2019; O’Keefe and Dostrovsky 1971; Rubin et al. 2014), cells we recorded displayed small and unstable spatial fields and orientation selectivity in the centre of the maze, compatible with an active task-dependent visual processing of the scene. “

4.4. Previous work in freely moving monkeys has shown that HPC activity is strongly modulated by the direction the animal is looking at (in 3D space) and by head tilt. However, in the present experiment, several head movements are restricted by head fixation. The authors should discuss how this behavioral constraint might affect the results they observed.

Yes, indeed, the animal’s head is fixed during virtual navigation in our set up, which prevents us to test the effect of head tilt, which in turn may influence opto-vestibular control to adjust visual retinal stability and support corresponding eye rotation. Further, the fixation of the head may also alter the behavioural sequence during which head turns are combined with eye movements in order to reach the target of a fixation (Goonetilleke et al. 2011; Kothari et al. 2020; McCluskey and Cullen 2007; Freedman and Sparks 1997). Therefore, the natural flow of head-turns, saccades and fixations may be altered in our task leading to larger saccades. See also response to Reviewer #2.9 which describes the dynamics of the saccades observed in our task compared to other tasks and page 6 line 206 in the manuscript:

“The long duration of saccades can be explained by the structural layout of the task maze, in which animals freely moved eyes from one part of the screen to another, as elements (landmarks, paths) of the environment entered or exited the field of view (see also Corrigan et al. 2017).”

4.5. It is unclear whether the sampling of different locations by the animals was uniform or biased. The authors report two experimental phases each day: one in which the animals learn the location of the reward location and presumably sample each arm of the maze in search of the reward, and a phase in which the reward location is learned and the animals presumably mostly visit the rewarded arm. Could the authors show examples of trajectories made by the animal at different stages of training? When was the modulation of PPC and HC cells by location assessed? If there was a sampling bias, could this have affected the reported results?

Before starting the recordings, animals were trained to find a reward location. On average, they learned the reward location in 12 trials, and then performing only correctly for about 80 trials. We only based our analysis on these correct trials resulting in 20 trials per inbound and outbound paths, and 80 laps for the reward path (see also response R3.1). We now clarified that we only used correct trials in the result section page 3 line 105:

“For this study, only the correct trials were selected for analyses. In this way, the path chosen by the animal was always the reward path, oriented toward the North on our maps.”

For both regions, activity was analysed in sessions with a comparable number of correct trials, leading to no specific bias as the activity was consistently analysed across regions. We previously provided a description of animal’s learning behaviour in a former paper (Baraduc *et al.* 2019) in which some trial-by-trial data can be found in the supplementary showing that when we recorded, animals had mastered the maze. Animals actually only made very few errors once they had learned because they probably acquired a strong schema or learning-set of the task. Further, in this task, animals did not have a full 2-D control of the joystick, and could only move forward as a constant speed, or left or right as choice points. Thus, while an analysis of trajectory would be interesting, the constraint on the joystick may not allow to reveal much.

4.6. Along the same lines, animals could use different navigational strategies at different stages of learning the reward location. For example, navigation could be based more on the external landmark at the beginning of the learning (place strategy) and on movement sequences in later phases (when animals only use landmarks to know the starting location then use only sequences of movements to reach the correct arm). Could the authors compare the occurrence of saccades and fixations before versus after the learning of reward location? Similarly, could the authors compare the modulation of PPC and HC cells by location, saccades, landmarks and path before versus after the learning of reward location? This is important as HC cells might be more modulated by position in an allocentric reference frame early during learning, but more modulated by position in an egocentric reference frame late during learning.

Yes, these are very interesting questions, and one may expect indeed that there may be a behavioural change and an adaptation of navigational strategy with learning. We agree that examining both behavioural and neural learning related changes would be of great value but we hope that the reviewer understands that this may be beyond the scope of the current paper.

Minor:

4.7. Could you show the recording sites for hippocampal recordings as well as in Fig. S1?

We clarify that recording sites can be found in Baraduc *et al.* 2019, and Wirth *et al.*, 2017. These recordings were made in the anterior 2 thirds of the hippocampus, which is the anatomical homolog of the ventral rodent hippocampus.

4.8. Fig. S1 G rightmost example, it seems to me that this cell is symmetrical for outgoing paths?

Figure S1G presents cells that are asymmetric in *inbound paths* specifically. This does not exclude a potential symmetry in the rest of the maze. We chose to present the full figure, including all maze segments, in a concern of thoroughness. With respect to the outbound path, in line with the reviewer's observation, the asymmetry was not significant ($SI_o=0.12$, Wilcoxon rank sum: $p=0.15$). To be more precise, we chose to represent in Figure S1G-H cells expressing asymmetry only in the inbound paths, and in Figure S1I-J cells asymmetric only in the outbound paths.

We clarified this information in the legend of Figure S1, page 46, line 1467 and 1469, adding "*asymmetrical activity in the inbound/outbound paths only*".

4.9. Were there any changes in theta rhythm activity between incoming (active) and outgoing (passive) pathways?

Sadly, we did not collect the LFP for this data and cannot answer the reviewer's question. However, other researchers have described that monkey's theta rhythm was different in the monkey compared to rodents, and only exhibited un-sustained theta bouts aligned to saccade or head shifts (Hoffman 2013; Mao *et al.* 2021; Piza *et al.* 2024). This may well be because of different strategies (visual versus body motion) in active exploration across species.

4.10. Typo in P14, first paragraph, sentence beginning with "Our results are consistent with "this" hypothesis" should be "the" hypothesis.

Thank you for catching this.

4.11. Could you report the information content sparsity index and depth of tuning for each illustrated cell in main and supplementary figures.

Yes. These are now included in the Figure 1 and S1 under the form of Table 1, reproduced below:

Figure	IC	S	DT
1B	0.51	0.56	1.00
1C	0.17	0.77	0.89
1D	0.19	0.73	0.91
1E	0.14	0.80	0.86
1F	0.32	0.81	0.85
1G	0.51	0.40	1.00
1H	1.34	0.23	1.00
1I	0.09	0.93	0.60
1J	1.71	0.21	1.00
1K	0.08	0.90	0.88
1L	0.86	0.25	1.00
1M	0.12	0.90	0.82
S1Ca	0.97	0.28	1.00
S1Cb	0.19	0.79	1.00
S1Da	0.30	0.72	1.00
S1Db	0.73	0.40	1.00
S1La	0.12	0.85	0.82
S1Lb	0.66	0.38	1.00
S1Lc	0.56	0.52	1.00
S1Ld	0.20	0.76	0.88
S1Ma	0.17	0.83	0.87
S1Mb	0.06	0.92	0.59
S1Mc	0.08	0.90	0.88
S1Md	0.02	0.97	0.39
S1Na	0.23	0.76	0.88
S1Nb	0.05	0.92	0.63
S1Nc	0.62	0.48	1.00
S1Nd	0.41	0.63	1.00
S1Oa	0.24	0.69	1.00
S1Ob	0.14	0.84	0.90
S1Oc	0.09	0.84	0.86
S1Od	0.05	0.88	0.74
S1Pa	0.09	0.88	0.71
S1Pb	0.92	0.26	1.00
S1Pc	0.55	0.51	1.00
S1Pd	0.14	0.78	1.00
S1Qa	0.18	0.65	1.00
S1Qb	0.09	0.95	0.50
S1Qc	0.62	0.37	1.00
S1Qd	0.85	0.36	1.00

Table R4.11. Summary table indicating, for each cell of which the neural place map is represented in Figure 1 or S1, their information content (IC), sparsity index (S) and depth of tuning index (DT).

References

- Acharya L., Aghajan Z. M., Vuong C., Moore J. J., Mehta M. R. (2016) Causal Influence of Visual Cues on Hippocampal Directional Selectivity. *Cell* **164**, 197–207.
- Andersen P., Morris R., Amaral D., Bliss T., O'Keefe J. (2006) *The Hippocampus Book*. Oxford University Press.
- Aronov D., Tank D. W. (2014) Engagement of Neural Circuits Underlying 2D Spatial Navigation in a Rodent Virtual Reality System. *Neuron* **84**, 442–456.
- Bahill A. T., Clark M. R., Stark L. (1975) The main sequence, a tool for studying human eye movements. *Mathematical Biosciences* **24**, 191–204.
- Baraduc P., Duhamel J.-R., Wirth S. (2019) Schema cells in the macaque hippocampus. *Science* **363**, 635–639.
- Barash S., Bracewell R. M., Fogassi L., Gnadt J. W., Andersen R. A. (1991a) Saccade-related activity in the lateral intraparietal area. II. Spatial properties. *Journal of Neurophysiology* **66**, 1109–1124.
- Barash S., Bracewell R. M., Fogassi L., Gnadt J. W., Andersen R. A. (1991b) Saccade-related activity in the lateral intraparietal area. I. Temporal properties; comparison with area 7a. *Journal of Neurophysiology* **66**, 1095–1108.
- Ben Hamed S., Duhamel J.-R., Bremmer F., Graf W. (2001) Representation of the visual field in the lateral intraparietal area of macaque monkeys: a quantitative receptive field analysis. *Exp Brain Res* **140**, 127–144.
- Bendiksy M. S., Platt M. L. (2006) Neural correlates of reward and attention in macaque area LIP. *Neuropsychologia* **44**, 2411–2420.
- Bischof W. F., Anderson N. C., Doswell M. T., Kingstone A. (2020) Visual exploration of omnidirectional panoramic scenes. *Journal of Vision* **20**, 23.
- Bisley J. W., Mirpour K. (2019) The neural instantiation of a priority map. *Current Opinion in Psychology* **29**, 108–112.
- Blatt G. J., Andersen R. A., Stoner G. R. (1990) Visual receptive field organization and cortico-cortical connections of the lateral intraparietal area (area LIP) in the macaque. *J. Comp. Neurol.* **299**, 421–445.
- Bremmer F., Distler C., Hoffmann K.-P. (1997) Eye Position Effects in Monkey Cortex. II. Pursuit- and Fixation-Related Activity in Posterior Parietal Areas LIP and 7A. *Journal of Neurophysiology* **77**, 962–977.
- Bremmer F., Duhamel J.-R., Hamed S. B., Graf W. (2002a) Heading encoding in the macaque ventral intraparietal area (VIP). *European Journal of Neuroscience* **16**, 1554–1568.
- Bremmer F., Klam F., Duhamel J.-R., Hamed S. B., Graf W. (2002b) Visual–vestibular interactive responses in the macaque ventral intraparietal area (VIP). *European Journal of Neuroscience* **16**, 1569–1586.
- Bremmer F., Kubischik M., Hoffmann K.-P., Krekelberg B. (2009) Neural Dynamics of Saccadic Suppression. *J. Neurosci.* **29**, 12374–12383.
- Bremmer F., Schlack A., Kaminiarz A., Hoffmann K.-P. (2013) Encoding of movement in near extrapersonal space in primate area VIP. *Front. Behav. Neurosci.* **7**.
- Chen A., DeAngelis G. C., Angelaki D. E. (2011) Representation of Vestibular and Visual Cues to Self-Motion in Ventral Intraparietal Cortex. *Journal of Neuroscience* **31**, 12036–12052.

- Colby C. L., Duhamel J.-R. (1991) Heterogeneity of extrastriate visual areas and multiple parietal areas in the Macaque monkey. *Neuropsychologia* **29**, 517–537.
- Corrigan B. W., Gulli R. A., Doucet G., Mahmoudian B., Abbass M., Roussy M., Luna R., Sachs A. J., Martinez-Trujillo J. C. (2023) View cells in the hippocampus and prefrontal cortex of macaques during virtual navigation. *Hippocampus* **33**, 573–585.
- Corrigan B. W., Gulli R. A., Doucet G., Martinez-Trujillo J. C. (2017) Characterizing eye movement behaviors and kinematics of non-human primates during virtual navigation tasks. *Journal of Vision* **17**, 15.
- Derdikman D., Whitlock J. R., Tsao A., Fyhn M., Hafting T., Moser M.-B., Moser E. I. (2009) Fragmentation of grid cell maps in a multicompartment environment. *Nat Neurosci* **12**, 1325–1332.
- Dick S., Ostendorf F., Kraft A., Ploner C. J. (2004) Saccades to spatially extended targets: the role of eccentricity. *Neuroreport* **15**, 453–456.
- Doucet G., Gulli R. A., Corrigan B. W., Duong L. R., Martinez-Trujillo J. C. (2020) Modulation of local field potentials and neuronal activity in primate hippocampus during saccades. *Hippocampus* **30**, 192–209.
- Duhamel, Colby C. L., Goldberg M. E. (1992) The updating of the representation of visual space in parietal cortex by intended eye movements. *Science* **255**, 90–92.
- Duhamel J.-R., Bremmer F., Hamed S. B., Graf W. (1997) Spatial invariance of visual receptive fields in parietal cortex neurons. *Nature* **389**, 845.
- Duhamel J.-R., Colby C. L., Goldberg M. E. (1998) Ventral Intraparietal Area of the Macaque: Congruent Visual and Somatic Response Properties. *Journal of Neurophysiology* **79**, 126–136.
- Fanini A., Assad J. A. (2009) Direction Selectivity of Neurons in the Macaque Lateral Intraparietal Area. *Journal of Neurophysiology* **101**, 289–305.
- Frank L. M., Brown E. N., Wilson M. (2000) Trajectory Encoding in the Hippocampus and Entorhinal Cortex. *Neuron* **27**, 169–178.
- Freedman E. G., Sparks D. L. (1997) Eye-Head Coordination During Head-Unrestrained Gaze Shifts in Rhesus Monkeys. *Journal of Neurophysiology* **77**, 2328–2348.
- Fuchs A. F. (1967) Saccadic and smooth pursuit eye movements in the monkey. *J Physiol* **191**, 609–631.
- Furuya Y., Matsumoto J., Hori E., Boas C. V., Tran A. H., Shimada Y., Ono T., Nishijo H. (2014) Place-related neuronal activity in the monkey parahippocampal gyrus and hippocampal formation during virtual navigation: PPA and Hippocampal Responses to Place and view. *Hippocampus* **24**, 113–130.
- Garbutt S., Harwood M. R., Harris C. M. (2001) Comparison of the main sequence of reflexive saccades and the quick phases of optokinetic nystagmus. *British Journal of Ophthalmology* **85**, 1477–1483.
- Georges-François P., Rolls E. T., Robertson R. G. (1999) Spatial View Cells in the Primate Hippocampus: Allocentric View not Head Direction or Eye Position or Place. *Cerebral Cortex* **9**, 197–212.
- Gnadt J. W., Andersen R. A. (1988) Memory related motor planning activity in posterior parietal cortex of macaque. *Exp Brain Res* **70**, 216–220.
- Goonetilleke S. C., Gribble P. L., Mirsattari S. M., Doherty T. J., Corneil B. D. (2011) Neck muscle responses evoked by transcranial magnetic stimulation of the human frontal eye fields. *European Journal of Neuroscience* **33**, 2155–2167.

- Grefkes C., Fink G. R. (2005) The functional organization of the intraparietal sulcus in humans and monkeys. *J Anat* **207**, 3–17.
- Grunewald A., Linden J. F., Andersen R. A. (1999) Responses to Auditory Stimuli in Macaque Lateral Intraparietal Area I. Effects of Training. *Journal of Neurophysiology* **82**, 330–342.
- Hagan M. A., Dean H. L., Pesaran B. (2012) Spike-field activity in parietal area LIP during coordinated reach and saccade movements. *J Neurophysiol* **107**, 1275–1290.
- Hayhoe M., Ballard D. (2005) Eye movements in natural behavior. *Trends in Cognitive Sciences* **9**, 188–194.
- Heiser L. M., Colby C. L. (2006) Spatial Updating in Area LIP Is Independent of Saccade Direction. *Journal of Neurophysiology* **95**, 2751–2767.
- Henderson J. M. (2003) Human gaze control during real-world scene perception. *Trends in Cognitive Sciences* **7**, 498–504.
- Hoffman K. L. (2013) Saccades during visual exploration align hippocampal 3-8Hz rhythms in human and non-human primates. *Frontiers in Systems Neuroscience*, 10.
- Ipata A. E., Gee A. L., Bisley J. W., Goldberg M. E. (2009) Neurons in the lateral intraparietal area create a priority map by the combination of disparate signals. *Exp Brain Res* **192**, 479–488.
- Jercog P. E., Ahmadian Y., Woodruff C., Deb-Sen R., Abbott L. F., Kandel E. R. (2019) Heading direction with respect to a reference point modulates place-cell activity. *Nat Commun* **10**, 2333.
- Johnston W. J., Tetrick S. M., Freedman D. J. (2022) *The lateral intraparietal area preferentially supports stimulus selection in directed tasks compared to undirected free behavior.* bioRxiv.
- Jutras M. J., Fries P., Buffalo E. A. (2013) Oscillatory activity in the monkey hippocampus during visual exploration and memory formation. *Proceedings of the National Academy of Sciences* **110**, 13144–13149.
- Katz C. N., Schjetnan A. G. P., Patel K., Barkley V., Hoffman K. L., Kalia S. K., Duncan K. D., Valiante T. A. (2022) A corollary discharge mediates saccade-related inhibition of single units in mnemonic structures of the human brain. *Current Biology* **32**, 3082-3094.e4.
- Killian N. J., Jutras M. J., Buffalo E. A. (2012) A map of visual space in the primate entorhinal cortex. *Nature (Lond.)* **491**, 761–764.
- Kothari R., Yang Z., Kanan C., Bailey R., Pelz J. B., Diaz G. J. (2020) Gaze-in-wild: A dataset for studying eye and head coordination in everyday activities. *Sci Rep* **10**, 2539.
- Kravitz D. J., Saleem K. S., Baker C. I., Mishkin M. (2011) A new neural framework for visuospatial processing. *Nature Reviews Neuroscience* **12**, 217–230.
- Lakshminarasimhan K. J., Avila E., Neyhart E., DeAngelis G. C., Pitkow X., Angelaki D. E. (2020) Tracking the Mind’s Eye: Primate Gaze Behavior during Virtual Visuomotor Navigation Reflects Belief Dynamics. *Neuron* **106**, 662-674.e5.
- Leathers M. L., Olson C. R. (2012) In monkeys making value-based decisions, LIP neurons encode cue salience and not action value. *Science* **338**, 132–135.
- Levichkina E., Saalman Y. B., Vidyasagar T. R. (2017) Coding of spatial attention priorities and object features in the macaque lateral intraparietal cortex. *Physiol Rep* **5**.
- Mao D., Avila E., Caziot B., Laurens J., Dickman J. D., Angelaki D. E. (2021) Spatial modulation of hippocampal activity in freely moving macaques. *Neuron* **109**, 3521-3534.e6.

- Mazzoni P., Bracewell R. M., Barash S., Andersen R. A. (1996) Spatially tuned auditory responses in area LIP of macaques performing delayed memory saccades to acoustic targets. *Journal of Neurophysiology* **75**, 1233–1241.
- McCluskey M. K., Cullen K. E. (2007) Eye, Head, and Body Coordination During Large Gaze Shifts in Rhesus Monkeys: Movement Kinematics and the Influence of Posture. *Journal of Neurophysiology* **97**, 2976–2991.
- Munuera J., Duhamel J.-R. (2020) The role of the posterior parietal cortex in saccadic error processing. *Brain Struct Funct* **225**, 763–784.
- Nitz D. A. (2006) Tracking Route Progression in the Posterior Parietal Cortex. *Neuron* **49**, 747–756.
- O’Keefe J. (1979) A review of the hippocampal place cells. *Progress in Neurobiology* **13**, 419–439.
- O’Keefe J., Dostrovsky J. (1971) The hippocampus as a spatial map. Preliminary evidence from unit activity in the freely-moving rat. *Brain Research* **34**, 171–175.
- Opstal A. J. van, Kasap B. (2019) Chapter 2 - Maps and sensorimotor transformations for eye-head gaze shifts: Role of the midbrain superior colliculus, in *Progress in Brain Research*, (Ramat S., Shaikh A. G., eds), Vol. 249, pp. 19–33. Elsevier.
- Paré M., Wurtz R. H. (1997) Monkey Posterior Parietal Cortex Neurons Antidromically Activated From Superior Colliculus. *Journal of Neurophysiology* **78**, 3493–3497.
- Piza D. B., Corrigan B. W., Gulli R. A., Do Carmo S., Cuello A. C., Muller L., Martinez-Trujillo J. (2024) Primacy of vision shapes behavioral strategies and neural substrates of spatial navigation in marmoset hippocampus. *Nat Commun* **15**, 4053.
- Platt M. L., Glimcher P. W. (1999) Neural correlates of decision variables in parietal cortex. *Nature* **400**, 233–238.
- Rao V., DeAngelis G. C., Snyder L. H. (2012) Neural correlates of prior expectations of motion in the lateral intraparietal and middle temporal areas. *J. Neurosci.* **32**, 10063–10074.
- Rayner K. (2009) The 35th Sir Frederick Bartlett Lecture: Eye movements and attention in reading, scene perception, and visual search. *Quarterly Journal of Experimental Psychology* **62**, 1457–1506.
- Rayner K., Pollatsek A. (1992) Eye movements and scene perception. *Canadian Journal of Psychology / Revue canadienne de psychologie* **46**, 342–376.
- Robertson R. G., Rolls E. T., Georges-François P. (1998) Spatial View Cells in the Primate Hippocampus: Effects of Removal of View Details. *Journal of Neurophysiology* **79**, 1145–1156.
- Rockland K. S., Van Hoesen G. W. (1999) Some Temporal and Parietal Cortical Connections Converge in CA1 of the Primate Hippocampus. *Cereb Cortex* **9**, 232–237.
- Rolls E. T. (1999) Spatial view cells and the representation of place in the primate hippocampus. *Hippocampus* **9**, 467–480.
- Rolls E. T., O’Mara S. M. (1995) View-responsive neurons in the primate hippocampal complex. *Hippocampus* **5**, 409–424.
- Rolls E. T., Wirth S. (2018) Spatial representations in the primate hippocampus, and their functions in memory and navigation. *Progress in Neurobiology* **171**, 90–113.
- Rubin A., Yartsev M. M., Ulanovsky N. (2014) Encoding of head direction by hippocampal place cells in bats. *J. Neurosci.* **34**, 1067–1080.
- Selemon L. D., Goldman-Rakic P. S. (1988) Common cortical and subcortical targets of the dorsolateral prefrontal and posterior parietal cortices in the rhesus monkey: evidence for a distributed neural network subserving spatially guided behavior. *J. Neurosci.* **8**, 4049–4068.

- Sereno M. I., Pitzalis S., Martinez A. (2001) Mapping of Contralateral Space in Retinotopic Coordinates by a Parietal Cortical Area in Humans. *Science* **294**, 1350–1354.
- Shelley L. E., Nitz D. A. (2021) Locomotor action sequences impact the scale of representation in hippocampus and posterior parietal cortex. *Hippocampus* **31**, 677–689.
- Steenrod S. C., Phillips M. H., Goldberg M. E. (2013) The lateral intraparietal area codes the location of saccade targets and not the dimension of the saccades that will be made to acquire them. *Journal of Neurophysiology* **109**, 2596–2605.
- Sunkara A., DeAngelis G. C., Angelaki D. E. (2016) Joint representation of translational and rotational components of optic flow in parietal cortex. *Proceedings of the National Academy of Sciences* **113**, 5077–5082.
- Swaminathan S. K., Masse N. Y., Freedman D. J. (2013) A Comparison of Lateral and Medial Intraparietal Areas during a Visual Categorization Task. *J Neurosci* **33**, 13157–13170.
- Viswanathan P., Nieder A. (2017) Comparison of visual receptive fields in the dorsolateral prefrontal cortex and ventral intraparietal area in macaques. *European Journal of Neuroscience* **46**, 2702–2712.
- Wardak C., Olivier E., Duhamel J.-R. (2002) Saccadic Target Selection Deficits after Lateral Intraparietal Area Inactivation in Monkeys. *J. Neurosci.* **22**, 9877–9884.
- Wirth S. (2023) A place with a view: A first-person perspective in the hippocampal memory space. *Hippocampus* **33**, 658–666.
- Wirth S., Baraduc P., Planté A., Pinède S., Duhamel J.-R. (2017) Gaze-informed, task-situated representation of space in primate hippocampus during virtual navigation. *PLOS Biology* **15**, e2001045.
- Zhang T., Malevich T., Baumann M. P., Hafed Z. M. (2022) Superior colliculus saccade motor bursts do not dictate movement kinematics. *Commun Biol* **5**, 1–14.
- Zhou Y., Liu Y., Wu S., Zhang M. (2018) Neuronal Representation of the Saccadic Timing Signals in Macaque Lateral Intraparietal Area. *Cereb Cortex* **28**, 2887–2900.
- Zhu S., Lakshminarasimhan K. J., Arfaei N., Angelaki D. E. (2022) Eye movements reveal spatiotemporal dynamics of visually-informed planning in navigation. *eLife* **11**, e73097.

Reviewer #1 (Remarks to the Author):

The revised manuscript by Vericel introduces few new analyses and no new data. As a result, I find myself less convinced than before that the work can deliver a well-evidenced message that reveals something new about hippocampal (HPC) and posterior parietal cortex (PPC) function in navigation.

1.1. As before, there are too few neurons recorded in PPC and HPC to reach meaningful comparisons between the 2 regions. The authors point to technical reasons for low neuron numbers, but none stand as valid reasons to accept a less than reasonably complete dataset. To add to this, the observed differences are not large even where they are statistically significant and they sometimes concern values (correlations or information) that indicate less than robust tuning of spiking activity. Combined, these reasons make it impossible to make strong statements about complementary function between the two regions.

The observed information values are low. The 1st-2nd half reliability correlations for spatial firing patterns are low, especially as compared to data from track environments in rodents for both PPC and HPC. The authors chose not to invoke a coherence analysis to complement the information metric despite and cite incorrect reasons as to why it cannot be carried out. The correlations of neural activity for different orientations to the path versus landmark sinusoid are low. Saccade and landmark (equivalent to object) responses of primate PPC neurons are not novel findings. The manuscript lacks a coherent message that is strongly evidenced.

We take note of the reviewer's comment. We would like to point out that all results reached statistical significance. In spite of the low number of neurons, this actually demonstrates robustness of our results, since we obtained significant results without requiring a high statistical power. We want to emphasize that the results bring evidence with respect to saccades and fixations in a goal-directed navigation task, which to our knowledge is new.

1.2. Line 160 – In a variety of studies (e.g., Johnson/Redish J. Neurosci.), orientation-specific responses of place-specific neurons have been observed. The statement here is incorrect.

Actually, this is consistent with our statement, which suggests that the “classical” rodents place cell model, described for decades in the literature, is orientation-independent, yet more recent works evidenced some orientation selectivity (“*but see...*”).

However, we don't think that the reference Johnson and Redish (2007) is appropriate within the paragraph. The paper presents an analysis where the “orientation of motion” is correlated with path representation by neural assemblies. However, we believe that the other references we cited fit more with our point, as they display single unit activities modulated by the rodent's orientation.

Reviewer #2 (Remarks to the Author):

The authors have revised their manuscript substantially along the points raised by the reviewers. Speaking for those points raised in my review: the authors have addressed all of them carefully and convincingly. I have no further comments.

Reviewer #3 (Remarks to the Author):

The authors addressed all my concerns appropriately. They did a quite good job, and the manuscript deserves for the publication.

Reviewer #4 (Remarks to the Author):

In their revised manuscript, Vericel et al. have included several new analyses that support their results and strengthen the manuscript. In particular, they better characterized the reliability of the spatially modulated firing of HPC and PPC cells by plotting rasters, and several examples convincingly show that the activity of these cells is indeed spatially modulated, as they consistently discharge when animals pass through the same virtual locations. They also correlated the spatial stability of firing (assessed by correlating neural maps between the first and second half of the session) with the number of fields, field size, and firing rate peaks, and found correlations consistent with higher stability of more spatially modulated cells (cells with a low number of place fields and high firing rates).

4.1. However, I found the interpretation of the results as presented in the discussion unclear P12 Ln 475 “We suggest that the apparent parietal position code can result from visual processing of relevant cues, contributing to a task-relevant representation of visual space, rather than pure place code for self-position in Euclidian coordinates. Further, we also extend the latter interpretation for the position selectivity found in hippocampal cells in this task. Indeed, unlike rodents place cells (Andersen et al. 2006, but see Acharya et al. 2016; Jercog et al. 2019; O’Keefe and Dostrovsky 1971; Rubin et al. 2014), cells we recorded displayed small and unstable spatial fields and orientation selectivity in the centre of the maze, compatible with an active task-dependent visual processing of the scene.” It is unclear to me why small and unstable fields would be more compatible with active task-dependent visual processing of the scene than, say, large and stable place fields. To me, active and task-dependent visual processing indicates that the animals are engaged in the task, and this should be associated with higher and more stable spatial modulation in the hippocampus, as recently shown in rodents (Pettit et al., Nat Neurosci 2022, doi: 10.1038/s41593-022-01050-4).

We thank the reviewer for this comment. We altered the paragraph in the discussion to convey that view fields are more dependent on the behavior (where the animal looks at in time) rather than where it is, and this may therefore be more labile as gaze behaviour is brief and less stereotyped than laps on an alley.

Page 12, line 481: *“Further, we also extend the latter interpretation for the position selectivity found in hippocampal cells in this task. Indeed, unlike rodents place cells (Andersen et al. 2006, though recent works have evidenced orientation-modulated responses in hippocampal cells of rodents: Acharya et al. 2016; Jercog et al. 2019; O’Keefe and Dostrovsky 1971; Rubin et al. 2014), cells we recorded displayed small and unstable spatial fields and orientation selectivity in the centre of the maze. This is compatible with an active task-dependent visual processing of the scene, as gaze behaviour is highly dynamic and consists of brief saccades and fixations toward visual cues. “*

The authors also included a more detailed analysis of the activity of PPC and HPC cells in different paths of the track and found that landmark cells are more recruited in the inbound paths and path cells are mostly active during the rewarded path and the outbound paths. This much more detailed analysis adds significant new information to the manuscript.

Overall, the authors have performed new analyses and further discussed their results in relation to the previous literature.

To conclude the authors have addressed my concerns with new analyses and discussion and the manuscript has been strengthened as a result.

Minor:

4.2. P9 Ln348 first sentence. The sentence is confusing. First, it is unclear what the "partition in two" means, and we have to refer to the Figure 3S-V legend to figure it out, which is not convenient. Also, it is not clear to me how plotting landmark and path responsive cells separately will tell us about the relationship between view response and virtual position.

Thank you for the comment. The reasoning is two-fold: 1) we identify cells by their functional properties with respect to the view: this shows that cells distribute in a continuum between landmarks and paths-selectivities. 2) we parted the population in two depending on their landmark or path preference, and then computed the place map corresponding to these population of cells. This shows us, where in the maze cells are active depending on what visual features they respond to.

We clarified this page 9, line 348: *“In order to examine the relationship between neurons’ responses to view and the virtual position, our approach was two-fold: after having characterized the cells based on their visual selectivities and parted them into two populations depending on their landmark or path preference, as described above, we computed the average neural place map corresponding to these populations. These maps would evidence where in the maze the cells were active according to the visual features they preferentially responded to.”*

Reviewer #4 (Remarks on code availability):

I had to ask for authorization to access the code which I am still waiting for.

We apologize for the delay in making the codes available. Since the reviewers’ responses, the codes and data have been made publically available at this URL: <https://osf.io/eqp6z/>.